# Deforestation-induced drying lowers Amazon climate threshold

Nico Wunderling[1,2,3 ✉], Boris Sakschewski[2], Johan Rockström[2,4,5], Bernardo M. Flores[6,7], Marina Hirota[8,9,10] & Arie Staal[11 ✉]

Humanity is putting unprecedented pressures on the Amazon forest system through global warming and land use changes[1,2]. As the Amazon forest may undergo self-reinforcing transitions, these pressures could lead to system-wide changes across major parts of Amazonian ecosystems[1-4]. Here we apply a dynamical systems model to assess the local and far-reaching cascading transition risks towards degraded ecosystems in the Amazon biome under different Shared Socioeconomic Pathways. For these emission scenarios, we constructed how moisture is transported through the atmosphere within the Amazon basin using an established atmospheric moisture-tracking model[5]. Without accounting for deforestation, we find a critical global warming threshold of 3.7–4.0 °C, beyond which up to a third of the Amazon forest risks losing stability. However, when considering deforestation, we find a near system-wide transition of the Amazon forest (62–77% of the area) under the combination of a lower threshold range of global warming of 1.5–1.9 °C and deforestation of 22–28%. The large majority of the simulated transitions is caused by spatial knock-on effects from increasing drought intensities, leading to long-ranging and self-propelling cascades on scales of hundreds to thousands of kilometres. Overall, our results reinforce the need to keep global warming levels below 1.5 °C and halt deforestation, as well as ecologically restore degraded forests to avoid high transition risks across the Amazon forest system.

Globally, native biomes have been unprecedentedly threatened by anthropogenic activities and are already showing signs of decreasing resilience[6-8]. Among those endangered biomes is the Amazon forest, where increasing droughts, the loss of biodiversity, degradation and deforestation[9-14] outpace natural variability[15]. Furthermore, the forest is transitioning from one of the largest terrestrial carbon sinks to a carbon source[16-18]. Importantly, direct and indirect stressors, such as deforestation and increases in extreme drought events, may be self-amplified by the forest system itself[19]. Thus, it is considered a tipping element of the Earth system whereby critical transitions may occur if local thresholds are crossed, which could trigger self-amplified changes as stabilizing feedbacks shift to destabilizing ones[20,21]. So far, Earth system models have focused primarily on forcing climate-induced state shifts and found critical global warming thresholds in the Amazon between 2 and 6 °C (refs. 20,22–26). On the other hand, some Earth system model studies[27,28] and an empirical study[29] identify regional forest shifts before these levels of global warming. In fact, the Amazon forest system could experience critical transitions at lower global warming levels than previously predicted through a range of adverse compounding drivers (for example, through heating, degradation, droughts and deforestation) occurring simultaneously rather than due to global warming alone[1,3,4,30].

An important reason for an increased risk of critical transitions is that both droughts and deforestation undermine the biome's self-stabilizing mechanism of atmospheric moisture recycling and could therefore lead to earlier critical transitions.

Part of the Amazon's precipitation is externally sourced and part is recycled, to which the trees contribute considerably[31,32]. The trees do this through atmospheric moisture recycling by absorbing water from soil layers and releasing it through their leaves through transpiration, as well as by interception evaporation, whereby precipitation is held by the canopy and does not reach the soil[33-37]. Through cycles of precipitation and evapotranspiration, the system maintains forest precipitation levels and, consequently, its own existence. In numbers, locally up to 50% of the forest's precipitation is forest-generated from within the basin[35]. Notably, emergent canopy trees are particularly critical, contributing around 71% of transpired water, thereby recycling approximately 26% of the precipitation back to the atmosphere[38]. Overall, trees recycle around 36% of precipitation through transpiration, substantially exceeding interception evaporation rates that amount to 22% of recycled moisture[38]. Moreover, the forest's transpiration during the late dry season critically determines the onset of the wet season, meaning that deforestation can delay the wet season and reduce

[1]Center for Critical Computational Studies, Goethe University Frankfurt, Frankfurt am Main, Germany. [2]Potsdam Institute for Climate Impact Research, Member of the Leibniz Association, Potsdam, Germany. [3]Senckenberg Research Institute and Natural History Museum, Member of the Leibniz Association, Frankfurt am Main, Germany. [4]Institute for Earth and Environment, University of Potsdam, Potsdam, Germany. [5]Stockholm Resilience Centre, Stockholm University, Stockholm, Sweden. [6]Instituto Juruá, Manaus, Brazil. [7]Equalsea-lab, University of Santiago de Compostela, Santiago de Compostela, Spain. [8]Graduate Program in Ecology, Federal University of Santa Catarina, Florianópolis, Brazil. [9]Group IpES, Department of Physics, Federal University of Santa Catarina, Florianópolis, Brazil. [10]Instituto Relva, Rio de Janeiro, Brazil. [11]Copernicus Institute of Sustainable Development, Utrecht University, Utrecht, The Netherlands. ✉e-mail: wunderling@c3s.uni-frankfurt.de; a.staal@uu.nl

dry season precipitation[39,40]. The interdependence of forest and precipitation implies that disturbances from increasing drought intensity and deforestation can spread and erode forest resilience remotely[41,42]. Deforestation of primary and secondary forest, which has accumulated to more than 15% of the biome, has already reduced moisture recycling in the Amazon, particularly in the south[43].

Globally, uncertainties in the future development of both climate change and land use changes are covered by the Shared Socioeconomic Pathways (SSPs)[44]. The SSPs comprise a set of consistent scenarios that depend on different assumptions about economic and political developments globally and are used to study a range of plausible futures until 2100. Recently, a global model for assessing changes in atmospheric moisture recycling in different SSPs was developed[5]. It is a new version of the Lagrangian moisture tracking model UTrack. UTrack constructs the spatial connections between evapotranspiration and precipitation by following the trajectories of air parcels, diagnosing moisture transport based on input fields. Instead of using reanalysis data as output fields, this new version builds on the second version of the Norwegian Earth System Model (NorESM2), creating, among many other variables, daily output on precipitation, evaporation, wind speed and further environmental variables. This enables the computation of scenario-dependent changes in atmospheric moisture transport throughout this century (Methods). Additional deforestation scenarios can be used to explore the isolated effects of deforestation on moisture recycling.

In the Amazon, reduced internal moisture transport due to anomalous droughts and deforestation may push forests closer to, or beyond, their physiological limits[45]. Severe droughts, including those in 2005, 2010, 2015–2016 or 2023–2024, already have a regular impact on the Amazon forest and are projected to become more frequent[1,46–52]. However, these impacts are unequal across the biome, as the trees have adapted to varying levels of water stress. Adaptations manifest as different drought-tolerance strategies, such as deeper roots, deciduousness, trunk capacitance and resistance to embolism, which help to shape plant communities more suited to drier conditions[53–57]. While such adaptations allow trees to function close to the precipitation limits of Amazonian forests, they are exposed to high water stress[58] during extreme events, such as the recent droughts induced by El Niño. If drier conditions become permanent or persist longer than the forest is adapted to, these strategies may no longer suffice[45,59], followed by a local-scale transition of the forest. Respective observational evidence suggests that local-scale transitions (tipping points) in the Amazon are plausible[4]: studies have shown that vegetation can become trapped by fire in an open state once forest is lost[60–62], and satellite analyses have suggested bistability between forest and savanna under similar climatic conditions[63,64]. Moreover, a shift from wet- to dry-affiliated species[59] and a widespread loss of forest resilience across the Amazon basin[8] have been documented. Precipitation exclusion experiments[65] demonstrate that sustained droughts can trigger about 35% biomass decline after about 12–15 years in response to a 50% precipitation reduction treatment, highlighting possible collapse in Amazonian trees, particularly in larger ones, after prolonged drought stress[66,67]. While these experiments primarily reflect local-scale responses and do not capture broader system-wide self-reinforcing feedbacks, they provide critical insights into physiological thresholds relevant to local-scale tipping processes. How these local transitions propagate through the Amazon can be studied using a dynamical modelling approach relying on atmospheric moisture connections, causally linking local forest transitions in an Amazon-wide network. Examples of these atmospheric moisture recycling networks are shown in Extended Data Fig. 1.

So far, determining the risks for systemic configuration changes in the Amazon forest region through multiple pressures acting at the same time has mostly relied on expert assessments without extensive quantification[1,3]. However, building on the new conceptual, empirical

and methodological advances[1,5,45], we can now start to quantify limits to the Amazon forest adaptive capacity, given (1) climate change, (2) additional deforestation and (3) the forest's self-dependence through moisture recycling, in different future scenarios. We model the entire Amazon forest as a locally bistable system between a forest and an alternative state without trees using an established dynamical systems approach[45,68] covering the Amazon river basin at a resolution of about 1° × 1° grid cells (exact resolution of NorESM2: 1.25° × 0.9375°; Extended Data Fig. 2). We model the stability of local forest grid cells based on the mean annual precipitation (MAP), dry season intensity and duration (calculated using the maximum cumulative water deficit, MCWD) and the corresponding moisture transport network (Extended Data Figs. 1–4) in four SSPs: SSP1-2.6, SSP2-4.5, SSP3-7.0 and SSP5-8.5. We model the adaptation of the Amazon forest system to its local past environmental conditions through consistent data from historical model runs (Extended Data Figs. 3 and 4). Note that, while we do not explicitly simulate fire dynamics, our assumption of local-scale transitions implicitly accounts for fires, although additional drought–fire interactions may further exacerbate forest loss. This makes our approach conservative in this regard[69]. Lastly, we add deforestation to our experiments by using a severe deforestation scenario that enables us to assess the deforestation–climate change interaction in detail, including modest to severe deforestation[70] (Supplementary Fig. 1), also because the SSP scenarios do not include severe deforestation pathways.

## Transitions due to global warming

We quantify spatially resolved transition risk across the Amazon throughout the twenty-first century on a ~1° × 1° grid, defined as the fraction of ensemble members that end up in the transitioned regime, for the scenarios SSP2-4.5 (the scenario closest to current emission pathways), SSP3-7.0 and SSP5-8.5 using their respective MAP, MCWD and moisture transport network. In these experiments, following earlier literature[5], we use 10-year averages to cancel out the effect of single years and keep these conditions constant to evaluate the long-term committed damage if these conditions persist. We find that, without deforestation, the transition risk for a SSP2-4.5 scenario at the end of the century is very low (average from 2090 to 2099, representing global warming of 2.8 °C; Fig. 1a). By contrast, for the SSP3-7.0 and SSP5-8.5 scenarios at the end of the twenty-first century (average from 2090 to 2099, representing global warming of 4.0 °C and 4.9 °C, respectively), we reveal a strongly increasing transitioned area if global warming levels of 3.7–4.0 °C are reached (time series in Fig. 1b,c). We find the largest transition risks in the western and southwestern parts of the Amazon (map in Fig. 1b,c), which are the largest receivers of forest-generated precipitation[35]. For even higher levels of global warming (4.9 °C), these transition risks are robustly shown to be further exacerbated (see Fig. 1c and Supplementary Fig. 2 for a comparison). However, depending on the exact environmental conditions in MAP and MCWD, transition risk peaks can already become prevalent at lower levels of global warming (for example, the peak around 2.1 °C in SSP2-4.5; Fig. 1a). Thus, there is a non-trivial relationship between global warming and environmental changes regarding MAP, MCWD and moisture transport that subsequently determines the transition risks in the Amazon forest (Extended Data Fig. 3). We therefore ran a robustness analyses that investigated transition risks due to MAP only and MCWD only, separating out individual contributions to the overall transition risk (Extended Data Figs. 5 and 6 and Supplementary Figs. 3 and 4). We find that southern and southeastern parts are at risk due to decreasing MAP levels (Extended Data Fig. 5a) while more central parts of the forest are at risk due to MCWD (Extended Data Fig. 5b). Moreover, we quantify transition risks for the 2090s in SSP1-2.6. Consistently, we find very few transition events for a SSP1-2.6 emission scenario

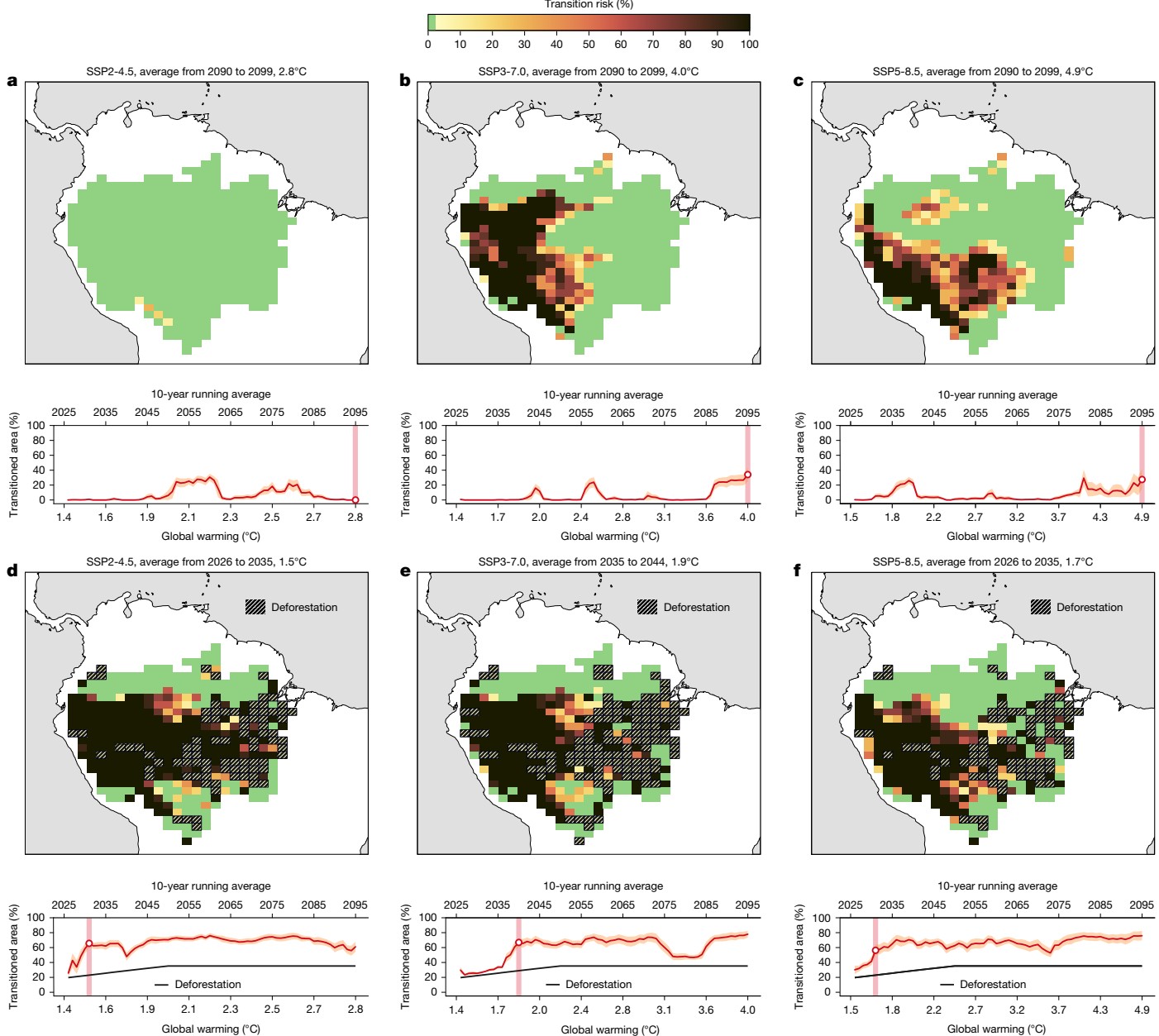

**Fig. 1 | Transition responses in the Amazon forest system with respect to global warming and deforestation. a**, Transition response to global warming only following an SSP2-4.5 emission scenario throughout the twenty-first century (time series) with average transition risks as a solid red line. The vertical pink shaded area with the hollow circle denotes the point at which the transition response is mapped out across the Amazon forest system (here, for a warming level of 2.8 °C, average from 2090 to 2099; see the map). **b,c**, The same but for SSP3-7.0 (average from 2090 to 2099, 4.0 °C; **b**) and SSP5-8.5 (average from 2090 to 2099, 4.9 °C; **c**). Individual contributions to the transition risks from MAP and MCWD are shown in Extended Data Fig. 5. **d–f**, The transition risks for SSP2-4.5 (average from 2026 to 2035, 1.5 °C; **d**), SSP3-7.0 (average from 2035 to 2044, 1.9 °C; **e**) and SSP5-8.5 (average from 2026 to 2035, 1.7 °C; **f**) including deforestation (hatched) pathways in addition to climate change. For all time series, the shading denotes the standard deviation of the transitioned area.

(Supplementary Fig. 2). Further robustness checks can be found in Supplementary Fig. 8 (for the mixture between adaptive thresholds and fixed critical thresholds following ref. 1), Supplementary Fig. 9 for constant evapotranspiration of 100 mm per month and Extended Data Fig. 9 for two further scenarios on adaptation capacities (see the 'Robustness checks' section of the Methods for further details). In each of our sensitivity analyses, we identify a robust transition occurring at around 3.7 °C of global warming or higher (Fig. 1a–c, Extended Data Figs. 6 and 7 and Supplementary Figs. 3, 4, 8 and 9). Intermediate peaks of transitions before this threshold in both scenarios correspond to particularly dry episodes, which do not reflect the overall status of the time series (Extended Data Fig. 3).

## Transitions including deforestation

As deforestation may amplify global warming-induced water stress downwind from the deforestation itself[1,3,12,41,71,72], we explore the additional effects of severe deforestation[70] (Fig. 1d–f). This scenario is characterized by strong deforestation throughout 2002–2050, commencing from the far south and east (currently the deforestation arc) into central Amazon parts further west. It assumes almost no efforts to reduce deforestation throughout the first half of the twenty-first century (Supplementary Fig. 1) and projects plausible Amazonian deforestation pathways until mid-century, explicitly linked to major infrastructure projects. Although the data may be dated, they enable

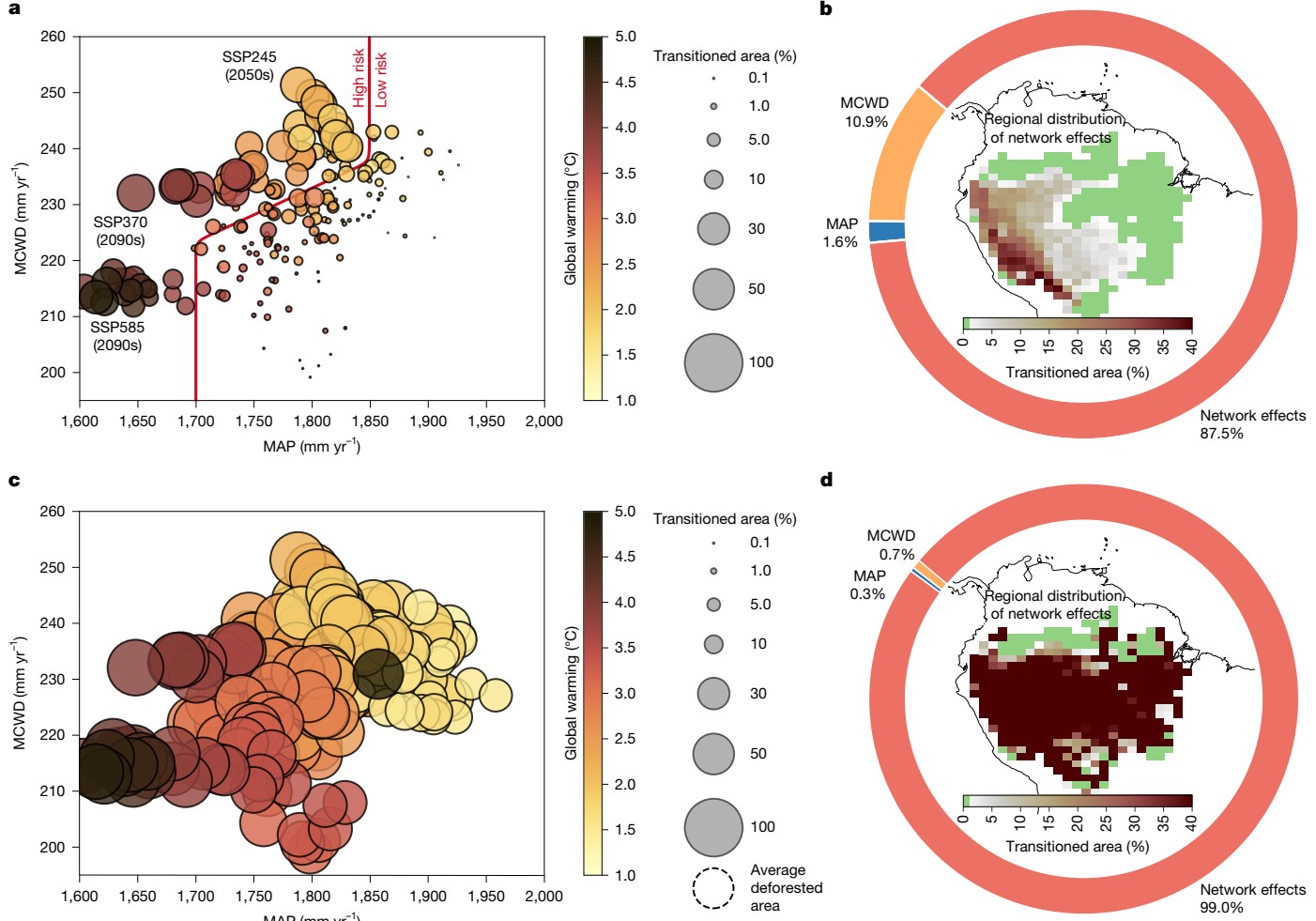

**Fig. 2 | Reasons for crossing a systemic transition in the Amazon forest system. a**, Transitioned area (size of the circle; as a percentage of the entire Amazon basin) across all investigated global warming scenarios dependent on MAP and MCWD without deforestation. The thick red line separates low transition risks from high transition risks in the two-dimensional plane of MAP and MCWD. The colour of the circle depicts the global warming level. **b**, Transition reason across all investigated emission scenarios without deforestation.

The pie chart shows the three transition reasons: MAP, MCWD and network effects (cascading transitions). The inset shows the locations where cascading transitions are the dominant reason for transitioning. **c,d**, The same analysis of transitioned area (**c**) and transition reason (**d**) as in **a** and **b**, but including deforestation. The dashed circle in **c** denotes the average deforested area across all plotted global warming scenarios.

us to systematically study many different deforestation levels between 18% in 2020 and nearly 35% in 2050. After 2050, we hold the cumulative deforestation constant until 2100. Following this deforestation scenario, we find widespread transitions for all SSPs at the end of the twenty-first century (Supplementary Fig. 2b), reaching more than 62% of the entire Amazon forest basin for SSP1-2.6 and SSP2-4.5 and 77% for SSP3-7.0 and SSP5-8.5. Importantly, focusing on the SSP2-4.5, SSP3-7.0 and SSP5-8.5 trajectories across the twenty-first century reveals that a systemic transition is crossed between 1.5 and 1.9 °C when combined with a basin-wide deforestation of 22–28%. Such conditions could be transgressed until mid-century (time series in Fig. 1d–f). For higher ratios of deforestation and stronger global warming levels towards the end of the twenty-first century, transition events occur across all considered emission scenarios. Note that transition risks can decrease if environmental conditions in MAP and MCWD become wetter in the respective scenario (Extended Data Fig. 3) (for example, beyond 2.7 °C in SSP2-4.5, or between 3.1 and 3.6 °C in SSP3-7.0; Fig. 1d,e). However, if deforestation is kept strictly at or close to today's levels (15%), a large-scale transition is absent (Supplementary Figs. 5 and 6). Note that all SSPs possess highly optimistic land use change assumptions with deforestation remaining close to the historical level (Supplementary

Fig. 7). Crucially, they assume that almost no deforestation occurs in the east of the Amazon basin, so downwind cascading transitions along the main wind direction are avoided. It is therefore particularly important not only how much further deforestation takes place, but also in which locations.

## Reasons for transitions

Following global warming pathways determined by the SSP-based emissions will lead to significant drying over the Amazon forest[49]. We therefore quantify the MAP and MCWD conditions in which changes in transition risks are large in a scenario in which we do not consider deforestation (Fig. 2a; high-risk zones are defined by scenarios with a transitioned area of at least 10% across the Amazon basin). In the MCWD–MAP plane, these levels are characterized by (1) MCWD reaching values of 225 mm yr$^{-1}$ and higher, or (2) Amazon basin-wide MAP levels below 1,850 mm yr$^{-1}$. Note that global warming levels of 3.7–4.0 °C or above consistently lead to high transition risks (Fig. 2a).

Next, we differentiate between three causes of transitions: MAP levels falling below a critical threshold, MCWD crossing a critical level and cascading transitions caused by transitions upwind. Without

deforestation, we find that cascading transitions strongly dominate the reasons for critical transitions in the Amazon forest (87.5%) followed by MCWD directly (10.9%) and MAP directly (1.6%) (Fig. 2b). This means that, despite the overall strength of the moisture recycling network declines for stronger climate change scenarios (Extended Data Fig. 8), these cascading network effects (cascading transitions) are the dominant reason for transitions under climate change. Once increasing MCWD and declining MAP levels cause initial transition events, their knock-on effects induce large-scale cascading effects across a range of hundreds to thousands of kilometres (Fig. 2b, inset map). These cascading transitions are mostly located downwind of the initial transition events triggered by MAP and MCWD (see their regional distribution in Supplementary Fig. 10). Results including deforestation show a high transitioned area consistently across the simulations (Fig. 2c), and transition reasons are dominated by cascading transitions (network effects, 99.0%; Fig. 2d).

## Discussion

We estimate that a systemic transition for major parts of the Amazon forest lies between 3.7 and 4.0 °C of global warming without considering deforestation. This represents water stress conditions that may be strong enough to induce a widespread transition event accounting for up to 35% of the Amazon forest basin. Thus, following the SSP3-7.0 (let alone SSP5-8.5) emission pathway is a highly unsafe scenario for the forest even in the absence of deforestation or land use change. However, we do not find a point where the complete Amazon system transitions, indicating that there are regions in the Amazon forest that maintain high resilience to global warming-induced precipitation reductions and increased dry season intensity. Consistent with earlier research[42], we find high transition risks for western and southwestern parts of the forest due to direct or indirect water stress impacts as a result of reduced atmospheric moisture transport (Fig. 2a and Extended Data Fig. 8). We separate highly risky environmental conditions from less risky conditions in response to drought stress as quantified by critical MCWD and MAP conditions. Specifically, we find large changes in transition risks as we cross certain levels in the MAP–MCWD plane. Those are average MAP levels below 1,850 mm yr$^{-1}$ and MCWD values above 225 mm yr$^{-1}$ (Fig. 2a). After including deforestation in our analysis, we find widespread transitions across the Amazon basin at deforestation levels of 22–28% combined with global warming levels of 1.5–1.9 °C (Fig. 1d–f). Our simulations suggest that severe deforestation scenarios could result in detrimental transition events across large parts (up to 77%) of the Amazon forest even at moderate emission scenarios also because projected deforestation is mostly located in the east of the Amazon basin. This can contribute strongly to downwind cascading transitions with particularly deteriorating impacts (Fig. 1d–f). Overall, our simulations quantitatively confirm independent modelling efforts and expert judgements that assess critical deforestation levels between 20% and 40% (refs. 71,73) with simultaneous global warming levels of 1.5–2.0 °C (refs. 3,12). We conclude that our quantification of a widespread transition risk of the Amazon forest is robust, in terms of commitment to irreversible change. However, the impact time for a full transition to a new degraded forest state may take decades to a few centuries to realize once a threshold has been crossed[20]. As our experiments impose only increasing global warming pathways and SSP2-4.5, SSP3-7.0 and SSP5-8.5 are not overshoot scenarios, we only quantify committed transitions in equilibrium, and do not resolve the pace of forest change.

Our modelling approach, while intentionally simplified to analyse transition dynamics, includes important uncertainties such as representations of vegetation dynamics (for example, a non-process-based inclusion of drought–fire feedbacks), adaptation capacities and the reliance on NorESM2 for hydrological inputs (MAP, MCWD and the atmospheric moisture transport network). Furthermore, adaptive forest capacities are covered by an aggregate measure—encapsulating processes such as rooting depth and water access, stomatal regulation and hydraulic safety, species composition and acclimatization to historical climate variability—thereby providing a compact way to map diverse ecological mechanisms onto a common dynamical systems parameter (see the 'Adaptation' section of the Methods). Together, these represent important simplifications of our approach that should be kept in mind when interpreting the results.

However, by systematically exploring sensitivities of forest resilience to global warming and deforestation, we offer a baseline that can guide and be refined by future studies using more complex vegetation models as well as observational data. To cover uncertainties in forest adaptations to climatic conditions and carefully propagate their uncertainties, we constructed a large-scale Monte Carlo ensemble for all our experiments and carried out extensive robustness analyses (see the 'Ensemble construction' section of the Methods). Our results turn out robust across a number of sensitivity experiments, (1) regarding transition risks in response to single transition drivers (MAP or MCWD only) but also (2) using a combination of adaptive thresholds and fixed thresholds (Extended Data Fig. 6 and Supplementary Figs. 3, 4 and 8), (3) using potential rather than actual evapotranspiration, and (4) testing against more moderate effects of forest transitions on evapotranspiration. The results are qualitatively equal and match quantitative expectations (Extended Data Figs. 6e,f and 7 and Supplementary Fig. 9; see the 'Robustness checks' section of the Methods). This sensitivity experiment provides a robustness check against the uncertainty of how strong precipitation reductions over the Amazon basin are when parts of the forest are deforested. We perform this sensitivity experiment because published estimates of precipitation sensitivity to Amazon deforestation span a wide range—from relatively modest basin mean reductions to substantially stronger regional impacts[74,75]. This reflects differences in observational measurements, modelling efforts and the extent to which forest–atmosphere feedbacks are resolved. Lastly, (5) we test two further scenarios on adaptation capacities and find robust results showing that qualitative changes in the stability landscape of the Amazon forest emerge beyond 3.5 °C irrespective of the exact adaptation capacity (Extended Data Fig. 9). An additional uncertainty is the effect of $CO_2$ fertilization on precipitation and drought intensities over the Amazon forest system. Some studies suggest that their effect may be on the same order of magnitude as deforestation[76,77]. It also needs to be acknowledged that $CO_2$ physiological effects on precipitation may implicitly be included in the forcing because our model is run offline and forced by climate variables derived from NorESM2. Consequently, $CO_2$-driven reductions in evapotranspiration and moisture recycling may contribute to changes in MAP and MCWD.

Earlier complex Earth system model analyses have resulted in different (or no) regions being affected by transitions at global warming levels mostly between 2 and 6 °C (refs. 22–27,78). Some current Earth system models have been shown to exhibit forest decline under deforestation and global warming even without the explicit inclusion of forest-precipitation feedbacks[30]. In this study, we explicitly included forest–precipitation feedbacks and revealed a large-scale Amazon transition indirectly from the output of an Earth system model (NorESM2) that was not present in the original model. Our approach paves the way for analysing output from other Earth system models in similar ways to identify large-scale biosphere transitions, for example, in the upcoming CMIP7 runs.

Importantly, our results show that Amazon transitions are not inevitable. With a best estimate of a large-scale transition between 3.7 and 4.0 °C of global warming compared to preindustrial levels, humanity has the abilities to limit global warming before the most dangerous levels are reached. However, the role of deforestation is crucial. With extensive Amazonian deforestation, current global warming would already be at a dangerous level. Thus, limiting deforestation is key to preventing a systemic Amazon transition, and the promises by the

Brazilian and further South American governments of entirely ending Amazon deforestation by the end of this decade are the right step towards this necessity. However, it is unclear whether and to which extent they will be fulfilled. Furthermore, transitions in the Amazon would disrupt atmospheric moisture transport to downwind regions outside the Amazon basin[79] that are under intensive agricultural use such as southern Brazil, Bolivia, Paraguay and down to the Río de la Plata basin in Argentina—potentially threatening crop yields and regional water security.

Yet our results also point at the potential of strengthening Amazon forest resilience by forest restoration. Thus, restoration may recover the negative effects of deforestation on moisture recycling more easily and faster than alleviating the loss of biodiversity or adaptive capacities. For example, primary forest possesses an extraordinarily high biodiversity and provides unique ecological functions that neither a regrown nor a managed forest reaches in terms of biodiversity[80]. Nevertheless, strategically restoring lost and degraded forests may help build resilience to global warming that is critically needed to prevent large-scale transitions[81]. One example of such important, while ambitious, plans are the ones by the Brazilian government to restore around 12 million hectares of forest in the Arc of Restoration[82,83]. Such efforts should be considered of global importance, as transitions in the Amazon will have impacts not only among riparian communities, but also on socioeconomic development across the world. In summary, our findings advance the attempt to pinpoint the safe boundaries of the Amazonian forest system, highlighting the need for global cooperation to halt emissions, reduce deforestation to zero and also restore forest cover across the region.

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

# Methods

## Moisture tracking in SSPs

We used UTrack, a Lagrangian atmospheric moisture tracking model, to track moisture forwards in time from evaporation to precipitation[5,84]. Being a three-dimensional Lagrangian tracking model that reconstructs moisture trajectories using evaporation and precipitation directly, UTrack is conceptually similar to some other Lagrangian methods[85,86], but differs from other widely used tracking methods that are Eulerian[87] or follow changes in specific humidity instead[88]. Using UTrack, we tracked the three-dimensional atmospheric trajectories of large numbers of individual 'parcels' of moisture and updated their positions every time step of 4 h based on evaporation, precipitation, humidity levels and three-dimensional wind speeds and directions. The respective forcing data were output of the medium-resolution Norwegian Earth System Model version 2 (NorESM2)[89], which provides sufficiently detailed model output for UTrack and comprises all tier 1 scenarios in ScenarioMIP[90] up until 2100: SSP1-2.6, SSP2-4.5, SSP3-7.0 and SSP5-8.5. Furthermore, it outperforms most CMIP6 models on reproducing historical observations of the hydrological cycle[91,92]. NorESM2 has a temporal resolution of 1 day and a spatial resolution of 1.25° × 0.9375°. We performed forward tracking from each of the 416 grid cells in the Amazon basin for each month and SSP. For each mm of evaporation at each grid cell during each time step of 4 h, we released 100 moisture parcels at random locations above the starting grid cell. Consistent with the ERA5-based UTrack model[84], this time step is considerably smaller than the temporal resolution of the forcing data, to prevent skipping of grid cells by parcels during a time step. The wind speeds are calculated for eight pressure levels: 1,000 hPa, 850 hPa, 700 hPa, 500 hPa, 250 hPa, 100 hPa, 50 hPa and 10 hPa. To compensate for underestimated vertical mixing of moisture in the forcing data, each parcel is additionally assigned an occasional quasi-random repositioning along the atmospheric column. This is set such that on average once every 24 h, a parcel repositions itself vertically, where the probability of the new position is weighted by the specific humidity along the column[5,84]. The moisture content of the parcels is updated if precipitation occurs at that time step in the grid cell corresponding to the position of the parcel and the precipitation moisture is allocated to that grid cell. The tracking and updating continue until 99% of the original moisture in the parcel has been allocated to precipitation or after 30 days have passed since parcel release. It is important to note that, as opposed to ref. 5, we tracked evapotranspiration from each grid cell of the Amazon separately, and stored the results per grid cell, per month and per SSP. As we released 100 parcels per mm of evapotranspiration for each grid cell, this results overall in more than 1 billion parcel releases for this study. As such, although previous studies analysed moisture recycling for the Amazon in CMIP5 (ref. 93) and CMIP6 (refs. 5,94) models, we present grid cell-to-grid cell simulations, enabling us to construct the full moisture flow network. Finally, we validated the NorESM2 wind speed data for the Amazon and the Amazon precipitation recycling ratios using ERA5 reanalysis data and ERA5-forced UTrack runs. We show good correspondence between them and find no systematic bias that can explain our main transition risk results. We present these results in the Supplementary Information (Supplementary Note (Validation of moisture recycling based on EAR5 reanalysis data) and Supplementary Figs. 13–18).

## Environmental data

We used MAP and evaporation values (to construct MCWD) from NorESM2 for the adaptation period from 1950 to 2014 (see the 'Adaptation' section) as well as for the four SSP scenarios that we used. For the three scenarios SSP2-4.5, SSP3-7.0 and SSP5-8.5, we evaluate the entire century using the now available moisture tracking data (2021–2099) while, for SSP1-2.6, we evaluate the decade 2090–2099 only. MAP and MCWD are computed as 10-year averages to cancel out

the effects of single years that are particularly dry or wet. We use 10-year averages to capture long-term climatic shifts that drive system-wide vegetation changes, as supported by rainfall exclusion experiments showing that Amazon forests typically respond to sustained drought conditions over timescales of ten years[1,47,65,95]. While individual drought years can be impactful, especially for large trees, long-term stress is more relevant for assessing transition dynamics at the basin scale. Moreover, instead of calendar years, we account for dry and wet season conditions by using hydrological years. Hydrological years start in October of one year and run until September of the following year. MAP is computed from adding the corresponding monthly precipitation data in the respective hydrological year. For MCWD, we follow ref. 24 and compute the cumulative water deficit (CWD) from the according monthly precipitation and evaporation values using hydrological years:

$$\text{MCWD} = \text{abs}[\min(\text{CWD}_i, \text{CWD}_{i+1}, ..., \text{CWD}_{i+11})],$$
$$\text{with } \text{CWD}_{i-1} + \text{Precipitation}_i - \text{Evaporation}_i \quad (1)$$
$$\text{and } \max(\text{CWD}_i) = 0$$

Note that we use absolute values of MCWD in this study. While we use monthly precipitation values and evaporation values directly from NorESM2, the resulting global warming levels (from the SSP scenarios) are based on the wider spread of the CMIP6 database to not rely on a single Earth system model and its specific equilibrium climate sensitivity. Specifically, we use the median global temperature change as simulated in MAGICC7 (based on Fig. 4.40a of ref. 96).

## Deforestation data

The deforestation data sets are taken from two different data sources. First, we use a severe deforestation data set that originates from ref. 70 and covers the Amazon basin from 2002 to 2050. This scenario assumes that the deforestation trends across the basin continue as well as additional deforestation occurring at locations of (planned) road pavements. At the same time, existing and proposed protected areas are ignored as reasons to limit or stop deforestation at these locations[97]. The projected deforestation rates were constructed by using historical images and their variations from 1997 to 2002 and then added to the effect of paving a set of major roads. We converted and regridded these data to the same grid as the environmental data, and kept deforestation levels from 2050 constant until the end of the century. From 2020 to 2050, the deforestation increased from ~0.55 million km$^2$ to ~0.9 million km$^2$ in this scenario (that is, from 18% to 35% of the Amazon basin being cleared), leading to an average yearly deforestation of ~18,000 km$^2$ in this period. Despite the fact that the deforestation data are already a bit old, they uniquely project plausible Amazonian deforestation pathways until mid-century, explicitly linked to major infrastructure projects. Thus, the data enable a systematic assessment of critical deforestation thresholds relevant for analysing potential transitions. Second, we also include the deforestation scenarios following the respective SSP-based land use change scenarios (Supplementary Fig. 7). These are conservative scenarios with very limited deforestation after 2020 and none of these scenarios crosses the 25% level of basin-wide deforestation. Furthermore, most deforestation takes place in the west of the Amazon basin rather than in the east where vulnerabilities would be transported downwind.

## Adaptation

Forests are not uniformly adapted to local climate conditions[54,55,98]. Various strategies exist both within and among forests to manage dry seasons and extreme droughts. We assume that local climate conditions have probably driven specific forest trait adaptations through processes such as environmental filtering, competitive exclusion and resilience. Specifically, we assume that forest ecosystems spread

throughout the Amazon forest system are adapted to local adaptation values (here on a -1° × 1° basis). This means that each grid cell is adapted to its past local environmental conditions. Our adaptation period ranges from 1950 to 2014 and includes the consistent historical simulation run, at which the four different SSP scenarios are branched off. Thus, the adaptation period is 1950–2014, while we evaluate the transition risk in the experimental period that ranges from 2021 to 2099 (averaged over a 10-year running average). With adaptation to past local conditions, we mean that the forest cells are adapted to their past MAP and MCWD values in the adaptation period, that is, to local precipitation and drought intensity values. Thus, they represent a local-scale tipping element with a threshold at MAP or MCWD values representing drier conditions than, on average, 1 s.d. away from those in the adaptation period. Locally, this means that critical thresholds can be vastly different; for example, drier regions in the Amazon forest are also capable of surviving drier conditions in the future. Overall, 1 s.d. is a conservative choice as losing 1 s.d. of moisture means on average losing around 25% of its MAP (Extended Data Fig. 4 and Supplementary Fig. 11), or becoming approximately 33% drier (that is, dry season intensity increase; Extended Data Fig. 4 and Supplementary Fig. 11). With this procedure, we are following and extending ref. 45, and follow the hypothesis that safety margins of forest ecosystems to droughts are similar regardless of the present (local) MAP[58]. However, we also find that our results are robust to the assumption that drier regions have lower safety margins than wetter regions as well as the other way round (see the 'Robustness checks' section for details). Lastly, although much of the forest may be adapted to local drought conditions, absolute thresholds are likely to exist for critical transitions in the Amazon forest system[1,63] beyond which trees cannot survive. Thus, we ran a robustness analysis taking into account local adaptation as well as discrete thresholds in MAP and MCWD, in which the hard-wired thresholds follow the recent review on critical transitions in the Amazon forest[1] with robust results in a qualitative and quantitative sense (see the 'Robustness checks' section for details).

## Ensemble construction

As ecological adaptation varies stochastically across the Amazon forest system, they are drawn from a uniform distribution between 0.75 and 1.25 s.d. based on the values for MAP and MCWD of the local forest grid cell. These locally different adaptation values account for random variations in, for example, stomatal closure or heightened respiration. Ultimately, for each SSP scenario in each analysed decade, we draw ten different samples that are randomly drawn values for each cell from $\sigma_i \in [0.75; 1.25]$ for $i = 1, 2, ..., N_{\text{grid cells}}$ with 416 grid cells. As mentioned earlier, the precise value of adaptation is uncertain and may vary across different regions, influenced by several factors that are not explicitly modelled in this study such as the soil quality or competition of different species. To cover these uncertainties, we create a large ensemble and compute transition risks across the Amazon forest. Our ensemble size amounts to more than 1.25 million simulations (>3,000 simulations per grid cell), including all of our robustness checks.

## Interacting dynamical systems approach

We extend the methodology developed in earlier literature[45,68], where individual grid cells are modelled as interacting differential equations as follows:

$$\frac{\mathrm{d}x_i}{\mathrm{d}t} = -x_i^3 + x_i + C_{\text{crit},i}(\text{MAP}_i, \text{MCWD}_i) \\ + \sum_{k=1, k \neq i}^{N_{\text{gridcells}}} R_{ki}\left(\Delta\text{MAP}_{ki}, \Delta\text{MCWD}_{ki}\right)\frac{x_k}{2}. \tag{2}$$

Here, each grid cell is modelled as a nonlinear dynamical system with two alternative stable equilibria, representing a forest state and

an alternative state (Extended Data Fig. 2). A transition occurs when hydroclimatic forcing (changes in MAP or MCWD) or the loss of stabilizing moisture inputs causes the forest equilibrium to lose stability, after which internal feedbacks drive the system towards the alternative state. This nonlinear equation is a typical dynamical system equation that can exhibit tipping point behaviour, where $C_{\text{crit}}$ and the summation terms can be interpreted as a time-evolving bifurcation parameter. Their dynamics follow the normal form of a fold bifurcation, a standard representation of threshold-driven regime shifts in ecological and climate systems[68,99–101]. While such dynamics are consistent with hysteresis and limited reversibility, reversed forcing is not simulated in our experiment as SSP2-4.5, SSP3-7.0 and SSP5-8.5 are not overshoot scenarios. In equation (2), $x_i$ represents the state of the forest at grid cell $i$, where $x_i = -1$ is forest and $x_i = +1$ is the alternative state, which is an (open-canopy) degraded ecosystem state (for example, a savanna or dry degraded forest state). The tipping point (transition threshold) with respect to two critical parameters MAP and MCWD is located at

$$C_{\text{crit},i}(\text{MAP}_i, \text{MCWD}_i) = \max(C(\text{MAP}_i), C(\text{MCWD}_i)) \\ + \left(1 - \frac{\max(C(\text{MAP}_i), C(\text{MCWD}_i))}{\sqrt{\frac{4}{27}}}\right) \\ \times \min(C(\text{MAP}_i), C(\text{MCWD}_i)) \tag{3}$$

with the components

$$C(\text{MAP}_i) = \sqrt{\frac{4}{27}} \times \left(\frac{\text{MAP}_i - \mu_{\text{MAP},i}}{\text{MAP}_{\text{crit},i} - \mu_{\text{MAP},i}}\right)^{-1} \\ C(\text{MCWD}_i) = \sqrt{\frac{4}{27}} \times \frac{\text{MCWD}_i - \mu_{\text{MCWD},i}}{\text{MCWD}_{\text{crit},i} - \mu_{\text{MCWD},i}} \tag{4}$$

$\mu_{\text{MAP},i}$ is the grid cell-specific long-term average from the adaptation period (1950–2014) and $\text{MAP}_{\text{crit},i}$ is the tipping point with $\text{MAP}_{\text{crit},i} = \mu_{\text{MAP},i} - \sigma_i \times \Delta_{\text{MAP},i}$, where $\Delta_{\text{MAP},i}$ is the local adaptive capacity of the grid cell to its past environmental conditions, which is measured as the s.d. from 1950 to 2014. This means that a region that experienced larger environmental fluctuations in the past is also adapted (that is, resilient) to such environmental fluctuations in the future. This is the mechanism with which we implement local adaptive capacities of the biosphere dependent on local past environmental conditions. $\sigma_i \in [0.75; 1.25]$ is the uncertainty in the adaptive capacity, on which we construct our ensemble (see the 'Ensemble construction' section). While drier conditions are represented by larger MCWD values (see equation (1)), they are also represented by lower values of MAP. Thus, the exponent −1 is needed in equation (4). Lastly, the specific critical value of $\sqrt{\frac{4}{27}}$ is derived from the normal form of equation (2), and more details can be found in literature[99,102].

Moreover, the moisture recycling network is parameterized in the last term of equation (2), where $R_{ki} = R_{ki}(\Delta\text{MAP}_{ki}, \Delta\text{MCWD}_{ki})$ is the moisture transport link from cell $k$ to cell $i$:

$$R_{ki}(\Delta\text{MAP}_{ki}, \Delta\text{MCWD}_{ki}) = R_{ki,\text{MAP}} + \left(1 - \frac{R_{ki,\text{MAP}}}{\sqrt{\frac{4}{27}}}\right) \times R_{ki,\text{MCWD}}, \\ \text{for } C(\text{MAP}_i) > C(\text{MCWD}_i) \tag{5}$$

or

$$R_{ki}(\Delta\text{MAP}_{ki}, \Delta\text{MCWD}_{ki}) = R_{ki,\text{MCWD}} + \left(1 - \frac{R_{ki,\text{MCWD}}}{\sqrt{\frac{4}{27}}}\right) \times R_{ki,\text{MAP}}, \\ \text{for } C(\text{MCWD}_i) > C(\text{MAP}_i) \tag{6}$$

with the following compartments:

$$R_{ki,\text{MAP}} = R_{ki}(\Delta\text{MAP}_{ki}) = \sqrt{\frac{4}{27}} \times \left(\frac{\Delta\text{MAP}_{ki}}{\text{MAP}_{\text{crit},i} - \mu_{\text{MAP},i}}\right)^{-1},$$

$$R_{ki,\text{MCWD}} = R_{ki}(\Delta\text{MCWD}_{ki}) = \sqrt{\frac{4}{27}} \times \frac{\Delta\text{MCWD}_{ki}}{\text{MCWD}_{\text{crit},i} - \mu_{\text{MCWD},i}}. \qquad (7)$$

Here $\Delta\text{MAP}_{ki}$ represents the difference of the MAP arising from the atmospheric moisture recycling link from cell $k$ to cell $i$. Note that we remove the evapotranspiration of a transitioned (tipped) grid cell. However, this assumption is in good agreement with the additional robustness checks in which we assume that the remaining evapotranspiration values equal those of secondary vegetation after deforestation or transitioning (tipping). The remaining evapotranspiration values of secondary vegetation are taken from literature[103] (see the 'Robustness checks' section).

Ultimately, in equation (2), all moisture transports to grid cell $i$ are summed up (over $k$) so that each interaction has a stabilizing effect on the local tipping element $i$. If individual grid cells transition, they lose their stabilizing effect on subsequent cells and their individual moisture transport is subtracted in equation (2). If a cell is sufficiently close to its tipping point and loses enough stabilizing interactions, a tipping event and subsequent cascading transitions through the loss of moisture transport can occur with respect to either MAP being too low or MCWD too high (or both). While this is a very simplified approach to modelling interacting tipping elements and cascading transitions, it can flexibly be used to take local adaptations into account (and absolute thresholds; see the 'Robustness checks' section), which makes this approach very fruitful. For more details on the specific modelling approach, also see ref. 45. In the future, not only the thresholds of equation (2) could better reflect Earth system knowledge, but also the functional form ($dx/dt - x^3 + x + ...$) could be adjusted from more complex dynamic global vegetation models or observational evidence directly[1,65,104]. This could be a very promising way forward replicating complex models into simplified dynamics as started in this work.

**Robustness checks**

Overall, we run five extensive robustness checks (the results are summarized in Extended Data Figs. 6 and 7 and Supplementary Figs. 3–5, 8 and 9).

First, we investigate the effects when only one of the two critical variables, either MAP or MCWD, determines the occurrence of critical transitions in the Amazon forest. We find that this decomposition breaks down the overall transition risk (Fig. 1b) consistently into its two components, namely the one from too-low MAP (Extended Data Fig. 5a) and another from too-high drought intensities (MCWD; Extended Data Fig. 5b). However, summed up, the overall results are robust to our simulations with both critical variables (Extended Data Fig. 6a,b and Supplementary Figs. 3 and 4).

The second robustness check is taking into account local adaptations as well as discrete thresholds in MAP and MCWD. We presume the same local adaptations but add critical thresholds for local MAP values above 1,850 mm yr$^{-1}$ and MCWD values below 350 mm yr$^{-1}$, where forest cells are forbidden to tip, following safe boundaries for MAP and MCWD in a recent review on the Amazon forest[1]. This procedure inevitably increases the resilience of the forest in a hard-wired sense because some regions in the Amazon forest are forbidden to tip and will not act as initiators for subsequent cascading transitions. We therefore expect higher resilience at the same $\sigma_i$ values between 0.75 and 1.25 s.d. However, at lower values of $\sigma_i \in [0.50; 1.0]$ we obtain qualitatively and quantitatively robust results as compared to the main manuscript's simulations (Extended Data Fig. 6c,d and Supplementary Fig. 8).

The third robustness check concerns the limitation that the evapotranspiration values we use are limited by the water availability. As such, the actual MCWD value is underestimated as the potential

evapotranspiration is likely to be higher than the one that is measured and limited by water availability. We therefore ran an additional conservative robustness check using a constant evapotranspiration of 100 mm per month, resulting in very good agreement to our results in the main manuscript (Extended Data Fig. 6e,f and Supplementary Fig. 9). Moreover, we show that the high climate risk zone for transitions is located at MCWD values of more than 300 mm per month (Supplementary Fig. 12). This robustness check is conservative as potential evapotranspiration is likely to be considerably higher.

The fourth robustness check assesses the sensitivity of our results when relaxing the assumption that evapotranspiration after transitioning or deforestation goes to zero. We perform this robustness check because there are large uncertainties in the reduction of precipitation comparing data-driven evidence with CMIP-type Earth system model results. While deforestation impacts on precipitation decrease in some CMIP6 models are between 5 and 10% (ref. 74), experimental studies based on the BrasilFlux database indicate that substantial deforestation can decrease regional precipitation by up to 40%, particularly in regions such as the Amazon with very high moisture recycling ratios[65,105]. These findings align with assessments of deforestation-induced transitions[71,106,107] and observational evidence from later onsets of the wet season in the Amazon region[39]. Taken together, this suggests that precipitation reductions in response to deforestation are probably underestimated in some CMIP6 models. However, due to these uncertainties, we here perform the following additional robustness check: we keep the evapotranspiration at values of secondary vegetation after removing the primary forest. These values are available from ref. 103. In our sensitivity experiment, we find very good agreement with our experiments without deforestation: Amazon forest transitions are found at the same levels of global warming and at the same locations albeit at a slightly lower transition risk (compare Fig. 1a–c with Extended Data Fig. 7a–c). The experiments with deforestation are practically identical with respect to global warming levels of identified transitions, locations and extent of transitioned regions—this represents very consistent results (compare Fig. 1d–f with Extended Data Fig. 7d–f). Even further, it is important to note that, once forest vegetation is lost, evapotranspiration will generally remain low as only 2–4% return to secondary vegetation with high evapotranspiration after about a decade due to repeated clearance[108,109]. However, if secondary vegetation would be allowed to regrow, evapotranspiration could show high regenerative capacities after several years[109]. This suggests that most cleared areas may maintain low evapotranspiration for decades. These sensitivities are indirectly also covered by our robustness analyses through varying evapotranspiration values in secondary vegetation types (Extended Data Fig. 7).

Lastly, the fifth robustness analysis concerns two further scenarios for the adaptation capacities of the Amazon forest system. In the first scenario, we assume that drier regions in the forest may operate closer to their physiological limits and have smaller safety margins. In this experiment, we scale adaptation capacities with regional precipitation levels, choosing 1.25 s.d. for the wettest regions (that is, higher resilience to drier conditions) and 0.75 s.d. for the driest regions (that is, lower resilience to drier conditions). In the second scenario, we assume the opposite: wetter regions are less resilient (that is, 0.75 s.d.) to precipitation decreases and drier regions are more (that is, 1.25 s.d.). In both scenarios, we find that the results are robust against our main results (compare Fig. 1b with Extended Data Fig. 9).

In summary, our extensive five robustness checks show the very high robustness of our results, in particular regarding the most vulnerable regions, the levels of global warming where transition risks become pertinent and their quantitative agreement in the Amazon forest.

Owing to the very high computational demands, note that our robustness checks were carried out using the decadal averages (2020s, 2030s, …, 2090s), while the main analyses were carried out using running 10-year averages from 2026 (using the years 2021–2030) to 2095 (using the years 2090–2099) if not noted otherwise.

**Note on colour maps**
This paper makes use of perceptually uniform colour maps developed by F. Crameri[110].

**Reporting summary**
Further information on research design is available in the Nature Portfolio Reporting Summary linked to this article.

## Data availability

Historical MAP and evaporation data as well as the SSP-based land-use change scenarios are supplied at figshare[111] (https://doi.org/10.6084/m9.figshare.28191128). The moisture recycling data can be found in ref. 5 together with the respective SSP-based precipitation and evaporation data. The deforestation data[70] can be accessed online (https://doi.org/10.3334/ORNLDAAC/1153). Requests and/or questions regarding data should be addressed to the corresponding authors.

## Code availability

The code for this Article is available at figshare[111] (https://doi.org/10.6084/m9.figshare.28191128). The interacting dynamical systems approach is based on PyCascades[68]. Requests and/or questions regarding code should be addressed to the corresponding authors.

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

**Acknowledgements** We thank F. Pharand-Deschênes (Globaïa) redesigning the graphics in this work and P. Meijer for providing a figure for the travel distance of moisture for the Supplementary Note. N.W. acknowledges the Center for Critical Computational Studies at Goethe University, Frankfurt am Main for providing funding for this research; A.S. acknowledges funding from the Dutch Research Council (NWO) Talent Program Grant VI.Veni.202.170 and the ERC-Synergy project RESILIENCE, proposal no. 101071417; M.H. acknowledges financial support of Serrapilheira Institute (grant no. Serra-1709-18983). We acknowledge the European Regional Development Fund, the German Federal Ministry of Education and Research and the Land Brandenburg for supporting this project by providing resources on the high-performance computer system at the Potsdam Institute for Climate Impact Research.

**Author contributions** N.W. and A.S. conceived the study. N.W. designed the study, wrote the code and performed the simulations. A.S. prepared the input data. N.W. and B.S. drafted the figures. N.W., B.S., J.R., B.M.F., M.H. and A.S. drafted the original manuscript and wrote the revisions. A.S. led the supervision of this study.

**Funding** Open access funding provided by Johann Wolfgang Goethe-Universität.

**Competing interests** The authors declare no competing interests.

**Additional information**
**Correspondence and requests for materials** should be addressed to Nico Wunderling or Arie Staal.

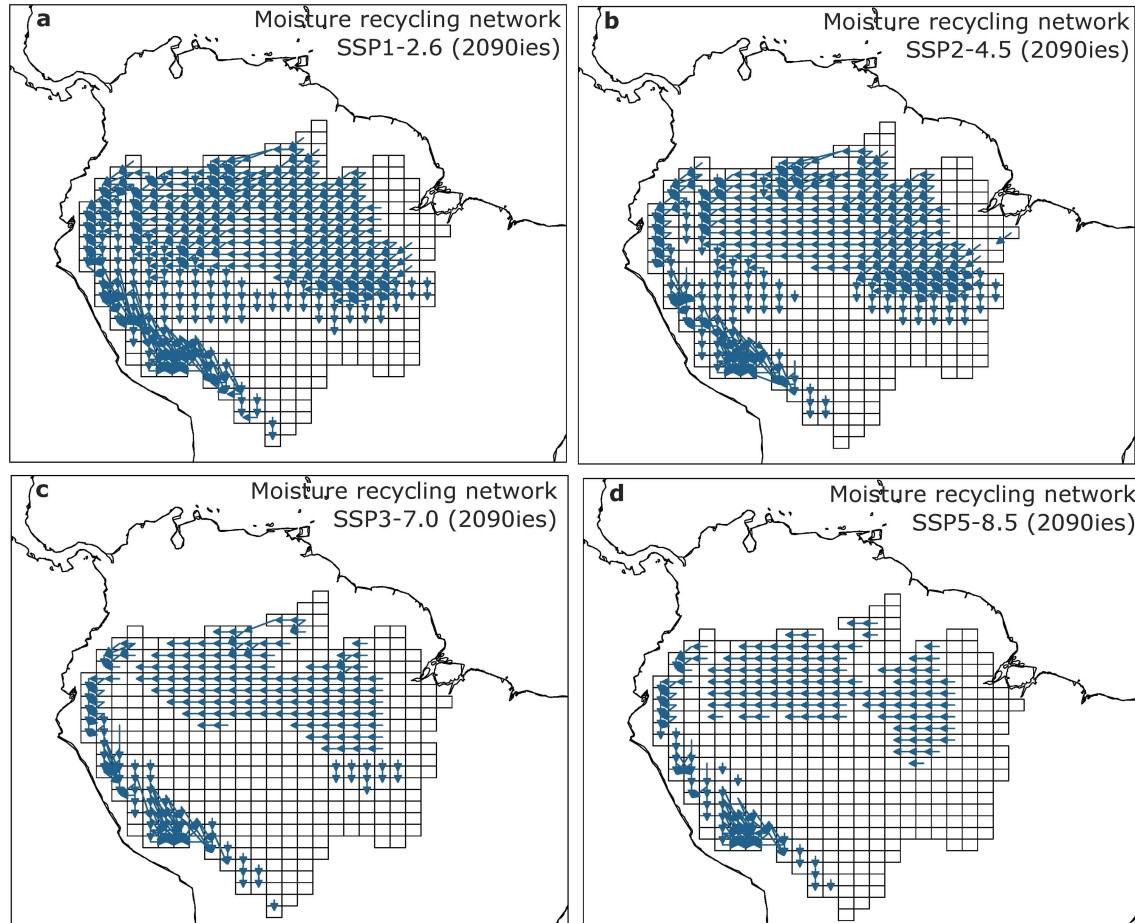

**Extended Data Fig. 1 | Atmospheric moisture recycling network.** Moisture recycling network for the decade 2090 for the scenarios **a**, SSP1-2.6, **b**, SSP2-4.5, **c**, SSP3-7.0 and **d**, SSP5-8.5. It can be seen that the strength of the moisture recycling network decreases with increasing global warming. This is expected since more global warming also means that precipitation over the Amazon forest region decreases (see also Fig. 2a, S2a). In all the simulations, moisture recycling values of 1 mm/month or more are taken into account but to increase visibility in this figure, only the moisture recycling links of more than 20 mm/month are shown.

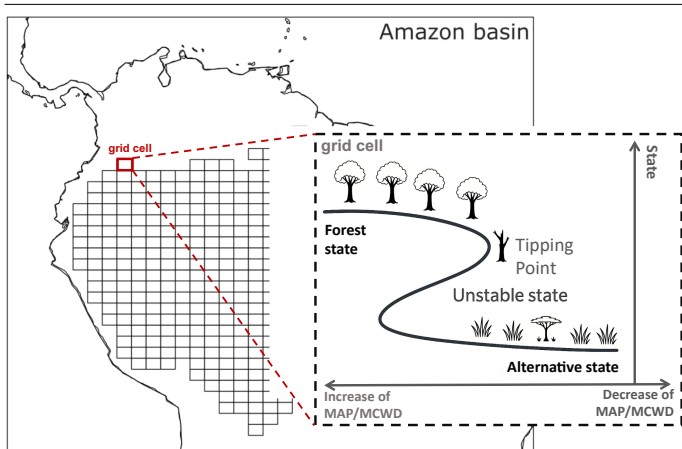

**Extended Data Fig. 2 | Depiction of the dynamical systems approach.** Map of the Amazon basin with a grid representing our study area. In this grid, each grid cell is represented by a dynamical systems equation (Eq. 2) with two stable equilibrium states (forest and alternative state, see inset). The individual and local scale tipping dynamics can spread via the coupling from the atmospheric moisture recycling network (examples are shown in Extended Data Fig. 1).

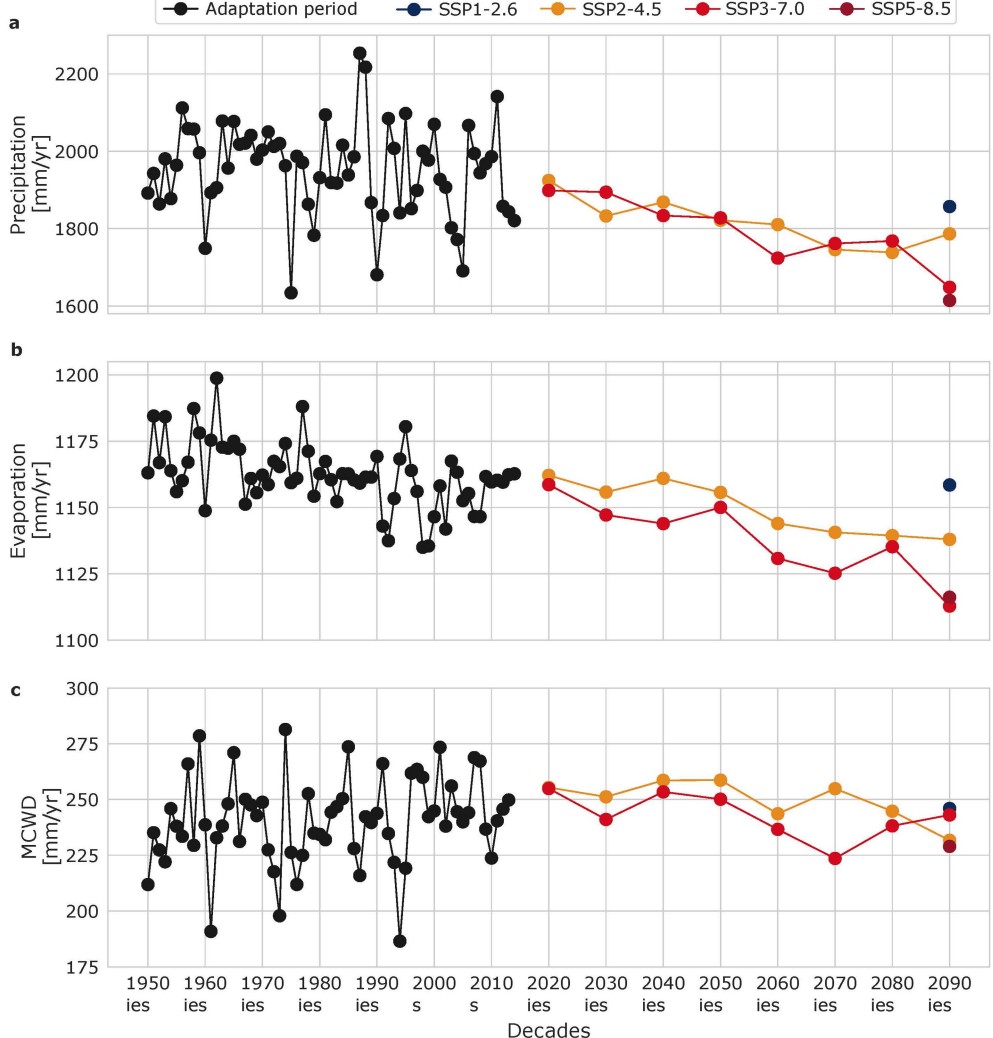

**Extended Data Fig. 3 | Time series of environmental variables.** Yearly values for **a**, precipitation (MAP), **b**, evaporation, and **c**, MCWD (derived from hydrological years starting in October until next year's September) for the adaptation period (1950-2014, see black dots) and for the four different SSP scenarios (averaged for the respective decade). The data is averaged over the entire Amazon basin. There is a clear trend that stronger climate change scenarios lead to lower precipitation and also to less evaporation across the Amazon basin. Following from that, the trend in MCWD does not show a clear trend across the scenarios. A spatial resolution for the average values of MAP and MCWD is shown in Extended Data Fig. 4.

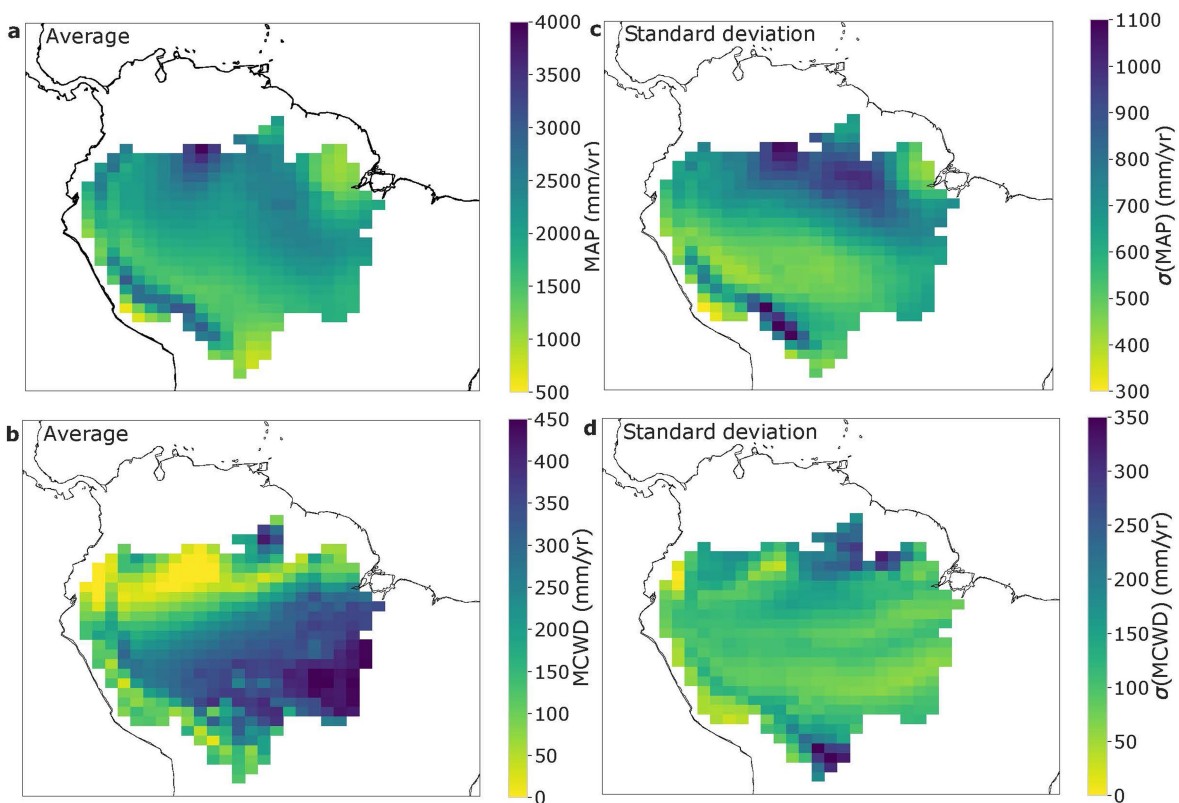

**Extended Data Fig. 4 | Moisture supply in the adaptation period (1950-2014) over the Amazon basin. a, c**, Average and standard deviation for MAP. **b, d**, Same for MCWD.

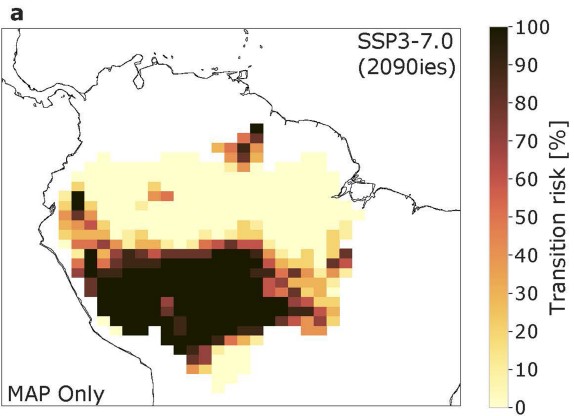

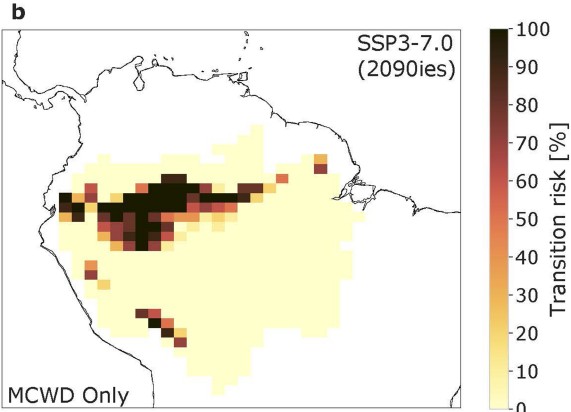

**Extended Data Fig. 5 | Transition risks in the Amazon forest for different SSPs broken down to MAP and MCWD contributions. a,b**, Respective contributions from MAP and MCWD as individual critical variables. The corresponding bar charts can be found in Extended Data Fig. 6a,b. Several robustness checks show robust results for (i) a mixture of adaptive capacities together with fixed critical thresholds following Flores et al.[1] (see Extended Data Fig. 6c,d) and (ii) for constant evapotranspiration of 100 mm/month (see Extended Data Fig. 6e, f). More details on robustness checks are denoted in the methods.

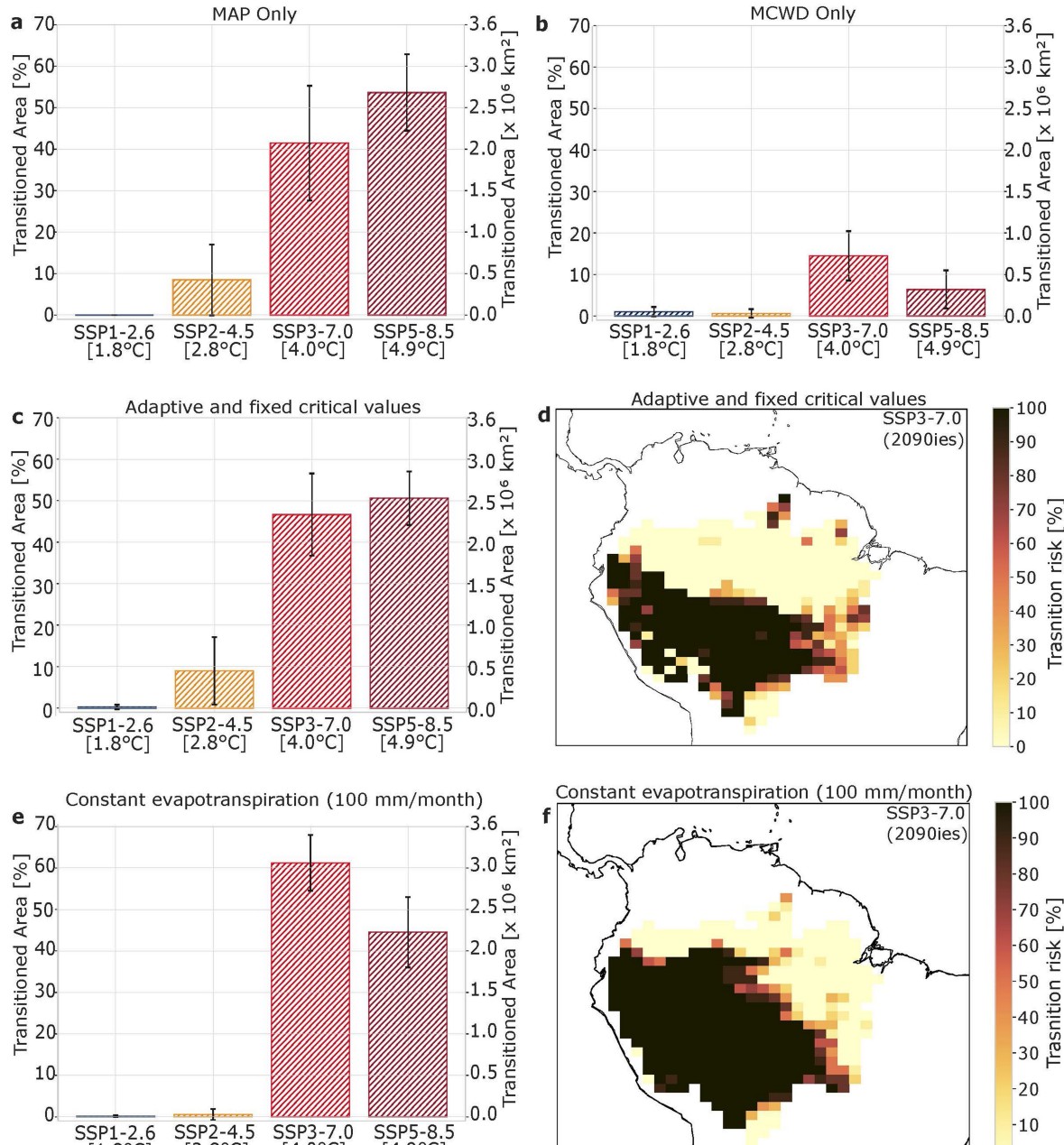

**Extended Data Fig. 6 | Crossing critical thresholds in the Amazon forest in response to global warming.** Supplementary investigations to Fig. 1 in the main manuscript. **a**, Robustly detected transition threshold for MAP as the sole critical threshold (not taking into account MCWD) as transitioned area in percent and km² of the entire Amazon basin, again with a sharp increase between an SSP2-4.5 (2.8 °C of global warming) and SSP3-7.0 (4.0 °C of global warming) scenario at 2090. **b**, Same as for panel a for the case that MCWD is the sole critical threshold (not taking into account MAP). Here, the overall increase of tipped area is still significant at SSP3-7.0 but not at the same magnitude as for MAP only. This hints at the fact that the region where MCWD induced transition events are relevant is smaller than for MAP (see also Extended Data Fig. 5).

**c**, Robustness check of our results for an admixture of adaptive critical thresholds (description in methods: adaptive threshold approach) as well as fixed critical thresholds following Flores et al. (2024)[1]. **d**, Regional resolution of transition risk for an SSP3-7.0 scenario in the 2090ies. **e,f**, Same as in c,d but for constant evapotranspiration values of 100 mm/month. For the details of all robustness checks, see Methods: robustness checks. For the panels **a-c, e**, the presented mean values (and error bar values) are the spatial average over the Amazon basin consisting of 416 grid cells. Of these 416 grid cell mean values (and error bar values), each is constructed from the mean (and the standard deviation) of ten ensemble members (see Methods: Ensemble construction).

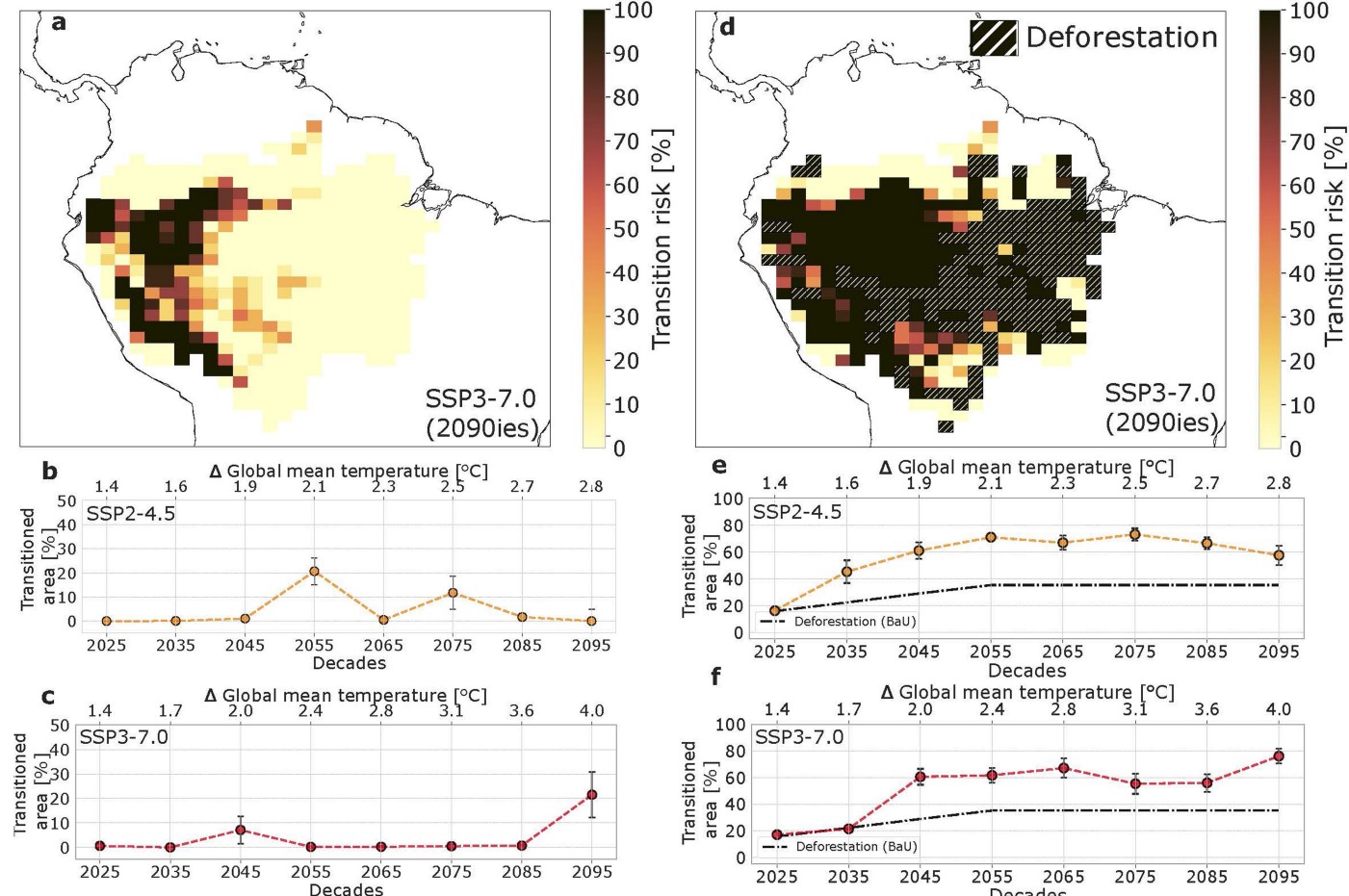

**Extended Data Fig. 7 | Robustness check with remaining evapotranspiration of secondary vegetation after transitioning and/or deforestation.** **a-c**, Robustness check with global warming only and no additional deforestation. **a**, Regions of the Amazon forest most at risk of crossing critical thresholds (western part of the Amazon basin) for an SSP3-7.0 scenario at 4.0 °C (2090ies decade) of global warming. **b**, Transitioned area across the 21st century for SSP2-4.5 and **c**, for SSP3-7.0. **d-f**, Same as for a-c but including additional deforestation. These sensitivity analyses show robust results as compared to our result in the main manuscript (compare with Fig. 1). For the panels **b,c, e,f**, the presented mean values (and error bar values) are the spatial average over the Amazon basin consisting of 416 grid cells. Of these 416 grid cell mean values (and error bar values), each is constructed from the mean (and the standard deviation) of ten ensemble members (see Methods: Ensemble construction).

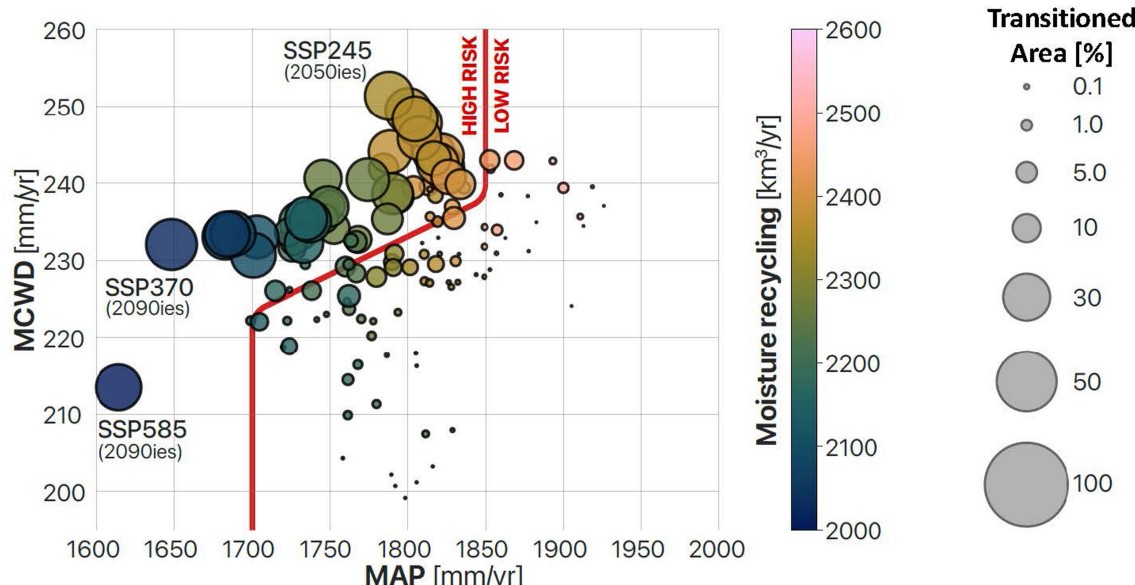

**Extended Data Fig. 8 | Critical thresholds in annual water supply (MAP, MCWD) with respect to strength of the moisture recycling network.** Transitioned area (size of the circle; in percent of the entire Amazon basin) dependent on MAP and MCWD. The thick red line separates low transition risks from high transition risks in the 2D-plane of MAP and MCWD. The colour of the circle depicts the strength of the moisture transport network across the entire Amazon basin that responds essentially linearly to MAP levels.

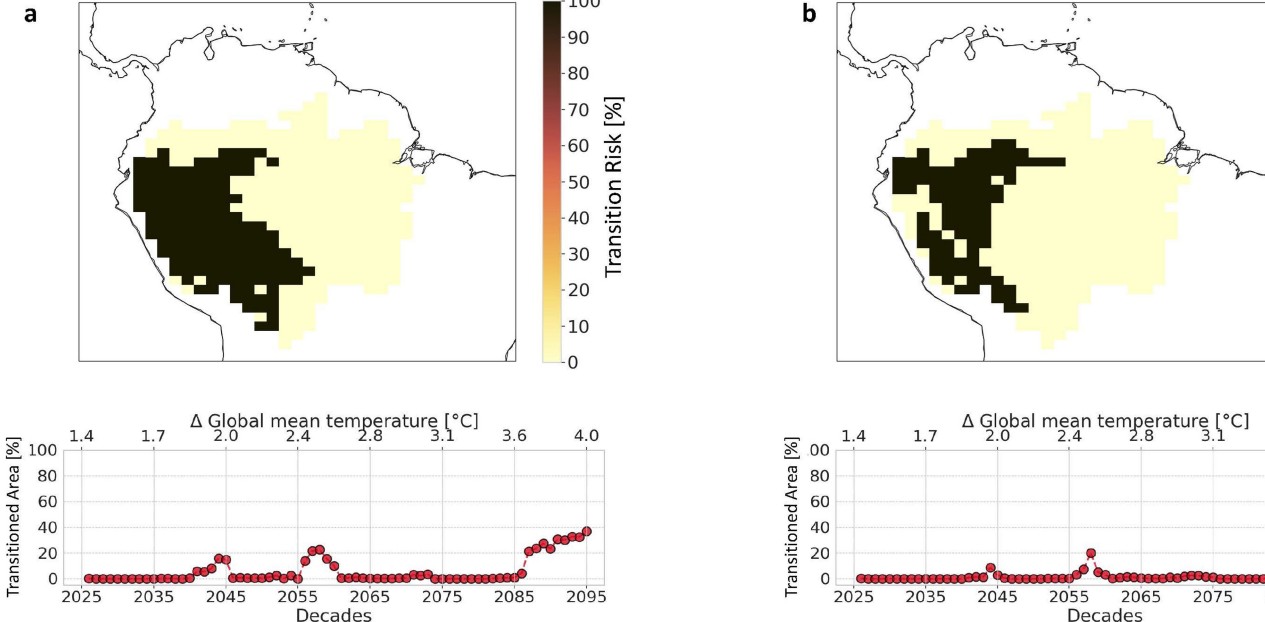

**Extended Data Fig. 9 | Robustness check with varying scenarios on adaptive capacities. a**, Scenario where adaptive capacities of drier vegetation is lower (0.75 standard deviations) and wetter vegetation is higher (1.25 standard deviations). The transition risk map is shown for the decade 2090-2099 in a SSP3-7.0 scenario and the transition response throughout the 21st century (time series) is shown following an SSP3-7.0 scenario. **b**, Same as in panel a but with a scenario for adaptive capacities where drier vegetation has higher resilience (1.25 standard deviations) and wetter regions lower resilience (0.75 standard deviations). In both cases, our results are qualitatively robust against Fig. 1b of the main manuscript concerning the regions most at risk and quantitatively robust concerning the level of dangerous global warming triggering a 20–40% tipping event in the Amazon forest (that is, between 3.7 and 4.0 °C of global warming).

*Double-anonymous peer review submissions: write DAPR and your manuscript number here instead of author names.*

# Reporting Summary

## Statistics

For all statistical analyses, confirm that the following items are present in the figure legend, table legend, main text, or Methods section.

| n/a | Confirmed | |
|---|---|---|
| ☐ | ☒ | The exact sample size (*n*) for each experimental group/condition, given as a discrete number and unit of measurement |
| ☐ | ☒ | A statement on whether measurements were taken from distinct samples or whether the same sample was measured repeatedly |
| ☒ | ☐ | The statistical test(s) used AND whether they are one- or two-sided *Only common tests should be described solely by name; describe more complex techniques in the Methods section.* |
| ☒ | ☐ | A description of all covariates tested |
| ☒ | ☐ | A description of any assumptions or corrections, such as tests of normality and adjustment for multiple comparisons |
| ☐ | ☒ | A full description of the statistical parameters including central tendency (e.g. means) or other basic estimates (e.g. regression coefficient) AND variation (e.g. standard deviation) or associated estimates of uncertainty (e.g. confidence intervals) |
| ☒ | ☐ | For null hypothesis testing, the test statistic (e.g. *F*, *t*, *r*) with confidence intervals, effect sizes, degrees of freedom and *P* value noted *Give P values as exact values whenever suitable.* |
| ☒ | ☐ | For Bayesian analysis, information on the choice of priors and Markov chain Monte Carlo settings |
| ☒ | ☐ | For hierarchical and complex designs, identification of the appropriate level for tests and full reporting of outcomes |
| ☒ | ☐ | Estimates of effect sizes (e.g. Cohen's *d*, Pearson's *r*), indicating how they were calculated |

*Our web collection on statistics for biologists contains articles on many of the points above.*

## Software and code

Policy information about availability of computer code

| Data collection | N/A |
|---|---|
| Data analysis | The code of this manuscript (also regarding data analysis) can be found at https://doi.org/10.6084/m9.775 figshare.28191128. The data analysis was further carried out based on the PyCascades software package version 1.0 (developed in Wunderling et al., 2021, European Physical Journal Sp. Topics, 1-14). |

For manuscripts utilizing custom algorithms or software that are central to the research but not yet described in published literature, software must be made available to editors and reviewers. We strongly encourage code deposition in a community repository (e.g. GitHub). See the Nature Portfolio guidelines for submitting code & software for further information.

## Data

Policy information about availability of data

All manuscripts must include a data availability statement. This statement should provide the following information, where applicable:
- Accession codes, unique identifiers, or web links for publicly available datasets
- A description of any restrictions on data availability
- For clinical datasets or third party data, please ensure that the statement adheres to our policy

The historical MAP and evaporation data as well as SSP-based land-use change scenarios are supplied under https://doi.org/10.6084/m9.775 figshare.28191128.

The moisture recycling data can be found in Staal et al. (2025, Earth System Dynamics, 16, 215-238) together with the respective SSP-based precipitation and evaporation data. The deforestation data can be accessed via the following doi: 10.3334/ORNLDAAC/1153.

# Research involving human participants, their data, or biological material

Policy information about studies with human participants or human data. See also policy information about sex, gender (identity/presentation), and sexual orientation and race, ethnicity and racism.

| Reporting on sex and gender | N/A |
|---|---|
| Reporting on race, ethnicity, or other socially relevant groupings | N/A |
| Population characteristics | N/A |
| Recruitment | N/A |
| Ethics oversight | N/A |

Note that full information on the approval of the study protocol must also be provided in the manuscript.

# Field-specific reporting

Please select the one below that is the best fit for your research. If you are not sure, read the appropriate sections before making your selection.

☐ Life sciences  ☐ Behavioural & social sciences  ☒ Ecological, evolutionary & environmental sciences

For a reference copy of the document with all sections, see nature.com/documents/nr-reporting-summary-flat.pdf

# Ecological, evolutionary & environmental sciences study design

All studies must disclose on these points even when the disclosure is negative.

| Study description | For the Amazon tipping experiments due to deforestation and global warming, 1.25 million ensemble members were generated (~3000 per grid cell in the study area of the Amazon basin) and run through the PyCascades:Amazon software framework leading to transition risk assessments. |
|---|---|
| Research sample | Precipitation, evaporation based on NorESM2, and moisture transport networks computed using the UT rack global hydrological model. Deforestation data publicly provided (see Data and Code availability statement). |
| Sampling strategy | Monte Carlo ensemble with 1.25 million ensemble members |
| Data collection | Earth System Model data that are publicly available |
| Timing and spatial scale | From 1950-2099 on a monthly timescale and a 1x1° grid basis for the study area (the Amazon basin). |
| Data exclusions | no exlusions |
| Reproducibility | For the five robustness checks (see methods section "Robustness checks"), the data needed to be prepared individually, and also the code was prepared and checked separately. Through these separate test, high confidence in reproducibility is given. Further, regarding data and code availability, see the respective statement above. |
| Randomization | Random sampling (see methods section "Ensemble construction") |
| Blinding | N/ A because random ensemble was generated |

Did the study involve field work?  ☐ Yes  ☒ No

# Reporting for specific materials, systems and methods

We require information from authors about some types of materials, experimental systems and methods used in many studies. Here, indicate whether each material, system or method listed is relevant to your study. If you are not sure if a list item applies to your research, read the appropriate section before selecting a response.

## Materials & experimental systems

| n/a | Involved in the study |
|-----|----------------------|
| ☒ | ☐ Antibodies |
| ☒ | ☐ Eukaryotic cell lines |
| ☒ | ☐ Palaeontology and archaeology |
| ☒ | ☐ Animals and other organisms |
| ☒ | ☐ Clinical data |
| ☒ | ☐ Dual use research of concern |
| ☒ | ☐ Plants |

## Methods

| n/a | Involved in the study |
|-----|----------------------|
| ☒ | ☐ ChIP-seq |
| ☒ | ☐ Flow cytometry |
| ☒ | ☐ MRI-based neuroimaging |

## Plants

| Seed stocks | N/A |
|-------------|-----|
| Novel plant genotypes | N/A |
| Authentication | N/A |

