## [Peer Review File · Nature]

Quantified Amazon transitions under joint warming and deforestation

Corresponding Author: Dr Nico Wunderling

Version 0:

Reviewer comments:

Referee #1

(Remarks on code availability)

(Remarks to the Author)

The manuscript addresses a critical issue: how changes in mean annual precipitation (MAP) and maximum climatological water deficit (MCWD) may drive Amazon forests toward a tipping point. Building on prior research on moisture transport across the Basin, the study makes a valuable contribution by quantifying potential changes in forest stability in the Amazon. The results suggest that while Amazon forests may exhibit resilience to regional warming, deforestation could significantly reduce their ability to cope with climatic changes, lowering critical thresholds beyond previous estimates. These findings are highly relevant to discussions on the Amazon's long-term stability/resilience. However, several points require clarification and further development:

1. The manuscript assumes that the Amazon exhibits two alternative stable states, with crossing a threshold leading to a critical transition within decades. While some models support this idea, observational evidence remains limited. For example, drought experiments suggest biomass reductions of 20-40% over a decade with a 50% precipitation decline. Furthermore, without drought-fire interactions, there is little evidence of a sharp, irreversible shift. Given the challenges in validating the model, the results should be presented with greater nuance, contextualized based on observational studies, experimental approaches, and other types of models (DGVMs).
2. The mechanisms linking deforestation and MAP/MCWD needs further clarification. If the linkage is primarily through deforestation-related reductions in evapotranspiration (ET), it is important to note that most CMIP6 models (but not all) already estimate some levels of deforestation effects on MAP and MCWD, which tend to be relatively small (<10%) (see comments below and Li et al., Nature Communications). It is unclear how such reductions in deforestation could lower the temperature threshold for forest collapse from 3.7-4.0°C to 1.5-1.9°C. A more detailed explanation of this mechanism is necessary. Note also that if the CMIP6 model already incorporate deforestation effects, raising the concerns about potential double counting deforestation effects climate -- if additional effects on climate are applied post hoc.
3. While deforestation reduces ET, particularly during transition months, the land uses replacing forests (e.g., pasturelands) maintain low ET levels (~1 mm). The discussion should also highlight the dominant role of transpiration over evaporation in these processes.
4. The manuscript presents adaptation capacity as a key factor in resilience, but it is unclear whether the assumed adaptations align with reality. While plants near physiological limits may develop traits to tolerate drier conditions, they often have smaller safety margins despite higher resistance. Treating adaptation capacity as a scenario (among others representing different levels of adaptation) rather than a quantitative metric may better reflect its uncertainty. Additionally, exploring how variability in vegetation adaptation influences tipping point probabilities would provide a more nuanced perspective. The assumption that drier forests are inherently more adapted to climate change warrants further scrutiny.
5. The manuscript assumes that once forests reach a tipping point, they remain in a low-ET state. However, in reality, forest recovery often occurs in the Amazon if seed dispersal and climatic conditions permit. For example, 19% of cleared areas experience regrowth, although most are later cleared again. While climate change may slow recovery rates, ET is typically one of the first ecosystem functions to return. The manuscript should provide a stronger justification for why forests would remain in a permanently degraded state rather than exhibit some level of recovery. Again, the non-ET recovery state could

be treated as a scenario rather than a representation of the current reality in the region.

6. The deforestation scenarios used in the study appear outdated and may not align with current trends. Given that the dataset's author has cautioned against overinterpreting these projections, a brief discussion of their limitations should be included. Nonetheless, incorporating these scenarios is still preferable to omitting deforestation effects entirely.

7. Although CO₂ fertilization is not the primary focus of this study, it is a key driver in CMIP6 climate projections and significantly affects both regional precipitation and temperature patterns over the Amazon Basin. Some estimates suggest that CO₂ fertilization contributes to up to 40% of regional rainfall changes, which in turn influences MAP and MCWD. Given that MAP and MCWD are critical factors in determining biomass storage and forest resilience in the proposed model, the sensitivity of vegetation to CO₂ could introduce a potential bias in our model interpretations and should be discussed in the manuscript.

Referee #2

(Remarks on code availability)

(Remarks to the Author)

Thank you for inviting me to review the paper: "Pinpointing Amazon Forest Tipping in Global Warming and Deforestation Pathways" by Wunderling et al.

The topic is, of course, of particular importance, and the risk of Amazon "dieback" remains one of the most frequently cited potential impacts of climate change. However, concurrently, deforestation also risks impacting Amazon lose, with its effects extending well beyond the specific areas of land use, due to adjusting terrestrial rainfall recycling.

To date, there is considerable debate regarding which factor poses a greater threat to resilience – climate change or deforestation. The main novelty of this manuscript lies in its simultaneous consideration of both drivers (climate change and deforestation), enabling a combined weighted assessment of the likely timing of rainforest loss, contingent on changes to either factor.

The immediate impression is of a high-quality manuscript; it fits well with the Nature format, and the diagrams are informative. I particularly appreciate the mapping onto an underlying dynamical system, which facilitates a deeper understanding of the system. This is a notably novel feature, as it enables the manuscript to connect with the theory of nonlinear dynamical systems. Dynamical systems form the foundation of the "tipping point" theory developed in the mathematical sciences, yet very few have rigorously mapped environmental issues onto such equations. Indeed, the authors might consider adding a line that highlights this advancement, as it provides methodological pointers to researchers attempting to characterise tipping points in other areas of the Earth system.

My main criticism of the paper is that it should be much clearer about where assumptions are made and where uncertainties remain. I would not want this to be used as a reason for the paper's rejection by Nature, because the messages of the manuscript are important and have not been addressed in such a comprehensive manner by existing papers. At the same time, Nature is a high-profile journal, and reviewers are frequently asked to ensure that all uncertainties are considered. When presented well, emphasising remaining caveats actually enhances a paper, rather than weakening it. Caveats do not undermine the analysis, which often remains valid in light of future data or research; it is simply that the full precision is not yet known.

Some of the uncertainties are as follows:

(1) I am willing to accept that the only ESM capable of providing boundary conditions to the atmospheric transport model is NorESM2. However, one possible additional "robustness" check is to compare the atmospheric fields, at least for the contemporary period, against one of the reanalysis products. Is that possible and sensible?

(2) As the authors will be aware, the original Cox et al. 2000 paper on Amazon "dieback" assessed numerous carbon cycle-related factors that contribute to forest risk. Some of these factors may indicate that, under climate change, the parameters inherent in the two-state dynamical system could change. The influence of these factors may either foster resilience or result in the opposite effect. Factors encompass stomatal closure, which can conserve soil moisture, aiding resilience, or heightened respiration in a warming climate, which could exceed photosynthesis and contribute to earlier dieback.

(3) If I have understood correctly, the global warming estimates needed to drive the evapotranspiration equations have similarly been based only on the NorESM2 ESM? There are, of course, around 30-40 different climate models available in the CMIP6 database, and sometimes their full use provides natural bounds of uncertainty. I realise that this is a big task to utilise all ESM data, and I am not necessarily suggesting that this should happen. One way to circumvent this is to express changes in terms of global warming itself. The authors do this in their Figure 2, namely, the third coloured "axis" characterised by the colour bar showing temperature thresholds. However, it is important to note that depending on the rate of warming projected by different ESMs for a given SSP scenario, the timing of crossing identified thermal thresholds is less certain. At the very least, the authors should acknowledge this more fully. Statements suggesting that a tipping point could be reached at some point in the 2030s may not be valid under a standard SSP if calculations are made with an ESM that has low climate sensitivity. (It might be possible to incorporate this uncertainty into the authors' Monte Carlo methods, without downloading the full CMIP6 ensemble, if a reference can be found for the range of projected warmings in South America for a given standard SSP).

One aspect I particularly appreciate about the manuscript is that it gives equal importance to assessing climate forcings and deforestation forcings. However, while Figure 2 is very informative, it does not address deforestation. The corresponding diagram that represents deforestation (for a business-as-usual scenario) is Figure S13. Incorporating Figure S13 into the main paper could greatly enhance its usefulness, either as a new Figure 3 or as additional panels in the existing Figure 2. The paper has space available – the current version of the main paper features only two figures.

There are some small edits that could improve readability. As far as possible, please make the captions self-contained. It might, for instance, help in Figure 2 to remind the reader what the NAP and MCWD represent. Something like: "...global warming scenarios dependent on MAP and MCWD, that is annual rainfall and dry season intensity".

For Figure 2b, another possibility is to make the large external circulate ring instead as an inset histogram?

The authors might consider speculating in the Discussion section about whether the coefficients in the dynamical system (Equations (2) and onwards) could be more directly related to ecological knowledge. Are there certain features of the more reliable dynamic global vegetation models (DGVMs) that would reveal and/or elucidate the parameter values of the dynamical system? Proposing this for future analysis could effectively establish a roadmap between physical climate modelling and ecological assessment – a path that this current paper has initiated and navigates very well.

In summary, this manuscript is refreshing as it considers the simultaneous effects of climate change and deforestation. It genuinely strives to align understanding with dynamical systems that can demonstrate tipping point behaviour in a rigorous manner, allowing for transitions between states and hysteresis, as indicated by the cubic behaviour of Equation (2). It appears to be a paper that, with a bit more refinement, could significantly contribute to our knowledge of climate change and its impacts, making Nature a very appropriate journal for its publication.

Therefore, the notes above represent a set of suggestions that may be beneficial to the authors. In my opinion, the primary concern is the need to slightly elaborate on the caveats and to be more explicit about uncertainties. By doing so, it will enhance the manuscript rather than detract from it. Such descriptions enable a strong focus for future researchers to "home in" on the remaining areas where knowledge is incomplete. I would be very pleased to review subsequent versions of the manuscript, and I sincerely hope that Nature provides an opportunity for the paper to be considered for potential publication, allowing the authors to create a new, slightly more refined version.

Referee #3

(Remarks on code availability)

(Remarks to the Author)

The present paper focuses on the on the very interesting topic of the tipping points, a topic of undoubted scientific and socio-economic and ecological interest given its importance for the water resources.

It is well known that the Amazon biome is threatened by deforestation, which affects the hydrological cycle in the region and in the adjacent regions that it serves as a source of moisture for rainfall. Changes in its internal structure due to deforestation, changes in land use and internal droughts make the Amazon one of the most interesting and important regions for analysing future or even present changes. It is therefore of scientific interest to study the recycling capacity of the Amazon and its changes; and to link it to the expected deforestation.

The present manuscript discusses the adaptive capacity of the Amazon rainforest in the face of climate change and deforestation, using climate modelling scenarios, adding additional projected deforestation data, and the study of moisture recycling as this is crucial for determining the resilience to changes.

The methodology combines several data sets, some of which are a sub-product of modelling. In this line, the authors should add more information about all uncertainties of the datasets and the possible influences in the results. The authors show a robust check, but concerning the results of the paper, but it is needed a more explanation about the data itself. In some cases, reference is made to other papers, and it could be difficult for readers to check all the papers.

The study assesses, using a dynamic system model, the risks of the tipping points under different emissions (focussing mainly in two) and land-use scenarios. It shows that without deforestation, the critical threshold for global warming is 3.7-4.0°C, where a third of the forest could lose stability. With deforestation, the tipping point is reached with only 1.5-1.9°C of warming and a 22-28% loss of forest cover, which could occur as early as the 2030s. Most of these events are triggered by intense droughts, which have large-scale cascading effects. The study highlights, and recommend, the urgency of keeping global warming below 1.5°C, halting deforestation (around 15%) and restoring affected areas to avoid critical risks in the Amazon.

All these results combined, as commented, data and models. The model to construct the story line of adaptation and deforestation is out of the expertise of this reviewers to provide a detail analysis of the results and argumentations. Therefore, the review of this point of the paper should be based on the opinion of other reviewers.

However, this reviewer is able to offer a critical opinion on the using of the dataset derived from the atmospheric moisture tracking model UTrack (as moisture transport is the central theme of this reviewer's research), and which is used here to incorporate evaporation and precipitation due to recycling process.

In this paper, the UTrack model and method of computing associated is employed to analyse the contribution of the Amazon itself in terms of evaporation to its own precipitation, the recycling. Lagrangian methods are used for this purpose, but it is not the only one (this comment should be corrected in the manuscript, line 58-59). Moreover, exists more Lagrangian models using SSPS scenarios, even if not for the calculation of recycling, but they can be used. So, the line 84 about "the unique position" is a sentence that needs to be corrected [1, and references therein].

Focusing on the moisture source concepts, there is a wide range of techniques that do not always give the same results, since the UTrack model chosen by the authors is a correct method, but not the state-of-the-art tracking model (line 13). This should be discussed.

In terms of moisture transport, the article focuses on the recycling capacity of the Amazon, but it is true for any region that its precipitation depends not only on internal flow, but also on external air masses with moisture that also influence its precipitation and thus the water cycle in its upper atmosphere and ground [2, and references therein]. Also missing is any reference to the impact of deforestation and rising temperatures under global change on precipitation downstream of the Amazon, on its natural sinks such as much of South America, or on other large basins such as La Plata. This reviewer is aware that the paper deals with consequences within the Amazon, but its ecological and global system importance should not be forgotten, and reference to possible consequences in external regions is a necessity for the scientific community.

As Amazon faces unprecedented pressures from global warming, deforestation and land-use change, a comprehensive analysis of the region is needed. This reviewer points out that all the data from the recycling simulation are referenced as coming from a previous work by one of the authors [3] that analyses several river basins, of which the Amazon is one. However, Amazon is not just a "random" region and, in particular, needs to be re-examined in greater depth. It is imperative to realise a complete and individual check for the Amazon Basin.

Particularly, UTrack uses NorESM data in this study and in Staal et al (2024), and in this paper submitted to Nature uses the same periods and scenarios and validations.

It is in this part where this reviewer has more doubts derived from the previous article.

The influence of using 10-year averages on the results is not clear, as the variability and particularity of each year in cases of drought events between wetter periods is smoothed out to a large extent. This should be taken into account and seen if there is an effect on the results.

Perhaps, and as the previous work is global, this issue was not emphasised, but as the authors now are dealing with a basin such as the Amazon, a particular study of the basin should be made, and not use previous data, without a double check (and shown in this actual paper), from a more generalised results. In addition, because the reanalysis data and models for the Amazon have many gaps, and for the past period analysed it is recommended to check any results with station data.

Simulation settings: are the same for the paper of Staal et al 2004? Why? The Amazon should be re-calculated. Is it enough this number of simulated particles in a region with high evapotranspiration?

Model evaluation: The authors evaluate by comparing the outputs from NORESM2 and ERA5, but the common period is too small. Figure A5 in Staal et al 2024 shows values with ERA5 from 2008-2017 and the simulations with NorESM2 with SPSS2-4.5 from 2015-2024, only 2 years in common. The period is not the same, and similarities in the map have not undergone statistical analysis. The evaluation should be redone for the Amazon region, using the complete period, and SPSS historical period. Why is the reason for using SPSS2-4.5 scenario for comparison? And why use this scenario as the baseline period? This aspect must be explained in the paper submitted to Nature.

For this reviewer, this aspect is not clear in the previous paper, and is a difficult aspect inherent in the new submission.

In Staal et al 2024 they said that the global pattern is "qualitatively similar". This is not statistically valid. And the authors stated that the choice of a 10-year average for the recycling ratios largely overlaps with a 30-year period. In addition, the authors confirm the existence of a systematic bias. Does it occur over the Amazon? In which sense?

The authors used a "quasi-wind" speed as monthly mean U and V values (as explained in Staal et al 2024, but not in the actual paper). Is this approach valid over the Amazon? The authors should take into account that over the Amazon a high-intensity wind structure as is the South American Low-Level Jet (SALLJ) acts as the main flow and defines the transport patterns in the region and south of the Amazon [4].

Moreover, what is the deviation from this quasi-wind vs the reanalysis and NorESM winds? What is the effect of using wind at 00 UTC (only one time) in the study for recycling over the Amazon? Does the maximum wind speed correspond with this time? Or not? It is very important to see the differences between the time with minimum and maximum wind speed, as the moisture is transported differently.

How is the moisture recycling uncertainty due to this approach? The bias in the wind pattern could be enlarged or not when the Lagrangian model was applied. This concern should be clarified in depth as the Amazon has a very impact on the local climate, and on regional climate.

Concerning the SPSS scenarios: It should be much clearer in the article the reasons for using only two climate change

scenarios (SPSS2-4.5 and SPSS3-7.0), and consider SPSS5-8.5 as a complementary scenario. In addition to explaining why to use as baseline period one of the scenarios to show the results, the SPSS2-4.5.

All the commented doubts and the results (and previous results) carried over from the moisture modelling (evaporation and precipitation) should be explained and highly validated, as the region is of very high interest and because the data are used in the following models and calculations in the paper submitted. The moisture recycling results may be affected by the lack of robustness and representativeness of the NorESM data and future scenarios.

This is why I recommend a more in-depth analysis of recycling, its behaviour in the future, and above all, a stronger statistical and comparative analysis to give robustness to the final results.

References

- [1] J. Eiras-Barca, J. C. Fernández-Álvarez, G. Alvarez-Socorro, S. Rahimi-Esfarjani, P. Carrasco-Pena, R. Nieto, L. Gimeno (2025) Projected changes in moisture sources and sinks affecting the US East Coast and the Caribbean Sea, *Annals of the New York Academy of Sciences*, DOI: 10.1111/nyas.15289
- [2] L. Gimeno, M. Vázquez, J. Eiras-Barca, R. Sorí, M. Stojanovic, I. Algarra, R. Nieto, A.M. Ramos, A.M. Durán-Quesada, F. Dominguez (2020) Recent progress on the sources of continental precipitation as revealed by moisture transport analysis, *Earth Science Reviews*, <https://doi.org/10.1016/j.earscirev.2019.103070>
- [3] Staal, A., Meijer, P., Nyasulu, M. K., Tuinenburg, O. A., and Dekker, S. C.: Global terrestrial moisture recycling in Shared Socioeconomic Pathways, *EGUsphere* [preprint], <https://doi.org/10.5194/egusphere-2024-790>, 2024.
- [4] Jones, C., Mu, Y., Carvalho, L.M.V. et al. The South America Low-Level Jet: form, variability and large-scale forcings. *npj Clim Atmos Sci* 6, 175 (2023). <https://doi.org/10.1038/s41612-023-00501-4>

Version 1:

Reviewer comments:

Referee #1

(Remarks on code availability)

(Remarks to the Author)

The authors invested a tremendous amount of effort in revising the manuscript and have carefully addressed several reviewer comments, particularly points 3, 4, and 6. The manuscript is clearly stronger as a result of these efforts. However, there remain important issues that require clarification and refinement.

Point 1. Tipping point unclear.

The authors provided an extensive response to concerns about evidence for critical transitions in the Amazon. However, their reply conflates processes such as natural variability, transient reductions in biomass, and disturbance-driven changes (e.g., fire), none of which necessarily represent critical transitions. Some of these processes may be better described as temporary state changes, as in *Silvério et al. (2022)*. By contrast, the work of *Hirota and Staver* emphasizes the distribution and coexistence of alternative biomes under particular climatic conditions, which is conceptually distinct from forest degradation.

This highlights a central issue: the manuscript does not offer a clear or consistent definition of what is meant by a “tipping point.” In several passages, the authors claim to show a change in stable state. By definition, such a change would imply a transition from forest to a non-forest ecosystem. Yet, with the exception of *Hirota and Staver*, most of the cited examples do not represent true critical transitions. This is important because many readers will interpret the title and abstract to mean that most Amazon forests will cease to exist once thresholds are crossed. That is not what the manuscript is currently demonstrating.

If the intended meaning is a biomass reduction (e.g., 40%), the system still qualifies as forest, since biomass is largely stored in large trees. Even fire-induced biomass losses beyond that threshold do not necessarily represent a new stable state, as many studies show substantial recovery potential over decades. Without greater precision, the manuscript risks conflating abrupt but transient biomass reductions with genuine tipping points leading to critical transitions.

On drought, it is important to recognize that while experiments do not capture long-term recycling, historical events such as the Little Ice Age suggest that even under extreme dry conditions, widespread tropical forest collapse was not observed—impacts were largely confined to forest edges.

Further clarification is also required regarding the following statement:

“Furthermore, without drought-fire interactions, there is little evidence of a sharp, irreversible shift. Given the challenges in validating the model, the results should be presented with greater nuance, contextualized based on observational studies, experimental approaches, and other types of models (DGVMs).”

The authors explain that they simulate risks of transitions without allowing for recovery. While mathematically understandable, this assumption weakens claims that the Amazon will shift to an alternative **stable** state. Moreover, the absence of explicit timescales makes the results difficult to interpret in conservation or climate mitigation/adaptation contexts. The reference to ten years as a relevant timescale is particularly problematic, as this is far too short to capture the long-term vegetation–climate feedbacks involved – which should occur in much longer time scales, likely in thousands of years. This raises concerns that the interpretation of the results does not adequately reflect the model’s assumptions and limitations.

Point 2. Deforestation impacts on rainfall

The manuscript emphasizes studies suggesting strong impacts of deforestation on rainfall. However, many other studies report considerably smaller effects. For example, Spracklen’s review concludes that even complete deforestation of the Amazon would reduce rainfall by approximately 16–18%. Given that the authors rely on a single modeling framework, it is essential to acknowledge this broader body of work and to present a more balanced discussion of the uncertainties in rainfall sensitivity to deforestation.

Point 3. Addressed.

Point 4. Addressed.

Point 5. Forest regeneration capacity

This point remains insufficiently developed. The central issue is that Amazonian forests have high natural regeneration capacity, but this is often reduced by repeated clearing before they reach the legal threshold for recognition as forest under Brazilian law. This reflects land-use dynamics more than ecological limitations. Moreover, several studies—including one cited by the authors—show that evapotranspiration tends to recover rapidly after disturbance. This nuance should be incorporated into the manuscript.

Point 6. Addressed.

Point 7. Role of CO₂

The response here does not fully resolve the concern. The issue is not what is known or unknown about CO₂ effects per se, but rather that the framing of “tipping points” could be strongly influenced by CO₂-driven reductions in rainfall in the datasets used for modeling, especially given that the model is run offline. This potential bias should be explicitly acknowledged.

Referee #2

(Remarks on code availability)

(Remarks to the Author)

Thank you once more for inviting me to assess the paper: “Pinpointing Amazon forest tipping in global warming and deforestation pathways” by Wunderling et al.

What is immediately obvious from scanning the entire set of requests and replies is how thorough the suggestions are and how, overall, the authors have responded with great care.

After reviewing the responses to my requests, I have a few further points below that the authors might consider.

Author Answer #3. Thank you for considering the possibility of using ERA5 data to force the UTrack transport model. I find it particularly interesting that the authors regard NorESM to be more accurate, given that ERA5 is likely “too wet”. Ultimately, we need the best possible data and a broader range of climate models to (1) ensure there are no initial biases when simulating the current climate, and (2) enable us to estimate a wide range of uncertainties for future projections. I am content that this particular paper does not scan across all CMIP6 models, but this should be more clearly acknowledged as a “next step” in this line of research. Although not perfect, differences across CMIP6 models naturally and usefully define uncertainty bounds on expected future changes. Maybe stronger recommendations can be made in the Discussion and Conclusions, urging reanalysis products to become as accurate as possible, especially in remote regions? And, additionally, it should be requested that the CMIP6 ensemble ideally stores the relevant variables needed to force tracking models at finer scales (and finer temporal resolutions?).

Author Answer #4. An attractive feature of the manuscript is the mapping of findings onto a nonlinear dynamical system structure, with the full equation (2). The cubic equation format is a generic form for illustrating tipping points with two states. The advantage of simpler descriptions is that they allow scanning of parameters. It is appreciated that the authors have done exactly this, by modifying parameter sigma (which indirectly, is a form of bifurcation parameter). The authors refer to this parameter as describing adaptation capacity. It is good that the authors included the related diagrams as a new Figure S20. Please make the citation in the main paper to this figure as clear as possible, as it provides a robust form of uncertainty

analysis.

Author Answer #4. Would the authors be willing to expand a bit more (for example, in the Discussion) on the factors they expect to influence the sigma parameter, such as water access features, stomatal response, and acclimation effects? An additional benefit of a simpler underlying model with clearly defined parameters is that it simplifies mapping future research and findings (for example, on land surface resilience) onto this common framework. Emphasising this point in the Discussion is important, as it will also encourage citation of the manuscript for its contribution in providing a key dynamical system with parameters that are useful for others to set.

Author Answer #5. Thank you for this amendment, and I can see these additional lines (i.e. 509-514). Please help the reader a little more though, so it is possible to see how a different ECS value translates through Eqn (1). Presumably, this is in the standard equation for evaporation. The reason this is helpful is because it allows translation of other analyses, such as the ongoing efforts to constrain ECS more tightly, onto the dynamical system present in this paper for Amazon multi-states.

Author Answer #8. This partly answers the point raised above about linking parameter sigma to underlying physical or ecological processes.

In summary, I believe the paper is now very close to publication. The minor points raised above concern ensuring that the underlying equations, assumptions, and open questions are communicated as clearly and unambiguously as possible. Therefore, I suggest the authors review the manuscript one more time, considering how other researchers conducting Amazonian studies might most easily grasp the ideas and equations presented here, so they can incorporate them into future work. The significant benefit of presenting a set of underlying equations is that it provides a framework for future insights and datasets to align with. This clarity should be emphasised as much as possible in the concluding section of the manuscript.

Again, I have enjoyed reviewing this manuscript.

Referee #3

(Remarks on code availability)

(Remarks to the Author)

Revised Reviewer Comments

As I mentioned in my initial review, this manuscript focuses on the use of the trajectory model. The authors have made a commendable effort to recheck the results as requested and to clarify the methodology and its limitations. The inclusion of an additional SSP has increased the manuscript's value, and the comparison of precipitation and wind across different datasets is a useful contribution. The manuscript has improved considerably.

However, a number of important issues still remain and need careful revision before the manuscript can be considered for publication.

Major points:

- The UTrack model is not properly introduced. Its first mention (line 66) appears disconnected from the context. A clear introduction is needed, including a brief description and a reference to the Methods section.
- Figure captions throughout the manuscript contain results, interpretations, or robustness statements that should be moved to the main text. Captions should be limited to figure description.
- Several figures (e.g., S1–S3, S8–S12, S20) either lack clarity, omit explanation of elements (e.g., dashed circles, reference lines), or exclude SSP scenarios inconsistently. These require revision and, in some cases, expansion (e.g., SSP5-8.5 maps).
- The rationale for using SSP2-4.5 because it aligns with recent climate trajectories is important and must appear in the main text, not only in responses to reviewers.
- The methodology requires more context. The UTrack model should be discussed in relation to other Lagrangian approaches, acknowledging differences in results. Additionally, the choice of a 30-day trajectory period should be justified, as it exceeds typical atmospheric residence times.
- The supplementary validation with ERA5 data requires clarification: (i) the meaning of “quasi-validation,” (ii) the reason for selecting only 5 years, and (iii) why only SSP2-4.5 was compared. Figure R10 should be added with an explanation.

Minor points:

- References are missing in several places (e.g., SSPs, 10-year averages, optimistic deforestation scenarios).
- Acronyms (e.g., MAP, MCWD) should not be repeated once defined.
- Redundant or unclear text appears at multiple points (e.g., lines 262–268, figure references).

- Typographical and language issues persist (e.g., line 67 on NorESM2 data type).

General comment:

The study calculates tipping points based on recycled precipitation. It should be stated more clearly that Amazon rainfall is not exclusively sourced from within the basin and that external moisture contributions could affect results. Please add references on Lagrangian studies of Amazon moisture sources.

Overall, the manuscript is promising but requires substantial clarification and restructuring, particularly regarding figure captions, model introduction, and methodological justification.

Point by point comments:

1. Line 62: A reference on SSPs is required.
2. The first mention of the UTrack model appears in line 66, with the phrase “This version...”. However, the model has not been introduced in the text before this point. The sentence reads disconnected from the context. The model should be properly introduced here. This paragraph must be carefully rewritten, providing at least a minimal introduction to UTrack. Readers should also be directed to the Methods section for a complete description.
3. Line 67: NorESM2 is a model that generates data, but what type of data? The sentence is poorly written and lacks precision.
4. Line 96: Fig S1 show which is referred in the sentence?
5. Figures S1–S2: Captions currently contain results and interpretations, which should be removed. For example, Fig. 1 includes comments such as “In this scenario, the tipping risk is very low.” These should be deleted. Check all figure captions throughout the manuscript and remove such comments.
6. Line 108: Figure S3 does not show SSPs.
7. Lines 123–124: A reference is needed regarding the use of 10-year averages.
8. General comment on captions: Statements concerning robustness must be included in the main text, not in captions. Comments, results, or references to important issues should be removed from captions and incorporated into the manuscript body.
9. Figure 1 caption: The caption is unclear and must be rewritten. Panels d–f are not equivalent to panels a–c.
10. The map representations need more explanation in the text. Why is a certain percentage used in the tipped area? Figures 1d, e, and f do not show the same rationale.
11. Line 173: The term RCP appears for the first time here, within SSP-RCP-based. It should be introduced earlier, or changed.
12. Line 186. Confirm whether Fig. 1d actually illustrates the stated point.
13. Fig2: The dashed circle in panel c is not described in the caption.
14. Lines 262-268: Text is repeated. Please correct such errors carefully.
15. Methods: The UTrack model should be placed in the context of other Lagrangian methodologies. As mentioned in the first review, many techniques exist that do not always produce the same results. While UTrack is an appropriate choice, this discussion should be explicitly included.
16. Line 468: A density plot of particle altitude distributions would be useful. If most particles are not near the surface, results could be misinterpreted.
17. Lines: 472-476: The response in the “comment to reviewers” should be incorporated here. The distance traveled by particles is important to show their movement. Figure R4, currently in the supplementary material, is valuable and should be included.
18. Line 480: Why was a 30-day period chosen? This is too long. The mean residence time of water vapor in the atmosphere is typically 10–15 days, which would be sufficient. This is particularly relevant for recycling processes, where residence times are even shorter.
19. Line 488: The statement about good correspondence needs to be integrated into the main text, not just captions. Correlation coefficients (e.g., R^2 values) can be indicated in figures, but their statistical significance must be clearly discussed in the manuscript.
20. Line 489: The absence of bias requires explanation. Please include this, as in Supplementary Material (Answer #11, Review 3).
21. Supplementary material. Supplementary Note: Validation of moisture recycling based on ERA5 reanalysis data.
 - a. What is meant by “quasi-validation”?
 - b. Why only for 5 years? Were modes of climate variability (e.g., ENSO) absent during this period?
 - c. Why was SSP2-4.5 chosen for comparison with ERA5 in the historical period? What about the other SSPs?
 - d. The sentence “UTrack by default is forced by hourly ERA5 data for 25 pressure layers at 0.25° spatial resolution instead of daily data for 8 pressure layers at $1.25^\circ \times 0.9375^\circ$ in this study. Therefore, the time step is also set lower, to 0.25 hours instead of 4 hours is unclear. Please rewrite for clarity.
 - e. Figure R10 is illustrative. please include it in the supplementary material with explanatory text.
22. Line 496; why are different periods for SSPs?
23. In the response to reviewers, the authors state that “The rationale for using SSP2-4.5 was because it aligns most with the recent trajectory of the global climate.” This important rationale must be clearly stated in the manuscript.

24. Fig S12: Where is the map for SSP5-8.5? Also, the statement “Overall, the SSP-based deforestation scenarios are very optimistic.” requires a reference.

25. Lines 593: Acronyms MAP and MCWD MF have already been introduced; no need to repeat them here.

26. Fig S8 and S9: why not for the other SSPs, as in S7a,b?

27. Fig S10: Results must be moved from the caption to the main text. In addition, the meaning of the horizontal line at 20% in Fig. S10a is not explained. Please delete “Fig. S15 shows the tipped area dependent on MAP and MCWD, given its respective moisture transport or global warming level.”

28. Line 692: Results are reported in the caption of Fig. S20 but should be in the main text. All substantive results must be incorporated into the manuscript body, not left in figure captions.

29. General comment: The tipping point was calculated using recycled precipitation, but the text should make it clearer that the moisture for precipitation in the Amazon does not come solely from the region itself. The Amazon has other sources of moisture that could modify the precipitation calculated here. Add references to Lagrangian methods for moisture sources in the Amazon.

Version 2:

Reviewer comments:

Referee #1

(Remarks on code availability)

(Remarks to the Author)

The authors have done substantial work to address my previous comments, and the manuscript is considerably improved. The framing is now more nuanced, moving away from the stronger claims of the tipping point framework. However, I remain skeptical of some of the core elements of the study.

My primary concern is that the binary representation of vegetation states — either forest or savanna/dry forest — in the model is a huge oversimplification. Given the well-documented structural and functional complexity of Amazonian ecosystems, including the continuum of forest types, transitional vegetation, and the spatial heterogeneity of climate and soil conditions, I find it difficult to imagine that such a simplification can capture the dynamics of these systems. This concern is compounded by the absence of robust model validation, lack of uncertainty for projected changes in precipitation, and unrealistic deforestation scenarios, which make it difficult to assess the reliability of the projected transitions.

That said, I recognize this as an important contribution that addresses a major scientific gap. The study demonstrates, under a set of clearly stated and strong assumptions, that combined changes in regional climate driven by deforestation and global climate change could push parts of the Amazon toward non-forest states. This is a major finding that warrants attention, provided readers interpret the results with an awareness of the model's limitations.

Referee #2

(Remarks on code availability)

(Remarks to the Author)

Thank you for asking me to look again at the responses to reviewer requests for the paper: “Pinpoint Amazon forest transition under global warming and deforestation”.

I can see that the authors have again taken the reviewers' suggestions seriously (both mine and those of Reviewer #1 and Reviewer #2), and there are no further comments on the responses. They look fine.

Climate change continues to suffer from a mismatch between (1) Earth System Models, which differ in their projections but many indicate tipping points ahead (including, potentially, for the Amazon) and (2) the theory of nonlinear dynamical systems, which characterises well the mathematics of tipping caused by a change in a bifurcation parameter.

Unfortunately, to date, very few researchers have linked (1) and (2) above, which is a real shame. By making this link, the dynamical system becomes available for other researchers to test different forcing profiles (e.g. climate change “overshoot”), better understand what the tipping point might look like, and potentially use a calibrated dynamic system to develop early warning systems.

Hence, I have always liked this manuscript because Wunderling et al. make that link through their Equation (2). That equation is very clearly a dynamical system, and the “cubic” term gives the potential for a tipping point, along with hysteresis in an overshoot scenario.

So, I have one very final small request. To ensure this manuscript is picked up by the mathematical community and recognised as linking the two disciplines (climate science and nonlinear dynamics), could the authors add a sentence or two that makes this explicit?

Somewhere near Equation (2), something like: "This nonlinear equation is a typical dynamical system that can exhibit tipping point behaviour, and where the C_{crit} , and the Summation terms can be interpreted as a time-evolving bifurcation parameter". That then gets the keywords of "nonlinear, tipping point, dynamical system and bifurcation" together.

Otherwise, this is a superb paper, and I have enjoyed assessing it. I very much hope Nature will now formally accept the manuscript.

Referee #3

(Remarks on code availability)

(Remarks to the Author)

Overall, the manuscript has improved substantially in clarity, structure, and methodological transparency. The revisions have addressed the main concerns raised previously, and the paper is now much stronger.

Only a pair of minor issues remain:

Answer #15 -> Lines 70-71: The description of UTrack as "following the trajectories of moisture through the atmosphere" is somewhat misleading. In a strict Lagrangian sense, the model does not track moisture itself as an independent physical entity. Rather, it follows the trajectories of air parcels, to which specific humidity values are assigned based on reanalysis or model output fields. The moisture content is therefore not dynamically resolved along an independent moisture trajectory, but diagnostically attributed to moving air parcels using externally provided meteorological data.

I recommend clarifying this distinction to avoid conceptual ambiguity, particularly for readers from an atmospheric dynamics background, for whom the difference between tracking moisture as a substance and advecting air parcels with assigned humidity is non-trivial.

Line 46: "Part of the Amazon's precipitation is externally sourced ...", this sentence needs a citations: e.g. Gimeno et al., 2012 and Sorí, et al., 2017

Gimeno, L., Stohl, A., Trigo, R. M., Dominguez, F., Yoshimura, K., Yu, L., Drumond, A., Durán-Quesada, A. M., & Nieto, R. (2012). Oceanic and terrestrial sources of continental precipitation. *Reviews of Geophysics*, 50(4), RG4003. <https://doi.org/10.1029/2012RG000389>

Sorí, R., Nieto, R., Drumond, A., Gimeno, L., & Vicente-Serrano, S. M. (2017). A Lagrangian perspective of the hydrological cycle in major global river basins. *Earth System Dynamics*, 8(4), 1009–1022. <https://doi.org/10.5194/esd-8-1009-2017>

Dear Reviewers, dear Editor,

We are grateful for the very substantial feedback and comments by the reviewers. The suggestions, references and advice have helped to revise and improve our manuscript. The comments of the reviewers have led to the following major changes in the revised manuscript:

1. We put major effort into the following comprehensive additional simulations
 - (i) using two alternative scenarios for adaptation capacity, either with the assumption that dry adapted species have lower safety margins than wet adapted species and the other way around.
 - (ii) using SSP5-8.5 as an additional scenario (on top of SSP2-4.5 and 3-7.0).
We find that our results remain robust and within the expectation of the reviewers and us.
2. Atmospheric moisture recycling network: We have performed substantial dedicated simulations for this study with UTrack already in the original manuscript, which we have not clearly explained in our original manuscript. We revised our text accordingly and also outlined the sensitivity of our results dependent on the underlying atmospheric moisture recycling network. In addition, we run substantial additional robustness checks comparing the atmospheric moisture recycling based on Earth System Model (NorESM2) input with reanalysis (ERA5) products with regard to precipitation recycling ratios, wind speeds and tipping risks (see new SI Note).
3. We carry out careful explanations on our methodology and limitations of our study. In particular, we discuss the relation to CMIP6 results, rainfall exclusion experiments and CO₂-fertilization effects (among others).

We have also considered all further feedback points raised by the reviewers. Please find below a detailed point-by-point response to the comments. We also attached the new version of our manuscript and supplement, and marked the changes in blue. We are grateful for this opportunity to improve our manuscript and are confident that our revised manuscript meets the high standards of Nature.

We are looking forward to your further feedback.

Sincerely yours,

Nico Wunderling, Boris Sakschewski, Johan Rockström, Bernardo Flores, Marina Hirota, and Arie Staal

Referee #1 (Remarks to the Author):

The manuscript addresses a critical issue: how changes in mean annual precipitation (MAP) and maximum climatological water deficit (MCWD) may drive Amazon forests toward a tipping point. Building on prior research on moisture transport across the Basin, the study makes a valuable contribution by quantifying potential changes in forest stability in the Amazon. The results suggest that while Amazon forests may exhibit resilience to regional warming, deforestation could significantly reduce their ability to cope with climatic changes, lowering critical thresholds beyond previous estimates. These findings are highly relevant to discussions on the Amazon's long-term stability/resilience. However, several points require clarification and further development:

We thank the reviewer for their encouraging feedback on the relevance of our article, in particular regarding regional warming, deforestation and both pressures that act on the Amazon forests simultaneously. We are also very grateful for the helpful feedback that we respond to in a point to point fashion below.

1. The manuscript assumes that the Amazon exhibits two alternative stable states, with crossing a threshold leading to a critical transition within decades. While some models support this idea, observational evidence remains limited.

Answer #1.1: We agree with the reviewer that we assume nonlinearities in the Amazon forests on a local scale (and not per se on a system-wide scale; this is in line with a recent review article by Brando et al., 2025, ARER; Flores et al., 2024), which can spread via the lack of atmospheric moisture transport if certain thresholds of global warming and deforestation are transgressed. It is important to note, however, that we do not compute timescales of *how fast* a critical transition would be once critical thresholds are surpassed (while decadal timescales seem realistic as rainfall exclusion experiments indicate, see our answer #1.2).

Regarding observational evidence, several observational studies provide compelling evidence that the Amazon rainforest may be approaching local to regional tipping points (see below) and possibly be trapped in a more open state (Veldman & Putz, 2011, Biol.-Conservation; Silvério et al., 2013, Phil. Trans. Roy. Soc. B; Silvério et al., 2022, ERL; Flores & Holmgren, 2021, Ecosystems):

Specifically, at the local scale, *Staver et al. (2011, Science)* & *Hirota et al. (2011, Science)* identified bistability in vegetation cover using satellite data, showing that fire regimes can maintain either forest or savanna under similar climate conditions. The existence of forest and non-forest under the same climate without land use influence is indirect evidence of such bistabilities (Hirota et al., 2011, Science). *Flores & Holmgren (2021, Ecosystems)* found that repeated fires led to the expansion of white-sand savanna vegetation in Amazon floodplains, indicating ecological transformation after disturbance. *Nepstad et al. (2007, Ecology)* and *Brando et al. (2014, PNAS)* provided field-based

observations demonstrating how drought and fire increase tree mortality and can trigger forest collapse. At the sub-regional scale, *Fu et al. (2013, PNAS)* documented a significant lengthening of the dry season in the southern Amazon, suggesting weakening forest–rainfall feedbacks, while *Gatti et al. (2021, Nature)* observed that the southeastern Amazon has shifted from a carbon sink to a net carbon source, based on atmospheric CO₂ measurements. At the regional scale, *Boulton et al. (2023, Nat. Clim Change)* used satellite vegetation data to show widespread loss of forest resilience across over 75% of the basin. Also, shifts in vegetation composition—from wet- to dry-affiliated species—have been documented in various biomes worldwide, including the Amazon following drought-related mortality (*Esquivel-Muelbert et al., 2020, Nature Communications*; *Battlori et al., 2020, PNAS*). In addition, paleo evidence suggests that peripheral parts of the Amazon shifted to savanna in drier periods with more fires (*Flores et al., 2024, Nature*).

Further, *Flores et al (2024, Nature)* have estimated critical thresholds with Earth Observation efforts. In our study, we have used their values as constraints in a sensitivity experiment and results are robust to the main manuscript's results (see SI Fig. S7c, d).

Taken together, we believe that the evidence from Earth observation efforts point toward the possibility of localized tipping points in the Amazon forest systems as we hypothesize them in our study. We extend our manuscript with the above reasoning (see **II 83-88**).

References:

1. Brando, P.M., Barlow, J., Macedo, M.N., Silvério, D.V., Ferreira, J.N., Maracahipes, L., Anderson, L., Morton, D.C., Alencar, A., Paolucci, L.N. and Jacobs, S., 2025. Tipping Points of Amazonian Forests: Beyond Myths and Toward Solutions. *Annual Review of Environment and Resources*, 50.
2. Veldman, J.W. and Putz, F.E., 2011. Grass-dominated vegetation, not species-diverse natural savanna, replaces degraded tropical forests on the southern edge of the Amazon Basin. *Biological Conservation*, 144(5), pp.1419-1429.
3. Silvério, D.V., Brando, P.M., Balch, J.K., Putz, F.E., Nepstad, D.C., Oliveira-Santos, C. and Bustamante, M.M., 2013. Testing the Amazon savannization hypothesis: fire effects on invasion of a neotropical forest by native cerrado and exotic pasture grasses. *Philosophical Transactions of the Royal Society B: Biological Sciences*, 368(1619), p.20120427.
4. Silvério, D.V., Oliveira, R.S., Flores, B.M., Brando, P.M., Almada, H.K., Furtado, M.T., Moreira, F.G., Heckenberger, M., Ono, K.Y. and Macedo, M.N., 2022. Intensification of fire regimes and forest loss in the Território Indígena do Xingu. *Environmental Research Letters*, 17(4), p.045012.
5. Staver, A.C., Archibald, S. and Levin, S.A., 2011. The global extent and determinants of savanna and forest as alternative biome states. *science*, 334(6053), pp.230-232.

6. Hirota, M., Holmgren, M., Van Nes, E.H. and Scheffer, M., 2011. Global resilience of tropical forest and savanna to critical transitions. *Science*, 334(6053), pp.232-235.
7. Flores, B.M. and Holmgren, M., 2021. White-sand savannas expand at the core of the Amazon after forest wildfires. *Ecosystems*, 24(7), pp.1624-1637.
8. Wang, X., Edwards, R.L., Auler, A.S., Cheng, H., Kong, X., Wang, Y., Cruz, F.W., Dorale, J.A. and Chiang, H.W., 2017. Hydroclimate changes across the Amazon lowlands over the past 45,000 years. *Nature*, 541(7636), pp.204-207.
9. Nepstad, D.C., Tohver, I.M., Ray, D., Moutinho, P. and Cardinot, G., 2007. Mortality of large trees and lianas following experimental drought in an Amazon forest. *Ecology*, 88(9), pp.2259-2269.
10. Brando, P.M., Balch, J.K., Nepstad, D.C., Morton, D.C., Putz, F.E., Coe, M.T., Silvério, D., Macedo, M.N., Davidson, E.A., Nóbrega, C.C. and Alencar, A., 2014. Abrupt increases in Amazonian tree mortality due to drought–fire interactions. *Proceedings of the National Academy of Sciences*, 111(17), pp.6347-6352.
11. Fu, R., Yin, L., Li, W., Arias, P.A., Dickinson, R.E., Huang, L., Chakraborty, S., Fernandes, K., Liebmann, B., Fisher, R. and Myneni, R.B., 2013. Increased dry-season length over southern Amazonia in recent decades and its implication for future climate projection. *Proceedings of the National Academy of Sciences*, 110(45), pp.18110-18115.
12. Gatti, L.V., Basso, L.S., Miller, J.B., Gloor, M., Gatti Domingues, L., Cassol, H.L., Tejada, G., Aragão, L.E., Nobre, C., Peters, W. and Marani, L., 2021. Amazonia as a carbon source linked to deforestation and climate change. *Nature*, 595(7867), pp.388-393.
13. Esquivel-Muelbert, A., Phillips, O.L., Brienen, R.J., Fauset, S., Sullivan, M.J., Baker, T.R., Chao, K.J., Feldpausch, T.R., Gloor, E., Higuchi, N. and Houwing-Duistermaat, J., 2020. Tree mode of death and mortality risk factors across Amazon forests. *Nature communications*, 11(1), p.5515.
14. Battlori, E., Lloret, F., Aakala, T., Anderegg, W.R., Aynekulu, E., Bendixsen, D.P., Bentouati, A., Bigler, C., Burk, C.J., Camarero, J.J. and Colangelo, M., 2020. Forest and woodland replacement patterns following drought-related mortality. *Proceedings of the National Academy of Sciences*, 117(47), pp.29720-29729.
15. Flores, B.M., Montoya, E., Sakschewski, B., Nascimento, N., Staal, A., Betts, R.A., Levis, C., Lapola, D.M., Esquivel-Muelbert, A., Jakovac, C. and Nobre, C.A., 2024. Critical transitions in the Amazon forest system. *Nature*, 626(7999), pp.555-564.
16. Boulton, C.A., Lenton, T.M. and Boers, N., 2022. Pronounced loss of Amazon rainforest resilience since the early 2000s. *Nature Climate Change*, 12(3), pp.271-278.
17. Parry, I.M., Ritchie, P.D. and Cox, P.M., 2022. Evidence of localised Amazon rainforest dieback in CMIP6 models. *Earth System Dynamics*, 13(4), pp.1667-1675.

For example, drought experiments suggest biomass reductions of 20-40% over a decade with a 50% precipitation decline.

Answer #1.2: We thank the reviewer for pointing out drought experiments carried out over the last decades. The recent work by Sanchez-Martinez et al. (2025, Nature Ecology & Evolution) essentially finds a biomass decline of around 35% in a more than two decades long rainfall exclusion experiment (approximately halving available rainfall). Their Figure 1a (see Fig. R1 below) very nicely represents a clear state transition from a higher biomass to a lower biomass state with an about 10 year lead time (depending on rooting depths and groundwater availability), and a remarkably quick transition period of about 2-3 years. This means that the overall vegetation shift happened over a period of ~15 years.

Fig. R1: From Sanchez-Martinez et al. (2025): Estimated aboveground biomass for a rainfall exclusion experiment of 50% between 2002-2023 (green: control; dark orange: experiment; yellow shading: transition phase; green shading: stabilization phase at new and lower equilibrium).

Other rainfall exclusion experiments also show that hydraulic limitations cause increased tree mortality (Rowland et al., 2015, Nature; Meir et al., 2015, Bioscience), in particular in larger trees, even though mortality starts to increase not earlier than 3 years after the start of a rainfall exclusion experiment (sometimes only 5-7 years after the start of the experiment). The reason is likely the deep roots of the trees being adapted to occasional droughts but once the soil moisture content has halved, their physiological capabilities are transgressed (Meir et al., 2015, Bioscience).

In future work and given increased data availability from rainfall exclusion experiments on multiple plots and levels, it could be possible to use those experimentally determined biomass reductions to tune our Eq. (2), directly combining observational evidence with simplified nonlinear dynamical equation approaches.

However, it is important to note that rainfall exclusion experiments per-se can only test the effects of a very local step change response to precipitation reduction and do not test for self-reinforcing feedbacks other than the following: local canopy openness→drying of understorey→canopy openness. Also, rainfall exclusion experiments do not quantify effects on moisture recycling. Instead, they only test very local responses to reduced rainfall, such as self-reinforcing feedbacks like: canopy opening → understorey drying → further canopy opening. Because of this, they cannot account for regional or large-scale evapotranspiration (forest–rainfall) feedbacks and fire feedbacks that may drive more abrupt shifts. We have expanded our manuscript with regard to our above descriptions (see II 88-93).

References:

1. Sanchez-Martinez, P., Martius, L.R., Bittencourt, P., Silva, M., Binks, O., Coughlin, I., Negrão-Rodrigues, V., Athaydes Silva, J., Da Costa, A.C.L., Selman, R. and Rifai, S., 2025. Amazon rainforest adjusts to long-term experimental drought. *Nature Ecology & Evolution*, pp.1-10.
2. Meir, P., Wood, T.E., Galbraith, D.R., Brando, P.M., Da Costa, A.C., Rowland, L. and Ferreira, L.V., 2015. Threshold responses to soil moisture deficit by trees and soil in tropical rainforests: insights from field experiments. *BioScience*, 65(9), pp.882-892.
3. Rowland, L., da Costa, A.C.L., Galbraith, D.R., Oliveira, R.S., Binks, O.J., Oliveira, A.A., Pullen, A.M., Doughty, C.E., Metcalfe, D.B., Vasconcelos, S.S. and Ferreira, L.V., 2015. Death from drought in tropical forests is triggered by hydraulics not carbon starvation. *Nature*, 528(7580), pp.119-122.

Furthermore, without drought-fire interactions, there is little evidence of a sharp, irreversible shift. Given the challenges in validating the model, the results should be presented with greater nuance, contextualized based on observational studies, experimental approaches, and other types of models (DGVMs).

Answer #1.3: We agree with the reviewer that our approach does not explicitly simulate drought–fire feedbacks, given its simplicity. Instead, we represent fire effects indirectly by assuming local-scale tipping. Our framework acknowledges that forests exist within local climatic envelopes embedded in a larger moisture recycling network. By focusing only on water recycling feedbacks, our estimates are therefore conservative. Including drought–fire feedbacks would likely increase the extent of tipping and forest loss (see e.g. Brando et al., 2020, Science Advances). Our changes in the manuscript can be found in II 111-114.

It is important to note that our model represents only equilibrium responses and does not include an inherent timescale, although a decadal timescale appears reasonable (see answer #1.2). We also intentionally do not simulate recovery from the alternative state back to full forest cover, as our focus in this study is on the current risk of transition from forest to alternative states.

Reference:

Brando, P.M., Soares-Filho, B., Rodrigues, L., Assunção, A., Morton, D., Tuchsneider, D., Fernandes, E.C.M., Macedo, M.N., Oliveira, U. and Coe, M.T., 2020. The gathering firestorm in southern Amazonia. *Science Advances*, 6(2), p.eaay1632.

2. The mechanisms linking deforestation and MAP/MCWD needs further clarification. If the linkage is primarily through deforestation-related reductions in evapotranspiration (ET), it is important to note that most CMIP6 models (but not all) already estimate some levels of deforestation effects on MAP and MCWD, which tend to be relatively small (<10%) (see see

comments below and Li et al., Nature Communications). It is unclear how such reductions in deforestation could lower the temperature threshold for forest collapse from 3.7-4.0°C to 1.5-1.9°C. A more detailed explanation of this mechanism is necessary.

Answer #2.1: This is indeed an important point that the reviewer raises, which we discuss in the new version of the manuscript (see **II 662-671**) and in the following paragraph:

Some CMIP6 models show a very small response to deforestation experiments, which are on the order of 5-10% rainfall decrease after drastic deforestation of global forest systems (Li et al., 2022, Nature Communications) due to the very (and maybe too) strong role oceanic evaporation plays for precipitation over land. We put this result into context with review articles, observational evidence and results of global hydrological models, which strongly indicate that these effects simulated by CMIP6 models are too weak.

Based on experimental data from the BrasilFlux database, substantial deforestation has been found to decrease regional rainfall by up to 40% (Spracklen et al., 2018, ARER; Swann et al., 2015, Agricultural and Forest Meteorology). In particular, regions with high moisture recycling ratios (and the Amazon rainforest is one of the regions with the highest ratios globally: 20-70% dependent on the region) are subject to such considerable rainfall decreases (Spracklen et al., 2018, ARER). Similar reductions of rainfall have been summarized in Sanchez-Martinez et al. (2025, Nature Ecology & Evolution) following hydrological model efforts under deforestation in earlier literature (Staal et al., 2018, Nature Climate Change; Ruiz-Vásquez et al., 2020, Climate Dynamics). Further, this is in line with assessments of deforestation induced tipping points in the Amazon rainforest (Bochow & Boers, 2022, Science Advances) where deforestation was implemented as a 40% decrease of ET (based on Dominguez et al., 2022, JGRA; van Randow et al., 2004, TAC). This is in line from a particular observational study site in the central Amazon showing that up to 60% of the rainfall is returned to the atmosphere (Kunert et al., 2017, Agr. and For. Meteorology). Lastly, on local to regional scales, forest transpiration has been found responsible for a 2-3 months earlier onset of the wet season as compared to a start of the wet season initiated by the Intertropical Convergence Zone (ITCZ) only (Wright et al., 2017, PNAS). This highlights the critical role of primary forests versus largely deforested areas which could not provide this hydrologic self-service, and may represent a critical extension of the dry season beyond adaptive capacities.

Taken together, our assessment is that the effect of deforestation on rainfall in some CMIP6 models is underestimated.

In our study, we use the global hydrological model PCR-GLOBWB version 2 (Bosmans et al., 2017, HESS) to perform robustness checks regarding the residual evapotranspiration after tipping. We use PCR-GLOBWB experiments at 0.5 x 0.5 degree spatial and monthly temporal resolution for 1981-2010 of evapotranspiration under different land cover scenarios: tall vegetation (forest), short vegetation (grassland), cropland and pasture. We regridded these data to the resolution of the NorESM2 dataset. We use the PCR-GLOBWB

data to determine for each grid cell the difference in evapotranspiration between forest and the highest-evaporating alternative land cover. The exact approach was previously done by Tuinenburg et al. (2022, *Environmental Research Letters*) and the same model was used to estimate transpiration flows in the Amazon by among others Staal et al. (2018, *Nature Climate Change*) and Staal et al. (2020, *Nature Communications*).

In addition, we ran a robustness check keeping the evapotranspiration at values of alternative landcover after removing/tipping the primary forest based on values by Tuinenburg et al. (2022, *ERL*). The results are robust (see **SI Fig. S15** compared to Fig. 1) and methods section *Robustness checks*.

References:

1. Bochow, N. and Boers, N., 2023. The South American monsoon approaches a critical transition in response to deforestation. *Science Advances*, 9(40), p.eadd9973.
2. Bosmans, J.H., van Beek, L.P., Sutanudjaja, E.H. and Bierkens, M.F., 2017. Hydrological impacts of global land cover change and human water use. *Hydrology and Earth System Sciences*, 21(11), pp.5603-5626.
3. Dominguez, F., Eiras-Barca, J., Yang, Z., Bock, D., Nieto, R. and Gimeno, L., 2022. Amazonian moisture recycling revisited using WRF with water vapor tracers. *Journal of Geophysical Research: Atmospheres*, 127(4), p.e2021JD035259.
4. Kunert, N., Aparecido, L.M.T., Wolff, S., Higuchi, N., dos Santos, J., de Araujo, A.C. and Trumbore, S., 2017. A revised hydrological model for the Central Amazon: The importance of emergent canopy trees in the forest water budget. *Agricultural and Forest Meteorology*, 239, pp.47-57.
5. Li, Y., Brando, P.M., Morton, D.C., Lawrence, D.M., Yang, H. and Randerson, J.T., 2022. Deforestation-induced climate change reduces carbon storage in remaining tropical forests. *Nature Communications*, 13(1), p.1964.
6. Ruiz-Vásquez, M., Arias, P.A., Martínez, J.A. and Espinoza, J.C., 2020. Effects of Amazon basin deforestation on regional atmospheric circulation and water vapor transport towards tropical South America. *Climate Dynamics*, 54, pp.4169-4189.
7. Spracklen, D.V., Baker, J.C.A., Garcia-Carreras, L. and Marsham, J.H., 2018. The effects of tropical vegetation on rainfall. *Annual Review of Environment and Resources*, 43(1), pp.193-218.
8. Staal, A., Tuinenburg, O.A., Bosmans, J.H., Holmgren, M., van Nes, E.H., Scheffer, M., Zemp, D.C. and Dekker, S.C., 2018. Forest-rainfall cascades buffer against drought across the Amazon. *Nature Climate Change*, 8(6), pp.539-543.
9. Staal, A., Fetzer, I., Wang-Erlandsson, L., Bosmans, J.H., Dekker, S.C., van Nes, E.H., Rockström, J. and Tuinenburg, O.A., 2020. Hysteresis of tropical forests in the 21st century. *Nature communications*, 11(1), p.4978.
10. Swann, A.L., Longo, M., Knox, R.G., Lee, E. and Moorcroft, P.R., 2015. Future deforestation in the Amazon and consequences for South American climate. *Agricultural and Forest Meteorology*, 214, pp.12-24.

11. Tuinenburg, O.A., Bosmans, J.H. and Staal, A., 2022. The global potential of forest restoration for drought mitigation. *Environmental Research Letters*, 17(3), p.034045.
12. Von Randow, C., Manzi, A.O., Kruijt, B., De Oliveira, P.J., Zanchi, F.B., Silva, R.D., Hodnett, M.G., Gash, J.H., Elbers, J.A., Waterloo, M.J. and Cardoso, F.L., 2004. Comparative measurements and seasonal variations in energy and carbon exchange over forest and pasture in South West Amazonia. *Theoretical and Applied Climatology*, 78, pp.5-26.
13. Wright, J.S., Fu, R., Worden, J.R., Chakraborty, S., Clinton, N.E., Risi, C., Sun, Y. and Yin, L., 2017. Rainforest-initiated wet season onset over the southern Amazon. *Proceedings of the National Academy of Sciences*, 114(32), pp.8481-8486.

Note also that if the CMIP6 model already incorporate deforestation effects, raising the concerns about potential double counting deforestation effects climate -- if additional effects on climate are applied post hoc.

Answer #2.2: We agree with the reviewer that the SSP scenarios include land-use change effects (see our SI Fig. S12), however, the effect of double accounting is very small. The reason is that the SSP-based deforestation experiments are very conservative and essentially keep deforested areas constant throughout the 21st century (see SI Fig. S12). Levels of deforestation are constant around 20%. Multiplied with a reduction of rainfall of 5-10% (Li et al., 2022, Nat. Comms.), this would lead to rainfall reduction of 1-2% that we disregard in our study. So double accounting can almost completely be excluded.

3. While deforestation reduces ET, particularly during transition months, the land uses replacing forests (e.g., pasturelands) maintain low ET levels (~1 mm). The discussion should also highlight the dominant role of transpiration over evaporation in these processes.

Answer #3: We agree with the reviewer to discuss the dominant role of transpiration over evaporation in tropical forests as compared to other drier natural and managed land types (D'Acunha et al. 2024, Agricultural and Forest Meteorology; based on FluxTower measurements). In our new manuscript, we argue along the following lines of arguments (see II 49-55):

Tropical forests have a dense vegetation cover with multiple canopy layers shading the ground. The area for transpiration is large because the number of leaves per m² is large (large leaf area index) as compared to (managed) Cerrado and Pantanal biomes. Therefore, transpiration remains high across the year and also in the dry season (as compared to other vegetation types). Temperatures are usually warm and humid, allowing for constant transpiration. Stomatal closure in the dry season is thus limited. Managed biomes (cropland and pasture) show decreased evapotranspiration as compared to native biomes highlighting the importance of the natural Amazon forests for the hydrological health of the entire region.

Also, it has been shown that the emergent canopy trees are responsible for the majority of the transpired water (71%, Kunert et al., 2017, *Agr. and For. Meteorology*), recycling 26% of the rainfall back to the atmosphere. Those are the trees that are most vulnerable from increased drought frequencies and intensities (see also our answer #1.2; Sanchez-Martinez et al., 2025, *Ecol Evol*). Overall, the entire vegetation recycles 36% of the rainfall back to the atmosphere, while 22% are intercepted rainwater (Kunert et al., 2017, *Agr. and For. Meteorology*) - therefore, transpiration dominates over evaporation. In addition, the wet season onset is critically determined by the rainforest's transpiration during the late dry season in the southern Amazon rainforest (Wright et al., 2017, *PNAS*). This means that deforestation and land-use change have the potential to delay the onset of the wet season (Wright et al., 2017, *PNAS*), while dry season precipitation is also reduced (Liu et al., 2025, *AGU Advances*).

Reference:

- D'Acunha, B., Dalmagro, H.J., de Arruda, P.Z., Biudes, M.S., Lathuillière, M.J., Uribe, M., Couto, E.G., Brando, P.M., Vourlitis, G. and Johnson, M.S., 2024. Changes in evapotranspiration, transpiration and evaporation across natural and managed landscapes in the Amazon, Cerrado and Pantanal biomes. *Agricultural and Forest Meteorology*, 346, p.109875.
- Kunert, N., Aparecido, L.M.T., Wolff, S., Higuchi, N., dos Santos, J., de Araujo, A.C. and Trumbore, S., 2017. A revised hydrological model for the Central Amazon: The importance of emergent canopy trees in the forest water budget. *Agricultural and Forest Meteorology*, 239, pp.47-57.
- Liu, Y., Spracklen, D.V., Parker, D.J., Holden, J., Ge, J. and Guo, W., 2025. Recent forest loss in the Brazilian Amazon causes substantial reductions in dry season precipitation. *AGU Advances*, 6(2), p.e2025AV001670.
- Sanchez-Martinez, P., Martius, L.R., Bittencourt, P., Silva, M., Binks, O., Coughlin, I., Negrão-Rodrigues, V., Athaydes Silva, J., Da Costa, A.C.L., Selman, R. and Rifai, S., 2025. Amazon rainforest adjusts to long-term experimental drought. *Nature Ecology & Evolution*, pp.1-10.
- Wright, J.S., Fu, R., Worden, J.R., Chakraborty, S., Clinton, N.E., Risi, C., Sun, Y. and Yin, L., 2017. Rainforest-initiated wet season onset over the southern Amazon. *Proceedings of the National Academy of Sciences*, 114(32), pp.8481-8486.

4. The manuscript presents adaptation capacity as a key factor in resilience, but it is unclear whether the assumed adaptations align with reality. While plants near physiological limits may develop traits to tolerate drier conditions, they often have smaller safety margins despite higher resistance. Treating adaptation capacity as a scenario (among others representing different levels of adaptation) rather than a quantitative metric may better reflect its uncertainty. Additionally, exploring how variability in vegetation adaptation influences tipping point probabilities would provide a more nuanced perspective. The assumption that drier forests are inherently more adapted to climate change warrants further scrutiny.

Answer #4: We thank the reviewer for their clarifying thoughts on our adaptation mechanism and we agree this point requires further scrutiny. Following the reviewer's suggestion and also our own belief to increase reliability in our adaptation capacity assumptions, we develop the following additional scenarios of adaptation capacities:

Scenario A: We follow the reviewer with their assessment that drier regions in the rainforest may operate closer to their physiological limits and may have smaller safety margins. Therefore, we ran an experiment, where we scale adaptation capacities with regional rainfall levels that the forest receives. To compare these results with the results of our original simulations, we chose 1.25 standard deviations for the wettest regions (=higher resilience to drier conditions) and 0.75 standard deviations for the driest regions (=lower resilience to drier conditions).

Fig. R2: Effects on tipping risks for adaptation scenario A. (a) Tipping risk map for the decade 2090-2099 in a SSP3-7.0 scenario, (b) Tipping response throughout the 21st century (time series) following an SSP3-7.0 scenario. Compare with Fig. 1b of the main manuscript.

Scenario B: We check the effects of the opposite assumptions as in Scenario A: Wetter regions have lower safety margins (0.75 standard deviations as adaptation capacity) and drier regions have higher safety margins (1.25 standard deviations as adaptation capacity).

Fig. R3: Effects on tipping risks for adaptation scenario B. (a) Tipping risk map for the decade 2090-2099 in a SSP3-7.0 scenario, (b) Tipping response throughout the 21st century (time series) following an SSP3-7.0 scenario. Compare with Fig. 1b of the main manuscript.

The results of scenarios A and B (see Figs. R2 & R3) are robust against our original analysis (see Fig. 1b of the main manuscript). We find, however, that Scenario A leads to higher tipping risks than our original analyses (see Fig. 1b) because the dry regions are operating closer to their physiological limits and can therefore not sustain even further drying (as the reviewer suspected). Scenario B is a sensitivity check with the more hypothetical assumption that wetter regions operate closer to their physiological limits. Still, results are robust against our original analysis (see Fig. R3 and compare to Fig. 1b).

Therefore, our assumption in the main manuscript of drier regions being resilient to further drying is a conservative assumption and may overestimate the real resilience of the rainforest as the reviewer indicated above. Based on the results by Choat et al. (2012, Nature) “Safety margins are largely independent of mean annual precipitation, showing that there is global convergence in the vulnerability of forests to drought, with all forest biomes equally vulnerable to hydraulic failure regardless of their current rainfall environment.”, we keep our original notion of adaptation capacities in the main manuscript. We add these additional explanations and our new robustness check (see SI Fig. S20), and have adapted our manuscript accordingly, see caption of Fig. 1, II 238-244, II 558-562 and II 684-693.

5. The manuscript assumes that once forests reach a tipping point, they remain in a low-ET state. However, in reality, forest recovery often occurs in the Amazon if seed dispersal and climatic conditions permit. For example, 19% of cleared areas experience regrowth, although most are later cleared again. While climate change may slow recovery rates, ET is typically one of the first ecosystem functions to return. The manuscript should provide a stronger justification for why forests would remain in a permanently degraded state rather than exhibit some level of recovery. Again, the non-ET recovery state could be treated as a scenario rather than a representation of the current reality in the region.

Answer #5: The reviewer is correct that on some of the cleared area secondary vegetation is growing, in numbers this is 19% of the deforested area (Nunes et al., 2020, ERL). From these 19%, only 10-20% grow a secondary vegetation that is not re-cleared again (Nunes et al., 2020, ERL). According to these numbers, this would mean that $19\% \times 10\% = 1.9\%$ up to $19\% \times 20\% = 3.8\%$ of the cleared area is returning permanently to secondary vegetation and may thus return to a high ET state in the decades after clearing/tipping. In addition, these 1.9-3.8% of the cleared/tipped area have been suggested to not return to a high ET state until the first ~10 years of recovery (Brando et al., 2019, GCB). Even further, of the remaining old-growth forests, at least 38% have already been degraded due to land use disturbances and repeated extreme droughts. The consequences for moisture recycling remain uncertain (Flores et al., 2024).

In summary this means that most of the forest that is lost once has significant likelihood to stay in a low ET state over decades and may therefore contribute to further damages downwind. In addition, as the reviewer mentions, recovery rates may slow down even further in response to climate change. However, we agree that after a longer time (likely

on the timescales of decades or even beyond), the cleared/tipped Amazon forest regions may return to a higher ET state.

We cover these sensitivities in our robustness analyses indirectly in this study through our robustness check in Fig. S15 that uses the ET of highest-evaporating secondary vegetation (whichever this is: grass, pasture or crop; also see Answer 2.1). In addition, we checked the robustness of our results with different types of secondary vegetation with quite different remaining values of ET (ranging from ET values similar to pasture, soy up to ET values of dry forests) in an older similar study (Wunderling et al., 2022, PNAS; see supplementary Fig. S10 and methods section *Data*).

Furthermore, the other way around, a beneficial way to increase ET through reforestation, more accurately through *targeted rainfall enhancements* (Staal et al., 2024, GCB), could be a beneficial way forward. In principle this study can be turned around and identify the regions with highest rainfall enhancement effectiveness, which is beyond the scope of this work but much along the lines of our thought.

Therefore, we expand the discussion and also our limitations section based on the valid comments by the reviewer in the **II 678-683** of our manuscript.

References:

1. Nunes, S., Oliveira, L., Siqueira, J., Morton, D.C. and Souza, C.M., 2020. Unmasking secondary vegetation dynamics in the Brazilian Amazon. *Environmental Research Letters*, 15(3), p.034057.
2. Staal, A., Theeuwen, J.J., Wang-Erlandsson, L., Wunderling, N. and Dekker, S.C., 2024. Targeted rainfall enhancement as an objective of forestation. *Global Change Biology*, 30(1), p.e17096.
3. Wunderling, N., Staal, A., Sakschewski, B., Hirota, M., Tuinenburg, O.A., Donges, J.F., Barbosa, H.M. and Winkelmann, R., 2022. Recurrent droughts increase risk of cascading tipping events by outpacing adaptive capacities in the Amazon rainforest. *Proceedings of the National Academy of Sciences*, 119(32), p.e2120777119.
4. Flores, B.M., Montoya, E., Sakschewski, B., Nascimento, N., Staal, A., Betts, R.A., Levis, C., Lapola, D.M., Esquivel-Muelbert, A., Jakovac, C. and Nobre, C.A., 2024. Critical transitions in the Amazon forest system. *Nature*, 626(7999), pp.555-564.
5. Brando, P.M., Silvério, D., Maracahipes-Santos, L., Oliveira-Santos, C., Levick, S.R., Coe, M.T., Migliavacca, M., Balch, J.K., Macedo, M.N., Nepstad, D.C. and Maracahipes, L., 2019. Prolonged tropical forest degradation due to compounding disturbances: Implications for CO₂ and H₂O fluxes. *Global Change Biology*, 25(9), pp.2855-2868.
6. The deforestation scenarios used in the study appear outdated and may not align with current trends. Given that the dataset's author has cautioned against overinterpreting these projections,

a brief discussion of their limitations should be included. Nonetheless, incorporating these scenarios is still preferable to omitting deforestation effects entirely.

Answer #6: We thank the reviewer for this valuable comment and we agree that the deforestation scenario has already been designed a while ago but it is still one of the very few datasets that project deforestation until mid century.

In addition, there are land-use change scenarios coming along the SSP scenarios (see our SI Fig. S12). Unfortunately, they are very conservative and keep deforestation levels around 20% throughout the entire 21st century with deforestation focus areas in the far west of the Amazon basin. We believe that these scenarios are very optimistic and, more importantly, do not allow us to systematically quantify critical deforestation values at which tipping points in the Amazon forests may be transgressed. Therefore, we are happy that the reviewer agrees with us that incorporating deforestation effects is better than leaving them out. We have sharpened our argumentation in the manuscript on why we still think this scenario is useful (see II 151-153 and II 528-531), agreeing with the reviewer. This is the case not least since its deforestation pattern goes along major road and infrastructure projects and is superior to random or strictly directed deforestation pathways.

7. Although CO₂ fertilization is not the primary focus of this study, it is a key driver in CMIP6 climate projections and significantly affects both regional precipitation and temperature patterns over the Amazon Basin. Some estimates suggest that CO₂ fertilization contributes to up to 40% of regional rainfall changes, which in turn influences MAP and MCWD. Given that MAP and MCWD are critical factors in determining biomass storage and forest resilience in the proposed model, the sensitivity of vegetation to CO₂ could introduce a potential bias in our model interpretations and should be discussed in the manuscript.

Answer #7: We agree with the reviewer that we need a broader discussion of the role of CO₂ fertilization. For instance, Sampaio et al. (2021, Biogeosciences) found that CO₂ fertilization may have the same effect on rainfall decrease as deforestation using a dynamic vegetation model (CPTec BAM). Further, in a recent study evaluating eight CMIP6 models (Li et al., 2023, Nature Water), the authors find that a 13% precipitation decline in SSP3-7.0. In this scenario, 44% of the precipitation decrease is attributable to plant physiology (CO₂ fertilization: 35% attribution) and deforestation (9% attribution). These results highlight the role of both drivers (plant physiology under increased CO₂ and deforestation) for the hydrology of the Amazon forests. Overall, the effect of CO₂-fertilization is still very uncertain, depending on (1) how elevated CO₂ will affect trade-offs between growth and mortality, and hence forest carbon storage, (2) tree water-use efficiency may lead to reduced evapotranspiration and moisture flow, thus forest resilience, and (3) elevated CO₂ may not even make much difference due to the low soil fertility of most forests.

We added a discussion on this in our main text in II 238-244.

References:

1. Li, Y., Baker, J.C., Brando, P.M., Hoffman, F.M., Lawrence, D.M., Morton, D.C., Swann, A.L., Uribe, M.D.R. and Randerson, J.T., 2023. Future increases in Amazonia water stress from CO₂ physiology and deforestation. *Nature Water*, 1(9), pp.769-777.
2. Sampaio, G., Shimizu, M.H., Guimarães-Júnior, C.A., Alexandre, F., Guatura, M., Cardoso, M., Domingues, T.F., Rammig, A., von Randow, C., Rezende, L.F. and Lapola, D.M., 2021. CO₂ physiological effect can cause rainfall decrease as strong as large-scale deforestation in the Amazon. *Biogeosciences*, 18(8), pp.2511-2525.

Referee #2 (Remarks to the Author):

Thank you for inviting me to review the paper: “Pinpointing Amazon Forest Tipping in Global Warming and Deforestation Pathways” by Wunderling et al.

The topic is, of course, of particular importance, and the risk of Amazon “dieback” remains one of the most frequently cited potential impacts of climate change. However, concurrently, deforestation also risks impacting Amazon loss, with its effects extending well beyond the specific areas of land use, due to adjusting terrestrial rainfall recycling.

To date, there is considerable debate regarding which factor poses a greater threat to resilience – climate change or deforestation. The main novelty of this manuscript lies in its simultaneous consideration of both drivers (climate change and deforestation), enabling a combined weighted assessment of the likely timing of rainforest loss, contingent on changes to either factor.

The immediate impression is of a high-quality manuscript; it fits well with the Nature format, and the diagrams are informative. I particularly appreciate the mapping onto an underlying dynamical system, which facilitates a deeper understanding of the system. This is a notably novel feature, as it enables the manuscript to connect with the theory of nonlinear dynamical systems. Dynamical systems form the foundation of the “tipping point” theory developed in the mathematical sciences, yet very few have rigorously mapped environmental issues onto such equations. Indeed, the authors might consider adding a line that highlights this advancement, as it provides methodological pointers to researchers attempting to characterise tipping points in other areas of the Earth system.

Answer #1: We thank the reviewer for their very positive impression of our manuscript.

My main criticism of the paper is that it should be much clearer about where assumptions are made and where uncertainties remain. I would not want this to be used as a reason for the paper's rejection by Nature, because the messages of the manuscript are important and have not been addressed in such a comprehensive manner by existing papers. At the same time, Nature is a high-profile journal, and reviewers are frequently asked to ensure that all uncertainties are considered. When presented well, emphasising remaining caveats actually enhances a paper, rather than weakening it. Caveats do not undermine the analysis, which often remains valid in light of future data or research; it is simply that the full precision is not yet known.

Answer #2: We agree with the reviewer and describe remaining limitations and caveats in more detail, also in order to outline possible future research directions, in particular regarding the following points the reviewer outlines.

Some of the uncertainties are as follows:

(1) I am willing to accept that the only ESM capable of providing boundary conditions to the atmospheric transport model is NorESM2. However, one possible additional “robustness” check is to compare the atmospheric fields, at least for the contemporary period, against one of the reanalysis products. Is that possible and sensible?

Answer #3: Thank you for this valuable comment and indeed, NorESM2 is currently the only Earth System Model that allows for dedicated runs to construct the moisture recycling network. Therefore, the atmospheric moisture recycling networks can only be constructed using NorESM2 precipitation and evaporation values fed into UTrack (forward moisture tracking model).

We explored the following and renormalized ERA5 data to NorESM precipitation on an Amazon system wide level, keeping local fluctuations different in the adaptation period (1950-2014). This still results in locally different fluctuation levels that again depend on ERA5 rainfall levels being too wet. This suggests that internally consistent data sources are important. Therefore, we need to keep NorESM2 as our consistent data source throughout this study.

However, in an earlier study, we explored Amazon tipping risks in case droughts become more frequent (Wunderling et al., 2022, PNAS). In this study, we could use ERA5 reanalysis data in an internally consistent and uniform way, finding that more frequent droughts (as the new climate normal) would increase Amazon forest tipping risks.

- Wunderling, N., Staal, A., Sakschewski, B., Hirota, M., Tuinenburg, O.A., Donges, J.F., Barbosa, H.M. and Winkelmann, R., 2022. Recurrent droughts increase risk of cascading tipping events by outpacing adaptive capacities in the Amazon rainforest. *Proceedings of the National Academy of Sciences*, 119(32), p.e2120777119.**

(2) As the authors will be aware, the original Cox et al. 2000 paper on Amazon “dieback” assessed numerous carbon cycle-related factors that contribute to forest risk. Some of these factors may indicate that, under climate change, the parameters inherent in the two-state dynamical system could change. The influence of these factors may either foster resilience or result in the opposite effect. Factors encompass stomatal closure, which can conserve soil moisture, aiding resilience, or heightened respiration in a warming climate, which could exceed photosynthesis and contribute to earlier dieback.

Answer #4: We agree with the reviewer that numerous factors (e.g. stomatal closure, heightened respiration etc.) can increase or decrease resilience. In our original analyses, we have included a mechanism to take the variability of the adaptation capacity into account. We randomly vary the adaptation constant *sigma* between 0.75 and 1.25 (mean 1.0) on a cell by cell level and compute tipping risks for 100 such created ensemble

members (please see methods section *ensemble construction*). This covers random sensitivities of local adaptation capacities and comprises factors that may change resilience (see II 558-562).

In addition, following the reviewers suggestion but also a comment by reviewer #1 (see our answer #4 above), we ran two additional adaptation capacity scenarios in order to account for factors that may change the forest’s resilience:

1. **Scenario A:** Following the assessments of reviewer #1 that drier regions in the rainforest may operate closer to their physiological limits and may have smaller safety margins. Therefore, we ran an experiment, where we scale adaptation capacities with regional rainfall levels that the forest receives. To compare these results with the results of our original simulations (adaptation capacities were chosen randomly between 0.75-1.25 standard deviations, see methods section *Adaptation & Ensemble Construction*), we chose 1.25 standard deviations for the wettest regions (=higher resistance to drier conditions) and 0.75 standard deviations for the driest regions (=lower resilience to drier conditions).

Fig. R2: Effects on tipping risks for adaptation scenario A. (a) Tipping risk map for the decade 2090-2099 in a SSP3-7.0 scenario, (b) Tipping response throughout the 21st century (time series) following an SSP3-7.0 scenario. Compare with Fig. 1b of the main manuscript.

2. **Scenario B:** We check the effects of the opposite assumptions as in Scenario A as an additional hypothetical experiment: Wetter regions have lower safety margins (0.75 standard deviations as adaptation capacity) and drier regions have higher safety margins (1.25 standard deviations as adaptation capacity).

Fig. R3: Effects on tipping risks for adaptation scenario B. (a) Tipping risk map for the decade 2090-2099 in a SSP3-7.0 scenario, (b) Tipping response throughout the 21st century (time series) following an SSP3-7.0 scenario. Compare with Fig. 1b of the main manuscript.

The result of scenarios A (see Fig. R2) is robust against our original analysis. We find, however, that Scenario A leads to higher tipping risks than our original analyses (see Fig. 1b of the main manuscript) because the dry regions are operating closer to their physiological limits (and can therefore not sustain even further drying (as the reviewer suspected). Scenario B is a sensitivity check with the more hypothetical assumption that wetter regions operate closer to their physiological limits. Still, results are robust against our original analysis (see Fig. R3 and compare to Fig. 1b). We summarize our robustness check in **II 684-693 and SI Fig. S20**.

(3) If I have understood correctly, the global warming estimates needed to drive the evapotranspiration equations have similarly been based only on the NorESM2 ESM? There are, of course, around 30-40 different climate models available in the CMIP6 database, and sometimes their full use provides natural bounds of uncertainty. I realise that this is a big task to utilise all ESM data, and I am not necessarily suggesting that this should happen. One way to circumvent this is to express changes in terms of global warming itself. The authors do this in their Figure 2, namely, the third coloured “axis” characterised by the colour bar showing temperature thresholds. However, it is important to note that depending on the rate of warming projected by different ESMs for a given SSP scenario, the timing of crossing identified thermal thresholds is less certain. At the very least, the authors should acknowledge this more fully. Statements suggesting that a tipping point could be reached at some point in the 2030s may not be valid under a standard SSP if calculations are made with an ESM that has low climate sensitivity. (It might be possible to incorporate this uncertainty into the authors’ Monte Carlo methods, without downloading the full CMIP6 ensemble, if a reference can be found for the range of projected warmings in South America for a given standard SSP).

Answer #5: We fully agree with the reviewer and have changed our manuscript following this suggestion. We note that the global mean temperature change is not based on NorESM2-values (which would indeed make timing and global warming level dependent on model-specifics in NorESM2 such as ECS) but instead, we base our global mean temperature increases on the median global temperature change as simulated in MAGICC7 (based on IPCC AR6 WG1 chapter 4, Figure 4.40a). With this procedure, we take the AR6 ensemble spread into account. We denote this now in **II 509-514.**

Overall, the exact timings when dangerous conditions for the Amazon forest systems are reached are also dependent on the timing of a certain deforestation level. We are more carefully arguing along those lines in the new version of the manuscript.

References:

1. Lee, J.-Y., J. Marotzke, G. Bala, L. Cao, S. Corti, J.P. Dunne, F. Engelbrecht, E. Fischer, J.C. Fyfe, C. Jones, A. Maycock, J. Mutemi, O. Ndiaye, S. Panickal, and T. Zhou, 2021: Future Global Climate: Scenario-Based Projections and Near-Term Information. In *Climate Change 2021: The Physical Science Basis. Contribution of Working Group I to the Sixth Assessment Report of the Intergovernmental Panel on Climate Change* [Masson-Delmotte, V., P. Zhai, A. Pirani, S.L. Connors, C. Péan, S. Berger, N. Caud, Y. Chen, L. Goldfarb, M.I. Gomis, M. Huang, K. Leitzell, E. Lonnoy, J.B.R. Matthews, T.K. Maycock, T. Waterfield, O. Yelekçi, R. Yu, and B. Zhou (eds.)]. Cambridge University Press, Cambridge, United Kingdom and New York, NY, USA, pp. 553–672, doi:10.1017/9781009157896.006.

One aspect I particularly appreciate about the manuscript is that it gives equal importance to assessing climate forcings and deforestation forcings. However, while Figure 2 is very informative, it does not address deforestation. The corresponding diagram that represents deforestation (for a business-as-usual scenario) is Figure S13. Incorporating Figure S13 into the main paper could greatly enhance its usefulness, either as a new Figure 3 or as additional panels in the existing Figure 2. The paper has space available – the current version of the main paper features only two figures.

Answer #6: We agree and have added Fig. S13 as part of Fig. 2 in the main manuscript now (panels c and d).

There are some small edits that could improve readability. As far as possible, please make the captions self-contained. It might, for instance, help in Figure 2 to remind the reader what the NAP and MCWD represent. Something like: "...global warming scenarios dependent on MAP and MCWD, that is annual rainfall and dry season intensity".

Answer #7: Absolutely, we adopted the recommended changes of the reviewer (see caption of Fig. 2).

For Figure 2b, another possibility is to make the large external circulate ring instead as an inset histogram?

Answer #8: We thank the reviewer for this suggestion and have tried a histogram out. However, when adding the former Fig. S13 to Fig. 2, the histogram bars of MAP and MCWD induced tipping events (together ~1.0% of all tipping events) become so small that they are barely visible. We decided therefore to keep Fig. 2b as it is.

The authors might consider speculating in the Discussion section about whether the coefficients in the dynamical system (Equations (2) and onwards) could be more directly related to ecological knowledge. Are there certain features of the more reliable dynamic global vegetation models (DGVMs) that would reveal and/or elucidate the parameter values of the dynamical system? Proposing this for future analysis could effectively establish a roadmap between

physical climate modelling and ecological assessment – a path that this current paper has initiated and navigates very well.

Answer #9: We thank the reviewer for this positive assessment of our work and agree that DGVMs and also complex Earth System models could improve the parametrization of our nonlinear dynamical systems model. In systematic DGVM tests reducing the precipitation in the Amazon region step wise (Nian et al., 2024, *Sci. Tot. Env.*) have shown that deriving DGVM based forest response curves is feasible. Future studies could base the coefficients on single or ensembles of DGVM response curves to test uncertainties. Current endeavors from complex Earth System Models and DGVMs (such as from projects like TIPMIP, TIPESM, OPTIMESM, etc.) will become available in the coming years which enable to derive a better parametrization for several models under a standardized experimental set up. Moreover, trait and tree individual based DGVMs (Lichstein et al., 2023; Sakschewski et al., 2021, *Biogeosciences*; Fisher et al., 2018, *GBC*) which enable for enhanced state of the art ecological process detail could serve as basis for deriving simple response curves as they put special emphasis on local vegetation adaptation and response diversity as well as other biodiversity mediated aspects.

Also, studies based on observational data making the connection between climate regimes and continental biome distributions (Hirota et al., 2011, *Science*; Malhi et al., 2009, *PNAS*; Zemp et al., 2017, *Nat. Communs.*; Flores et al., 2024, *Nature*) may help future parametrization. Moreover local rainfall exclusion experiments give valuable insights into local resistance margins of different rainforest types (see also our answer #1.2 to reviewer #1, Fig. R1). Whilst observational data can serve as a basis for climatic envelopes of rainforest types, biomes and even fire regimes, rainfall exclusion experiments could serve as a basis for deriving average physiological thresholds of water stress. We denote that in the new version of our manuscript (see **II 626-630**).

References:

- Sakschewski, B., Von Bloh, W., Drüke, M., Sörensson, A.A., Ruscica, R., Langerwisch, F., Billing, M., Bereswill, S., Hirota, M., Oliveira, R.S. and Heinke, J., 2021. Variable tree rooting strategies are key for modelling the distribution, productivity and evapotranspiration of tropical evergreen forests. *Biogeosciences*, 18(13), pp.4091-4116.
- Nian, D., Bathiany, S., Sakschewski, B., Drüke, M., Blaschke, L., Ben-Yami, M., von Bloh, W. and Boers, N., 2024. Rainfall seasonality dominates critical precipitation threshold for the Amazon forest in the LPJmL vegetation model. *Science of the Total Environment*, 947, p.174378.
- Lichstein, J.W., Longo, M., Bereswill, S., Blanco, C.C., Bonal, D., Chave, J., Christoffersen, B.D., de Paula, M.D., Derroire, G., Fisher, R.A. and Hickler, T., 2023. A model intercomparison project to study the role of plant functional diversity in the response of tropical forests to drought.
- Fisher, R.A., Koven, C.D., Anderegg, W.R., Christoffersen, B.O., Dietze, M.C., Farrior, C.E., Holm, J.A., Hurtt, G.C., Knox, R.G., Lawrence, P.J. and Lichstein, J.W.,

2018. Vegetation demographics in Earth System Models: A review of progress and priorities. *Global Change Biology*, 24(1), pp.35-54.

- Hirota, M., Holmgren, M., Van Nes, E.H. and Scheffer, M., 2011. Global resilience of tropical forest and savanna to critical transitions. *Science*, 334(6053), pp.232-235.
- Zemp, D.C., Schleussner, C.F., Barbosa, H.M., Hirota, M., Montade, V., Sampaio, G., Staal, A., Wang-Erlandsson, L. and Rammig, A., 2017. Self-amplified Amazon forest loss due to vegetation-atmosphere feedbacks. *Nature communications*, 8(1), p.14681.
- Malhi, Y., Aragão, L.E., Galbraith, D., Huntingford, C., Fisher, R., Zelazowski, P., Sitch, S., McSweeney, C. and Meir, P., 2009. Exploring the likelihood and mechanism of a climate-change-induced dieback of the Amazon rainforest. *Proceedings of the National Academy of Sciences*, 106(49), pp.20610-20615.
- Flores, B.M., Montoya, E., Sakschewski, B., Nascimento, N., Staal, A., Betts, R.A., Levis, C., Lapola, D.M., Esquivel-Muelbert, A., Jakovac, C. and Nobre, C.A., 2024. Critical transitions in the Amazon forest system. *Nature*, 626(7999), pp.555-564.

In summary, this manuscript is refreshing as it considers the simultaneous effects of climate change and deforestation. It genuinely strives to align understanding with dynamical systems that can demonstrate tipping point behaviour in a rigorous manner, allowing for transitions between states and hysteresis, as indicated by the cubic behaviour of Equation (2). It appears to be a paper that, with a bit more refinement, could significantly contribute to our knowledge of climate change and its impacts, making *Nature* a very appropriate journal for its publication. Therefore, the notes above represent a set of suggestions that may be beneficial to the authors. In my opinion, the primary concern is the need to slightly elaborate on the caveats and to be more explicit about uncertainties. By doing so, it will enhance the manuscript rather than detract from it. Such descriptions enable a strong focus for future researchers to “home in” on the remaining areas where knowledge is incomplete. I would be very pleased to review subsequent versions of the manuscript, and I sincerely hope that *Nature* provides an opportunity for the paper to be considered for potential publication, allowing the authors to create a new, slightly more refined version.

Answer #11: We thank the reviewer for their very friendly evaluation of our work and we are convinced that our additional explanations, caveats and opportunities for improvements are helpful for the reviewers and the general readership. We have added a section to our manuscript outlining the caveats and conservativeness of our approach (see II 224-233). We are looking forward to further feedback.

Referee #3 (Remarks to the Author):

The present paper focuses on the on the very interesting topic of the tipping points, a topic of undoubted scientific and socio-economic and ecological interest given its importance for the water resources.

It is well known that the Amazon biome is threatened by deforestation, which affects the hydrological cycle in the region and in the adjacent regions that it serves as a source of moisture for rainfall. Changes in its internal structure due to deforestation, changes in land use and internal droughts make the Amazon one of the most interesting and important regions for analysing future or even present changes. It is therefore of scientific interest to study the recycling capacity of the Amazon and its changes; and to link it to the expected deforestation. The present manuscript discusses the adaptive capacity of the Amazon rainforest in the face of climate change and deforestation, using climate modelling scenarios, adding additional projected deforestation data, and the study of moisture recycling as this is crucial for determining the resilience to changes.

We thank the reviewer for their assessment.

The methodology combines several data sets, some of which are a sub-product of modelling. In this line, the authors should add more information about all uncertainties of the datasets and the possible influences in the results. The authors show a robust check, but concerning the results of the paper, but it is needed a more explanation about the data itself. In some cases, reference is made to other papers, and it could be difficult for readers to check all the papers.

Answer #1: We are happy that the reviewer appreciates our numerous robustness checks of our results and also agree that we need to explain and robustify the moisture recycling network itself in a more detailed manner. In order to do so, we not only extend our explanations but also perform several additional robustness checks focussing (see answer #4 to reviewers #1 and #2, II 224-252, II 678-693, and SI Fig. 20 and Fig. 1d-f). In addition, we extended our model description regarding our atmospheric moisture recycling data (see methods section II 454-490 and Supplementary Note, pages 18-22).

The study assesses, using a dynamic system model, the risks of the tipping points under different emissions (focussing mainly in two) and land-use scenarios. It shows that without deforestation, the critical threshold for global warming is 3.7-4.0°C, where a third of the forest could lose stability. With deforestation, the tipping point is reached with only 1.5-1.9°C of warming and a 22-28% loss of forest cover, which could occur as early as the 2030s. Most of these events are triggered by intense droughts, which have large-scale cascading effects. The study highlights, and recommend, the urgency of keeping global warming below 1.5°C, halting deforestation (around 15%) and restoring affected areas to avoid critical risks in the Amazon. All these results combined, as commented, data and models.

The model to construct the story line of adaptation and deforestation is out of the expertise of this reviewers to provide a detail analysis of the results and argumentations. Therefore, the review of this point of the paper should be based on the opinion of other reviewers. However, this reviewer is able to offer a critical opinion on the using of the dataset derived from the atmospheric moisture tracking model UTrack (as moisture transport is the central theme of this reviewer's research), and which is used here to incorporate evaporation and precipitation due to recycling process.

We thank the reviewer for their helpful comment. In our revisions, we have focussed our efforts on the atmospheric moisture tracking model UTrack. The details of our responses are provided in the following responses to the reviewer.

In this paper, the UTrack model and method of computing associated is employed to analyse the contribution of the Amazon itself in terms of evaporation to its own precipitation, the recycling. Lagrangian methods are used for this purpose, but it is not the only one (this comment should be corrected in the manuscript, line 58-59). Moreover, exists more Lagrangian models using SSPS scenarios, even if not for the calculation of recycling, but they can be used. So, the line 84 about "the unique position" is a sentence that needs to be corrected [1, and references therein].

Answer #2: The reviewer is of course correct that this is not the only way to compute SSP-based moisture tracking. Indeed, Baker & Spracklen (2022, GRL) also analyzed Amazon moisture recycling in SSPs, but used "bulk recycling" methods that do not allow for grid-cell-to-grid-cell analyses. Also, Arias et al. (2023, Climate Dynamics) used CMIP5 models to study moisture transport between large regions in South America (where the Amazon was divided in two areas) in RCP8.5. We added reference to these studies in the Methods section (see II 471-490), explaining how our approach differs from these. In our study, we specifically designed UTrack model runs for this manuscript in order to produce the needed moisture recycling networks. As opposed to the recently published work by Staal et al. (2025, ESD) where only the years 2050-2059 and 2090-2099 are computed, we compute the moisture recycling networks for the entire 21st century. Additionally, and also as opposed to Staal et al., (2025, ESD), we tracked evapotranspiration from each grid cell of the Amazon separately, and stored the results per grid cell, per month and per SSP. For this we released 100 parcels per mm of evapotranspiration for each grid cell (see II 468-470). This resulted in a huge amount of individual simulations (>1.0 billion individual parcel trackings for this manuscript) involving myriad grid-cell to grid-cell interactions, which was necessary to produce a detailed and reliable network in which each node (i.e. each grid cell) is connected to each other node. These dedicated simulations are compute intensive and were dedicatedly designed for this paper.

In particular, we computed the following additional simulations:

(i) using two alternative scenarios for adaptation capacity, either with the assumption that dry species may have lower safety margins than wet species (see answer #4 to reviewer 1, see manuscript II 238-244, II 684-693 and Fig. S20).

(ii) using SSP5-8.5 as an additional scenario (on top of SSP2-4.5 and 3-7.0), see answer #14 below. We find that our results remain robust and within the expectation of the reviewers and us (see Fig. 1c and Fig. 1f).

In addition, we corrected this in the former lines 58-59 and line 84 (see II 64-65, II 100-101 in the new manuscript).

References:

- Arias, P.A., Rendón, M.L., Martínez, J.A. and Allan, R.P., 2023. Changes in atmospheric moisture transport over tropical South America: an analysis under a climate change scenario. *Climate Dynamics*, 61(11), pp.4949-4969.
- Baker, J.C.A. and Spracklen, D.V., 2022. Divergent representation of precipitation recycling in the Amazon and the Congo in CMIP6 models. *Geophysical Research Letters*, 49(10), p.e2021GL095136.
- Staal, A., Meijer, P., Nyasulu, M.K., Tuinenburg, O.A. and Dekker, S.C., 2025. Global terrestrial moisture recycling in Shared Socioeconomic Pathways. *Earth System Dynamics*, 16(1), pp.215-238.

Focusing on the moisture source concepts, there is a wide range of techniques that do not always give the same results, since the UTrack model chosen by the authors is a correct method, but not the state-of-the-art tracking model (line 13). This should be discussed.

We are happy to read that UTrack is considered an appropriate model. In the previous version of the manuscript, we called UTrack “a” state-of-the-art moisture tracking model and meant one of several state-of-the-art models. To prevent possible interpretations of us claiming it to be better than similar state-of-the-art tracking models, we deleted reference to “state of the art”.

In terms of moisture transport, the article focuses on the recycling capacity of the Amazon, but it is true for any region that its precipitation depends not only on internal flow, but also on external air masses with moisture that also influence its precipitation and thus the water cycle in its upper atmosphere and ground [2, and references therein]. Also missing is any reference to the impact of deforestation and rising temperatures under global change on precipitation downstream of the Amazon, on its natural sinks such as much of South America, or on other large basins such as La Plata.

This reviewer is aware that the paper deals with consequences within the Amazon, but its ecological and global system importance should not be forgotten, and reference to possible consequences in external regions is a necessity for the scientific community.

Answer #4: We agree with the reviewer that the local precipitation is a sum of the internal (and resolved) recycled moisture and the externally recycled moisture (Staal et al., 2018, *Nature Climate Change*; Gimeno et al., 2020, *Earth Science Reviews*). The focus of this study is to investigate Amazon tipping risks dependent on the failure of internal forest-to-forest moisture recycling. In our approach, the externally sourced moisture is given by the precipitation fields according to the NorESM2 runs for the different SSPs minus the locally recycled precipitation according to our UTrack runs. Thus, we do account for external moisture fluxes, but assume that these do not change in response to tipping events within the Amazon. At the same time, the reviewer is also correct in saying that considerable impacts of deforestation lie outside the considered region. These regions are mostly the ones that are further down south of the Amazon basin with strong agricultural production in Brazil, Bolivia and even further down south in Paraguay and Argentina (Rio de la Plata basin). We agree that the importance of these moisture flows should be investigated in the future, however, for this paper they are out of scope. We added this to our manuscript (see **II 262-265**).

References:

- Staal, A., Tuinenburg, O.A., Bosmans, J.H., Holmgren, M., van Nes, E.H., Scheffer, M., Zemp, D.C. and Dekker, S.C., 2018. Forest-rainfall cascades buffer against drought across the Amazon. *Nature Climate Change*, 8(6), pp.539-543.
- Gimeno, L., Vázquez, M., Eiras-Barca, J., Sori, R., Stojanovic, M., Algarra, I., Nieto, R., Ramos, A.M., Duran-Quesada, A.M. and Dominguez, F., 2020. Recent progress on the sources of continental precipitation as revealed by moisture transport analysis. *Earth-Science Reviews*, 201, p.103070.

As Amazon faces unprecedented pressures from global warming, deforestation and land-use change, a comprehensive analysis of the region is needed.

This reviewer points out that all the data from the recycling simulation are referenced as coming from a previous work by one of the authors [3] that analyses several river basins, of which the Amazon is one.

However, Amazon is not just a "random" region and, in particular, needs to be re-examined in greater depth. It is imperative to realise a complete and individual check for the Amazon Basin.

Answer #5: We fully agree with the reviewer that the Amazon forest is not a *random* region and deserves particular attention and care when atmospheric moisture recycling networks are quantified. Therefore, we designed and performed extensive and dedicated simulations with UTrack in the following way: We loop over all Amazonian grid cells and all months as well as the three SSPs, we tracked the evapotranspiration, releasing 100 moisture parcels for each mm of ET. This resulted in detailed global maps of moisture destinations for each of these months and grid cells. In this study we focus on the moisture flows to the other Amazonian grid cells, so we postprocessed these results to construct networks of moisture flows within the Amazon. Given the typical (average)

monthly ET in the Amazon rainforest region of 94.45 mm/month, we tracked a very large number of *1.1315 billion parcels* over all years and all SSP scenarios.

Importantly, none of these simulations were part of the Staal et al. (2025, ESD) paper and are prepared for this work only. We apologize that this was not clear and have now outlined this more thoroughly in the new version of the manuscript (see II 479-490).

Particularly, UTrack uses NorESM data in this study and in Staal et al (2024), and in this paper submitted to Nature uses the same periods and scenarios and validations.

It is in this part where this reviewer has more doubts derived from the previous article.

Answer #6: It is correct that the simulations for this study use the forcing data and model settings from Staal et al. (2025, ESD). In addition, we provide below a justification of the time step of four hours, which is a critical parameter for the reliability of the simulations, especially if grid cell to grid cell moisture flows are of interest. We show that, for the Amazon, this time step is optimal: larger time steps would sometimes result in moisture parcels “skipping” a grid cell even when precipitation occurs over the respective grid cell during the respective time step. If that happens, the moisture flow to that grid cell would be underestimated. A time step of four hours prevents this, while keeping computational demands feasible. We explain this in more detail below.

The optimal length of timestep to use is the length at which most moisture parcels do not travel further than a single grid cell. To determine at which time step this is the case, the model was run for 2015 (SSP1-2.6) for the following time steps: 1h, 2h, 3h, 4h, 6h, 8h, 12h, and 24h, and the location of each parcel was saved at each timestep. The distances between these positions were then calculated, and the travel distance of a parcel was calculated (Fig. R4). In the Amazon, the spatial resolution of NorESM2 translates to 110 km by 111 km at the largest cells and 110 km by 104 km at the smallest cells. For a time step of 4 hours, the mean travel distance of a parcel in one timestep is 54.6 km with a standard deviation of 42.1 km. The mean plus one standard deviation is 7.3 km lower than the shortest possible grid cell dimension (104 km). This means that with a time step of 4 hours, most of the parcels do not skip over grid cells. We added a sentence about the justification of the time step in the Methods (II 468-470).

Fig. R4: Mean \pm one standard deviation of parcel travel distances in the Amazon per timestep of 1h, 2h, 4h, 6h, 8h, 12h and 24h in a run for 2015 (SSP1-2.6).

The influence of using 10-year averages on the results is not clear, as the variability and particularity of each year in cases of drought events between wetter periods is smoothed out to a large extent. This should be taken into account and seen if there is an effect on the results.

Answer #7: We agree with the reviewer that single years can have a tremendous effect on the Amazon forest systems such as 2005 or 2010 droughts (see e.g. Lewis et al., 2011, Science; Panisset et al., 2018, Int J. Climatol.) or the most recent drought in the year 2024/2025 (see e.g. Marengo et al., 2024, American Journal of Climate Change). However, rainfall exclusion experiments have shown that trees are able to withstand (on a system-wide scale) individual drought years, and even subsequent drought years (Anderson et al., 2018, Philos. Trans. R. Soc. Lond. B Biol. Sci.; Flores et al., 2024, Nature). In a recent study by Sanchez-Martinez et al. (2025, Nature Ecology & Evolution), it has been summarized that vegetation state shifts occur after about a decade of drought experiments in this particular long-term rainfall exclusion experiment (see also our answer #1.2 to reviewer #1). Here as well as in former studies, larger trees show the biggest vegetation shift towards smaller and less thirsty vegetation (Sanchez-Martinez et

al., 2025, *Nature Ecology & Evolution*; Nepstad et al., 2007, *Ecology*; Meir et al., 2015, *BioScience*).

This means that, instead of looking into individual drought effects, our model is designed to quantify the effects of long-term (decade-long) shifts in the climate on the Amazon forest systems. We extended our argumentation on why we believe 10-year averages are a good length for measuring system-wide vegetation impacts and we of course mention more clearly that individual droughts can have a large impact as well. Please see our new manuscript, II 498-503.

Having said this, we have quantified the potential impact of past droughts on the Amazon forest in a past study if such droughts become the new climate normal (Wunderling et al., 2022, PNAS).

References:

1. Lewis, S.L., Brando, P.M., Phillips, O.L., Heijden, G.M.F. and van der Nepstad, D., 2011. The 2010 Amazon drought. *Science*.
2. Panisset, J.S., Libonati, R. and Gouveia, C.M.P., 2018. Contrasting patterns of the extreme drought episodes of 2005. *Int. J. Climatol*, 38, pp.1096-1104.
3. Marengo, J.A., Cunha, A.P., Espinoza, J.C., Fu, R., Schöngart, J., Jimenez, J.C., Costa, M.C., Ribeiro, J.M., Wongchuig, S. and Zhao, S., 2024. The drought of Amazonia in 2023-2024. *American Journal of Climate Change*, 13(03), pp.567-597.
4. Flores, B.M., Montoya, E., Sakschewski, B., Nascimento, N., Staal, A., Betts, R.A., Levis, C., Lapola, D.M., Esquivel-Muelbert, A., Jakovac, C. and Nobre, C.A., 2024. Critical transitions in the Amazon forest system. *Nature*, 626(7999), pp.555-564.
5. Anderson, L.O., Ribeiro Neto, G., Cunha, A.P., Fonseca, M.G., Mendes de Moura, Y., Dalagnol, R., Wagner, F.H. and de Aragão, L.E.O.E.C., 2018. Vulnerability of Amazonian forests to repeated droughts. *Philosophical Transactions of the Royal Society B: Biological Sciences*, 373(1760), p.20170411.
6. Sanchez-Martinez, P., Martius, L.R., Bittencourt, P., Silva, M., Binks, O., Coughlin, I., Negrão-Rodrigues, V., Athaydes Silva, J., Da Costa, A.C.L., Selman, R. and Rifai, S., 2025. Amazon rainforest adjusts to long-term experimental drought. *Nature Ecology & Evolution*, pp.1-10.
7. Meir, P., Wood, T.E., Galbraith, D.R., Brando, P.M., Da Costa, A.C., Rowland, L. and Ferreira, L.V., 2015. Threshold responses to soil moisture deficit by trees and soil in tropical rainforests: insights from field experiments. *BioScience*, 65(9), pp.882-892.
8. Nepstad, D.C., Tohver, I.M., Ray, D., Moutinho, P. and Cardinot, G., 2007. Mortality of large trees and lianas following experimental drought in an Amazon forest. *Ecology*, 88(9), pp.2259-2269.

Perhaps, and as the previous work is global, this issue was not emphasised, but as the authors now are dealing with a basin such as the Amazon, a particular study of the basin should be

made, and not use previous data, without a double check (and shown in this actual paper), from a more generalised results. In addition, because the reanalysis data and models for the Amazon have many gaps, and for the past period analysed it is recommended to check any results with station data.

Answer #8: While we agree that ground-based data would be very valuable, such station data is much too sparse, would need to be averaged and is not representative of future SSP scenario impacts on Amazon tipping risks. Therefore, it is unfortunately not possible to include station data in this analysis.

Importantly, we prepared the atmospheric moisture recycling data for this work only (and did not use the data from Staal et al., 2025, ESD) as outlined in our answers #2, #4, and #5.

Simulation settings: are the same for the paper of Staal et al 2004? Why? The Amazon should be re-calculated. Is it enough this number of simulated particles in a region with high evapotranspiration?

Answer #9: Indeed we produced new runs for this study (see also answers #2, #4, and #5). Here we tracked for each individual grid cell the evapotranspiration, for which we released 100 parcels for each mm of ET in the grid cell. In contrast, in Staal et al. (2025, ESD), parcels were simultaneously released across the Amazon, specifically 100 parcels for each mm averaged across the Amazon. The level of detail (in terms of number of parcels) is therefore on average a factor 416 (that is, the number of grid cells in the Amazon) greater than in Staal et al. (2025). Also, crucially, those runs did not allow for constructing a network among the Amazonian grid cells, because the runs were not done on an individual grid cell basis, see II 479-490.

Model evaluation: The authors evaluate by comparing the outputs from NORESM2 and ERA5, but the common period is too small. Figure A5 in Staal et al 2024 shows values with ERA5 from 2008-2017 and the simulations with NorESM2 with SPSS2-4.5 from 2015-2024, only 2 years in common. The period is not the same, and similarities in the map have not undergone statistical analysis. The evaluation should be redone for the Amazon region, using the complete period, and SPSS historical period. Why is the reason for using SPSS2-4.5 scenario for comparison? And why use this scenario as the baseline period? This aspect must be explained in the paper submitted to Nature.

For this reviewer, this aspect is not clear in the previous paper, and is a difficult aspect inherent in the new submission.

Answer #10: A comparison of global mean annual terrestrial precipitation recycling ratios using NorESM and ERA5 has indeed been presented in Staal et al. (2025), but not in the current study. The ERA5-based simulations in Staal et al. (2025) were from a publication of Tuinenburg et al. (2020), accompanying a dataset based on global moisture tracking simulations for 2008-2017 at 0.5 degree spatial resolution; hence the lack of optimal

overlap in terms of period. However, the focus of neither Staal et al. (2025) nor the current study is on moisture flows in particular years, but rather on multi-decadal trends. The rationale of using the SSP2-4.5 scenario was because it aligns most with the recent trajectory of the global climate.

However, we present below a validation using the years 2015-2019, comparing NorESM- and ERA5-based estimates of precipitation recycling ratios across the Amazon. This validation is also included in a new Supplementary Information (see **Supplementary Note, pages 18-22**) on validations of the moisture tracking runs (also see below for a discussion on wind speeds). The R^2 for the linear relation between the two approaches is 0.59, indicating generally good correspondence. However, the mean basin precipitation recycling ratio using NorESM2 is slightly higher than for ERA5 (0.27 and 0.26, respectively, see Fig. R5).

Fig. R5: Basin precipitation recycling ratio for 2015-2019 as a result of using data from NorESM2 (SSP2-RCP4.5) as forcing for the adapted UTrack moisture tracking model plotted against basin precipitation recycling ratio as a result of using data from ERA5. The average basin precipitation recycling ratio from NorESM2 is 0.27 and the average from ERA5 is 0.26. A linear regression gives $R^2 = 0.59$.

In Staal et al 2024 they said that the global pattern is “qualitatively similar”. This is not statistically valid. And the authors stated that the choice of a 10-year average for the recycling ratios largely overlaps with a 30-year period. In addition, the authors confirm the existence of a systematic bias. Does it occur over the Amazon? In which sense?

Answer #11: Fig. R5 shows that such a bias does not exist in the Amazon for locally sourced precipitation. The correspondence between basin precipitation recycling ratio

for ERA5 and NorESM2 is high. Furthermore, Fig. R6 shows that the difference in basin recycling ratio between NorESM2 and ERA5 are centred around 0. Also, 209 data points in NorESM2 have a lower basin recycling than ERA5 and 207 have a higher ratio, confirming the absence of a systematic bias.

Fig. R6: The differences in basin precipitation recycling ratio between NorESM2 (SSP2-4.5) and ERA5. Negative values indicate a lower recycling ratio in NorESM2 and positive values indicate a higher recycling ratio. Data are the same as in Fig. R5

The authors used a “quasi-wind” speed as monthly mean U and V values (as explained in Staal et al 2024, but not in the actual paper). Is this approach valid over the Amazon? The authors should take into account that over the Amazon a high-intensity wind structure as is the South American Low-Level Jet (SALLJ) acts as the main flow and defines the transport patterns in the region and south of the Amazon [4]. Moreover, what is the deviation from this quasi-wind vs the reanalysis and NorESM winds? What is the effect of using wind at 00 UTC (only one time) in the study for recycling over the Amazon? Does the maximum wind speed correspond with this time? Or not? It is very important to see the differences between the time with minimum and maximum wind speed, as the moisture is transported differently.

Answer #12: This is an excellent suggestion, because indeed wind speed variabilities (Jones et al., 2023, npj Climate and Atmospheric Science) and biases may have a significant impact on our results. We looked into this and dedicated a new section in the Supplementary Information (see **Supplementary Note**) on the wind speed deviations between NorESM2 and ERA5. For this, we weighted the wind speeds across the pressure layers by the specific humidity in these layers, to reliably reflect differences in horizontal moisture transport as a result of wind speed differences. Below we give the explanations and figures as provided in the new Appendix (see **Supplementary Note**).

We compared the daily wind speeds for 2015-2019 in ERA5 and NorESM2 (SSP2-4.5). We took the average wind speeds across the (25 and 8) pressure layers, in which we weighted these layers by their specific humidity on the respective day. As with the precipitation recycling ratios, we regridded the ERA5 results to 1° and took from these the nearest neighbors of each of the NorESM2 grid cells. The weighted wind speeds in ERA5 are 7.9% higher than in NorESM2. In Fig. R7, these ERA5 and NorESM2 wind speeds are plotted against each other.

Figure R7: Daily mean humidity-weighted wind speeds in the Amazon during 2015-2019 in ERA5 and NorESM2. The data points represent the 416 cells on the NorESM2 grid. The dashed line represents $y=x$. The mean wind speed in NorESM2 is 4.3 m/s and the mean in ERA5 is 4.7 m/s. A linear regression gives $R^2 = 0.84$.

An important difference between the two data sources is that ERA5 has hourly data whereas NorESM2 is based on daily data at 00UTC. Potentially, this may lead to bias in the tipping cascades, if winds at 00UTC tend to either under- or overestimate daily average winds. To explore this, we related the daily average humidity-weighted wind speeds in ERA5 against those at 00UTC. The two have high correspondence with $R^2 = 0.99$, and the wind speeds at 00UTC across the Amazon underestimate the daily averages by less than 3% (Fig. R8).

Figure R8: ERA5-based daily average humidity-weighted wind speed in the Amazon against the wind speed at 00UTC. The data points represent the 416 cells on the NorESM2 grid. The dashed line represents $y=x$. The mean wind speed is 4.7 m/s and that at 00UTC is 4.5 m/s. A linear regression gives $R^2 = 0.99$.

Because of the potential role of sub-daily moisture fluxes in the Amazon, we also explored the relation between daily wind speed in ERA5 and the within-day variations in wind speed. As expected, we found that daily average wind speed positively correlates to the daily variation in wind speed, as defined by the maximum hourly wind speed minus the minimum hourly wind speed in a day, with $R^2=0.60$. However, there was no significant correlation ($p = 0.35$) between daily wind speed and the coefficient of variation of hourly wind speeds in a day. Therefore, we conclude that there is likely no systematic bias that is caused by the use of daily wind speed data as compared to sub-daily wind speed data.

Reference:

Jones, C., Mu, Y., Carvalho, L.M. and Ding, Q., 2023. The South America Low-Level Jet: form, variability and large-scale forcings. *Npj Climate and Atmospheric Science*, 6(1), p.175.

How is the moisture recycling uncertainty due to this approach? The bias in the wind pattern could be larger or not when the Lagrangian model was applied. This concern should be clarified in depth as the Amazon has a very impact on the local climate, and on regional climate.

Answer #13: Thank you for these suggestions. We studied in depth the effects of wind speed changes on our main results: the tipping risks by the end of the century. We conclude that possible biases in wind speed (which do not appear to exist, see answers

#11 and #12) could not explain the different tipping risks. We also added this to the new Supplementary Information (see **Supplementary Note**):

Moving beyond the comparison of NorESM2 with ERA5, we analyzed the wind speed changes across the 21st century in order to explore how they relate to the estimated tipping risks. If a strong relation exists, it may suggest an important role of wind speed in our main results, pointing to a potentially important source of bias. For this we take the change in humidity-weighted daily wind speeds in the Amazon for 2091-2100 for SSP2-4.5, SSP3-7.0 and SSP5-8.5 compared to 2021-2030 for SSP2-4.5. We relate these changes to the tipping risks in the respective scenarios and decade. We find that across SSPs, the relation between wind speed change and tipping risk is significantly positive ($R^2 = 0.33$, see Fig. R9). However, the spatial correspondence between tipping risk and wind speed changes is weak (Fig. R10), indicating that wind speed changes are not the primary cause of tipping risk.

Together with the absence of clear bias in wind speeds as indicated by Figs. R5-R10, we rule out that our findings about tipping points are caused by bias in wind speeds in NorESM2, which we denote in our new Appendix (**see Supplementary Note**).

Figure R9: Tipping risk in the 2090s across the three SSPs related to the change in humidity-weighted daily wind speed (m/s) in the respective SSP and decade compared to wind speed in SSP2-4.5 for 2021-2030. The data points represent the 416 cells on the NorESM2 grid, colored by SSP. The solid red line shows the linear regression, with $R^2 = 0.33$.

Figure R10: Spatial wind speed change for SSP scenarios. Average wind speed changes between the 2020s of SSP2-4.5 and the 2090s of each SSP for a, SSP2-4.5; b, SSP3-7.0; c, SSP5-8.5. Comparing this figure with Fig. 1 of the main manuscript indicates that wind speed changes are not the primary cause for tipping risks.

Concerning the SPSS scenarios: It should be much clearer in the article the reasons for using only two climate change scenarios (SPSS2-4.5 and SPSS3-7.0), and consider SPSS5-8.5 as a complementary scenario. In addition to explaining why to use as baseline period one of the scenarios to show the results, the SPSS2-4.5.

Answer #14: We agree with the reviewer that our arguments why we use SSP2-4.5 and SSP3-7.0 should be explained clearer. The reasons are:

1. **SSP2-4.5 is the scenario that comes closest to our current emission and global warming pathways (see also ClimateActionTracker 2024: 2.7°C by 2100)**
2. **Without deforestation, there is no tipping in SSP2-4.5 at 2.8°C of global warming observed. But following an SSP3-7.0 scenario, tipping occurs at global warming levels of 3.6-4.0°C**

Therefore, the SSP2-4.5 and SSP3-7.0 were the two scenarios of our choice in our original analysis since they show the transition from no tipping pathways to substantial tipping risks. However, following the reviewer's suggestion we decided to prepare SSP5-8.5 simulations with the respective moisture recycling networks (which we prepared for this paper and are not part of Staal et al., 2025, ESD). We find that our results are robust using SSP5-8.5 (see Fig. R11). In case without deforestation, tipping risks increase at global warming levels of around 3.7-4.0°C showing the same affected area as for SSP2-4.5 and SSP3-7.0 (see Fig. R11a, and compare with Fig. 1). Also, with deforestation, a large-scale tipping point sets in at temperature values of around 1.7°C

(see Fig. R11b). Therefore, the results using SS5-8.5 are in line with our results from SSP2-4.5 and SSP3-7.0.

We have added a discussion and the respective figures around the SSP5-8.5 scenario to our revised manuscript (see additions in the chapter *Tipping due to global warming and deforestation*, II 495-497, Fig. 1d and Fig. 1f as well as its caption).

Fig. R11: Tipping responses in an SSP5-8.5 scenario for the Amazon rainforest with respect to (A) global warming alone (tipping risk map is taken at 4.9°C of global warming), and (B) for global warming and deforestation at the same time (tipping risk map at 1.7°C of global warming and 22% deforestation).

All the commented doubts and the results (and previous results) carried over from the moisture modelling (evaporation and precipitation) should be explained and highly validated, as the region is of very high interest and because the data are used in the following models and calculations in the paper submitted. The moisture recycling results may be affected by the lack of robustness and representativeness of the NorESM data and future scenarios. This is why I

recommend a more in-depth analysis of recycling, its behaviour in the future, and above all, a stronger statistical and comparative analysis to give robustness to the final results.

Answer #15: We thank the reviewer for their detailed comments on the manuscript and, in particular, on the underlying moisture recycling network. We agree that a more thorough discussion, robustness checks and analyses are necessary. In light of the reviewer comments, we substantially changed (and clarified where necessary) the following parts of our manuscript and added the following analyses:

1. We re-computed and added the following robustness checks to our analyses:
 - **SSP5-8.5:** In addition to our calculations of critical thresholds with SSP2-4.5 and SSP3-7.0, we also commuted critical thresholds with SSP5-8.5 (see additions in the chapter *Tipping due to global warming and deforestation*, II 495-497, Fig. 1d, fig. 1f as well as its caption)
 - **Adaptation scenarios:** We computed two additional scenarios for adaptation capacity taking into account different hypotheses for vegetation resilience (see manuscript II 684-693, SI Fig. S20 and answer#4 to reviewer#1)
2. We ran dedicated atmospheric moisture recycling simulations for this study and explained how those simulations were done in detail (see II 479-490 and the additional **Supplementary Note on Validating the moisture tracking algorithm**), also covering relevant uncertainties and robustness checks (in terms of how many tipping points are crossed taking into account uncertainties in atmospheric moisture transport, see also our above answers #2, #5, #6, #11, #12, and #13).

We thank the reviewer for their thorough feedback and are looking forward to further comments.

References

- [1] J. Eiras-Barca, J. C. Fernández-Álvarez, G. Alvarez-Socorro, S. Rahimi-Esfarjani, P. Carrasco-Pena, R. Nieto, L. Gimeno (2025) Projected changes in moisture sources and sinks affecting the US East Coast and the Caribbean Sea, *Annals of the New York Academy of Sciences*, DOI: 10.1111/nyas.15289
- [2] L. Gimeno, M. Vázquez, J. Eiras-Barca, R. Sorí, M. Stojanovic, I. Algarra, R. Nieto, A.M. Ramos, A.M. Durán-Quesada, F. Dominguez (2020) Recent progress on the sources of continental precipitation as revealed by moisture transport analysis, *Earth Science Reviews*, <https://doi.org/10.1016/j.earscirev.2019.103070>
- [3] Staal, A., Meijer, P., Nyasulu, M. K., Tuinenburg, O. A., and Dekker, S. C.: Global terrestrial moisture recycling in Shared Socioeconomic Pathways, *EGUsphere* [preprint], <https://doi.org/10.5194/egusphere-2024-790>, 2024.
- [4] Jones, C., Mu, Y., Carvalho, L.M.V. et al. The South America Low-Level Jet: form, variability and large-scale forcings. *npj Clim Atmos Sci* 6, 175 (2023). <https://doi.org/10.1038/s41612-023-00501-4>

Dear Reviewers, dear Editor,

We are grateful for the very substantial feedback and comments by the reviewers. The suggestions, references and advice have helped to revise and improve our manuscript. The comments of the reviewers have led to the following major changes in the revised manuscript:

1. We have added a definition of what we mean by “tipping” in the present manuscript following, e.g. van Nes et al., 2016 *What do you mean ‘tipping point’?*). By tipping, we mean that local thresholds are surpassed under global warming and deforestation, leading to a qualitative shift in the dominant feedbacks from self-stabilizing to self-destabilizing. The point at which this feedback shift occurs marks the tipping point.

Then, crossing such a tipping point oftentimes does (but not necessarily always) result in hysteresis. In many cases, the emergence of self-destabilizing feedbacks makes it very difficult to return to the original state because the feedback mechanisms that previously maintained stability (here: the atmospheric moisture recycling) are lost.

Based on these preconditions, we have selected our mathematical modeling approach of local-scale tipping points (hereafter called critical transitions) that can be crossed either due to deforestation or global warming induced impacts. These local-scale transitions can then spread and cascade via the lack of atmospheric moisture transport to remote regions, and lead to regional to substantial transitions across the Amazon forest, beyond 1.5-2.0°C and 22-28% of deforestation.

2. Based on point 1. and the advice of the reviewers, we have decided to use the vocabulary of *transitions* instead of tipping point(s). In the discussion, however, we outline what would be needed for a full tipping point analysis, which we argue can be followed up in future studies based on our work. In light of the reviewer comments, we also went carefully through our manuscript and adapted our language accordingly.

We have also considered all further feedback points raised by the reviewers. Please find below a detailed point-by-point response to the comments. We also attached the new version of our manuscript and supplement, and marked the changes in blue. We are grateful for this opportunity to improve our manuscript.

Sincerely yours,

Nico Wunderling, Boris Sakschewski, Johan Rockström, Bernardo Flores, Marina Hirota, and Arie Staal

Reference:

- Van Nes, E.H., et al., 2016. What do you mean, ‘tipping point’?. *Trends in Ecol. & Evol.*

Referees' comments:

Referee #1 (Remarks to the Author):

The authors invested a tremendous amount of effort in revising the manuscript and have carefully addressed several reviewer comments, particularly points 3, 4, and 6. The manuscript is clearly stronger as a result of these efforts. However, there remain important issues that require clarification and refinement.

We thank the reviewer for their assessment on our efforts in revising our manuscript. We are also grateful for the further feedback provided below.

Point 1. Tipping point unclear.

The authors provided an extensive response to concerns about evidence for critical transitions in the Amazon. However, their reply conflates processes such as natural variability, transient reductions in biomass, and disturbance-driven changes (e.g., fire), none of which necessarily represent critical transitions. Some of these processes may be better described as temporary state changes, as in Silvério et al. (2022). By contrast, the work of Hirota and Staver emphasizes the distribution and coexistence of alternative biomes under particular climatic conditions, which is conceptually distinct from forest degradation.

Answer #1: We thank the reviewer for this important clarification. We agree that natural variability, transient biomass reductions, and disturbance-driven changes (e.g. fire, mentioned now in II 118-121) do not necessarily constitute critical transitions, and we did not intend to conflate these processes. Our study does not focus on reversible degradation or short-term state changes as discussed, for example, in Silvério et al. (2022, ERL).

Instead, we assume spatially explicit, feedback-driven transitions that arise when local ecological thresholds are exceeded under the combined influence of global warming and deforestation. Beyond these ecological thresholds, there is a shift in feedbacks from self-stabilizing to self-destabilizing due to the lack of moisture supply. These local transitions can then propagate through the atmosphere via weakened evapotranspiration and moisture recycling, potentially amplifying initially local forest loss into larger scale system reorganization. This perspective differs from studies emphasizing equilibrium biome coexistence (e.g. Hirota et al. 2011, Science; Staver et al., 2011, Science), as we focus on coupled dynamics rather than static distributions of vegetation states.

Further, the long-term biomass declines due to drought, for instance, may indeed not be hysteretic, but they can be considered permanent (in the new rainfall regime that does not automatically recover). Therefore, they fit the framework of our study (see also our next answer #2) as the biomass and ET changes also will not automatically revert as a consequence.

References:

1. Silvério, D.V., Oliveira, R.S., Flores, B.M., Brando, P.M., Almada, H.K., Furtado, M.T., Moreira, F.G., Heckenberger, M., Ono, K.Y. and Macedo, M.N., 2022. Intensification of fire regimes and forest loss in the Território Indígena do Xingu. *Environmental Research Letters*, 17(4), p.045012.
2. Hirota, M., Holmgren, M., Van Nes, E.H. and Scheffer, M., 2011. Global resilience of tropical forest and savanna to critical transitions. *Science*, 334(6053), pp.232-235.
3. Staver, A.C., Archibald, S. and Levin, S.A., 2011. The global extent and determinants of savanna and forest as alternative biome states. *Science*, 334(6053), pp.230-232.

This highlights a central issue: the manuscript does not offer a clear or consistent definition of what is meant by a “tipping point.” In several passages, the authors claim to show a change in stable state. By definition, such a change would imply a transition from forest to a non-forest ecosystem. Yet, with the exception of Hirota and Staver, most of the cited examples do not represent true critical transitions. This is important because many readers will interpret the title and abstract to mean that most Amazon forests will cease to exist once thresholds are crossed. That is not what the manuscript is currently demonstrating.

Answer #2: We thank the reviewer for explaining their point. Following the advice and as we cannot fully test for hysteresis and irreversibility in this manuscript, we remove the tipping point framing from the manuscript and instead speak of *transitions*. We changed the manuscript accordingly and also mention the limitation that we do not test for hysteresis in the discussion (II 242-246, see also our **answer #3).**

We have also included a definition of tipping points explicitly in the manuscript, which we use to simulate local-scale transitions. We include this in II 642-650 and **Extended Data Fig. 2 (Fig. R1 below)** and also outline it below. Based on the broad literature on tipping elements (e.g. the latest Global Tipping Points Reports 2025, 2023 and earlier key literature, e.g., Armstrong McKay et al., 2022, *Science*; van Nes et al., 2016, *TREE*), we define a tipping point as:

A change in state away from a forest state towards an alternative state. Beyond the tipping point, a destabilizing feedback drives the forest towards an alternative state in a self-reinforcing manner.

In our study, these state changes can occur on a local scale (Brando et al., 2025, ARER) and spread via the moisture recycling network.

Then, crossing such a tipping point oftentimes does (but not necessarily always, see e.g. van Nes et al., 2016, *TREE*; Armstrong McKay et al., 2022, *Science*) result in hysteresis. In many cases, the emergence of self-destabilizing feedbacks makes it very difficult to return to the original state because the feedback mechanisms that previously maintained stability are lost.

This irreversibility and hysteresis characteristics cannot be fully tested for by us because the forcing is not reversed in our scenarios (SSP2-4.5, SSP3-7.0, SSP5-8.5). This means

that the backward direction (from alternative to forest state) cannot be tested for. However, the forward direction (from forest to alternative state) has been assessed in the literature (e.g. in Flores et al., 2024, Nature).

Based on the previous, we have selected our mathematical model of local-scale tipping points (hereafter called critical transitions) that can be crossed either due to deforestation or global warming induced impacts, aligning with reviewer #2 (see below from answers #59 on). The resulting stability landscape is represented in Fig. R1. In our study, each 1x1° grid cell is represented by such a stability landscape (local scale transition), which can be modelled with the following dynamical systems equation Eq. R1 (see first three terms of Eq. 2 in the manuscript):

$$\frac{dx_i}{dt} = -x_i^3 + x_i + C_{\text{crit},i}(\text{MAP}_i, \text{MCWD}_i) \quad (\text{Eq. R1})$$

Fig. R1: Map of the Amazon basin with a grid representing our study area. In this grid, each 1x1° grid cell is represented by a dynamical systems equation with two stable equilibrium states (forest and alternative state, see inset) that result from increasing drought (MCWD) and decreasing rainfall (MAP), resembling the dynamical systems equation Eq. R1. We include this figure into our new version of the manuscript as *Extended Data Figure 2*.

This bistability on a local scale can then spread via the lack of moisture transport (domino effects) to regions far away (see full equation 2 of the main manuscript):

$$\frac{dx_i}{dt} = -x_i^3 + x_i + C_{\text{crit},i}(\text{MAP}_i, \text{MCWD}_i) + \sum_{k=1, k \neq i}^{N_{\text{grid cells}}} R_{ki}(\Delta\text{MAP}_{ki}, \Delta\text{MCWD}_{ki}) \frac{x_k}{2} \quad (\text{Eq. R2})$$

Fig. R1 is in line with the recent review by Brando et al. (2025, ARER) that has suggested local to regional tipping points instead of a global tipping point for the Amazon forest (summary point 2) that can spread via a domino effect (their summary point 3). We use this and show that under certain circumstances (i.e. global warming of 1.5-2.0°C plus deforestation 22-28%), local transitions (tipping events) can spread to large-scale (system-relevant) tipping events across substantial areas of the Amazon forest. In particular, the latest assessments by Flores et al. (2024, Nature) and Brando et al. (2025, ARER) provide observational and Earth System modelling efforts that support the notion of a true tipping point on local scales - in particular regarding the largest and most vulnerable trees in the Amazon (see our answer #3).

Again, in summary, as we cannot fully test for hysteresis and irreversibility, we decided to remove the tipping points framing and use *transitions* instead.

References:

- Lenton, T.M., Milkoreit, M., Willcock, S., Abrams, J.F., Mc Kay, D.L.A., Buxton, J.E., Donges, J., Loriani, S., Wunderling, N., Alkemade, F. and Barrett, M., 2025. Global tipping points report 2025.
- Armstrong McKay, D.I., Staal, A., Abrams, J.F., Winkelmann, R., Sakschewski, B., Loriani, S., Fetzer, I., Cornell, S.E., Rockström, J. and Lenton, T.M., 2022. Exceeding 1.5 C global warming could trigger multiple climate tipping points. *Science*, 377(6611), p.eabn7950.
- Brando, P.M., Barlow, J., Macedo, M.N., Silvério, D.V., Ferreira, J.N., Maracahipes, L., Anderson, L., Morton, D.C., Alencar, A., Paolucci, L.N. and Jacobs, S., 2025. Tipping points of Amazonian forests: beyond myths and toward solutions. *Annual Review of Environment and Resources*, 50.
- Flores, B.M., Montoya, E., Sakschewski, B., Nascimento, N., Staal, A., Betts, R.A., Levis, C., Lapola, D.M., Esquivel-Muelbert, A., Jakovac, C. and Nobre, C.A., 2024. Critical transitions in the Amazon forest system. *Nature*, 626(7999), pp.555-564.
- Van Nes, E.H., Arani, B.M., Staal, A., van der Bolt, B., Flores, B.M., Bathiany, S. and Scheffer, M., 2016. What do you mean, 'tipping point'? *Trends in Ecology & Evolution*, 31(12), pp.902-904.

If the intended meaning is a biomass reduction (e.g., 40%), the system still qualifies as forest, since biomass is largely stored in large trees. Even fire-induced biomass losses beyond that threshold do not necessarily represent a new stable state, as many studies show substantial

recovery potential over decades. Without greater precision, the manuscript risks conflating abrupt but transient biomass reductions with genuine tipping points leading to critical transitions.

Answer #3: We agree with the reviewer that a 40% biomass reduction is not sufficient as long as the system still qualifies as forest. However, as our first review has shown, it is mostly the largest trees that are most vulnerable and die first. In this case, when the largest trees die, this would qualify as a transition in our view.

Furthermore, a large die-off of the largest trees not only represents a pure biomass reduction but also a reduction in the ability to provide ecosystem services (and the moisture transport strengths) to other remote regions. Most important is the fact that at this point tree mortality is self-reinforced. We represent this in our Eq. (2) (see also Eq. R2 above).

However, the reviewer picks on an important point here as we do not explicitly investigate the timescales of tipping/transitions since our scenarios are non-overshoot scenarios (SSP2-4.5, SSP3-7.0, SSP5-8.5), also preventing a systematic hysteresis investigation. This means, while our study suggests the possibility of large scale transitions in the Amazon forest, the lack of timescales prevents a full proof. We outline this limitation of our study now more explicitly in our study and remove the tipping point framing (see II 242-246).

Nevertheless, exceeding critical levels of global warming and deforestation can trigger local-scale transitions that can self-amplify through their spatial connectedness (atm. moisture transport) and have the potential to scale up to larger-scale transitions across the forest.

On drought, it is important to recognize that while experiments do not capture long-term recycling, historical events such as the Little Ice Age suggest that even under extreme dry conditions, widespread tropical forest collapse was not observed—impacts were largely confined to forest edges.

Answer #4: While environmental changes in the recent past have not shown substantial declines even under drier conditions, deforestation did not play a role in these times. During the Little Ice Age potentially there is even a hypothesis of the opposite, namely increased natural reforestation due to the decline of indigenous population during colonization times. So there was no simultaneous pressure from both adverse drivers of climate change (drying) and deforestation. However, while the accumulated literature from paleorecords indicates no biome-wide tipping over millions of years, there are indications of less suitable climate and increased vulnerability in the northern Amazon region over the last 12,000-25,000 years (Akabane et al., 2024, Nat. Geosci.).

Reference:

Akabane, T.K., Chiessi, C.M., Hirota, M., Bouimetarhan, I., Prange, M., Mulitza, S., Bertassoli Jr, D.J., Häggi, C., Staal, A., Lohmann, G. and Boers, N., 2024. Weaker Atlantic overturning circulation increases the vulnerability of northern Amazon forests. *Nature Geoscience*, 17(12), pp.1284-1290.

Further clarification is also required regarding the following statement:

“Furthermore, without drought-fire interactions, there is little evidence of a sharp, irreversible shift. Given the challenges in validating the model, the results should be presented with greater nuance, contextualized based on observational studies, experimental approaches, and other types of models (DGVMs).”

Answer #5: We appreciate the reviewer’s call for nuance. Indeed, models alone (for example, DGVMs) often simplify fire, drought, and further feedbacks and it may therefore be challenging to find tipping points/transitions in them. But the empirical evidence from field and observational studies - notably abrupt increases in Amazonian tree mortality due to drought-fire interactions (Brando et al. 2014, PNAS) and the forest’s failure to recover after repeated wildfires in the Amazon (Flores et al. 2021, Ecosystems; Silvério et al. 2022 ERL) - supports a substantially more deterministic view that forest to alternative state transitions may be irreversible.

In our model, we do not explicitly resolve biogeophysical processes but map the dynamics onto the local dynamics of a nonlinear dynamical equation (Eq. (2) in the main manuscript) and therefore only indirectly include drought-fire feedbacks (which would be important in process-resolving models). As such, we agree with the reviewer that a more cautious statement is needed and we emphasize that we do not resolve biogeophysical processes in the manuscript. We included this in the manuscript and (II 248-256) and outline thereafter how we cover such uncertainties/sensitivities.

References:

1. Brando, P.M., Balch, J.K., Nepstad, D.C., Morton, D.C., Putz, F.E., Coe, M.T., Silvério, D., Macedo, M.N., Davidson, E.A., Nóbrega, C.C. and Alencar, A., 2014. Abrupt increases in Amazonian tree mortality due to drought–fire interactions. *Proceedings of the National Academy of Sciences*, 111(17), pp.6347-6352.
2. Flores, B.M. and Holmgren, M., 2021. White-sand savannas expand at the core of the Amazon after forest wildfires. *Ecosystems*, 24(7), pp.1624-1637.
3. Silvério, D.V., Oliveira, R.S., Flores, B.M., Brando, P.M., Almada, H.K., Furtado, M.T., Moreira, F.G., Heckenberger, M., Ono, K.Y. and Macedo, M.N., 2022. Intensification of fire regimes and forest loss in the Território Indígena do Xingu. *Environmental Research Letters*, 17(4), p.045012.

The authors explain that they simulate risks of transitions without allowing for recovery. While mathematically understandable, this assumption weakens claims that the Amazon will shift to an alternative ****stable**** state. Moreover, the absence of explicit timescales makes the results difficult to interpret in conservation or climate mitigation/adaptation contexts. The reference to ten years as a relevant timescale is particularly problematic, as this is far too short to capture the long-term vegetation–climate feedbacks involved – which should occur in much longer time scales, likely in thousands of years. This raises concerns that the interpretation of the results does not adequately reflect the model’s assumptions and limitations.

Answer #6: While recovery is allowed in principle, we do not explicitly include timescales but only run equilibrium experiments. This is also because our input emission scenarios do not feature overshoot scenarios. Consequently, once a local tipping point is crossed, the system can therefore not return to its previous state as the forcing is held constant. We clarify this in the new version of the manuscript (see II 242-246).

The reviewer also raises the issue of transition timescales, which we address in two ways:

First, while the effects of feedbacks on forest extent and structure may unfold relatively slowly - over decades to centuries, depending on spatial scale and the direction of change - moisture recycling itself operates on much shorter timescales. The atmospheric residence time of water is on the order of 10 days, implying that once a transition is initiated, the breakdown of this feedback can occur rapidly.

Second, repeated fires can transform forest ecosystems into savanna within a few decades (Flores and Holmgren, 2021, *Ecosystems*) or into a degraded open-vegetation state (Veldman and Putz, 2011, *Biol Conservation*), whereas at regional scales such transitions may take a century or longer. The key mechanism is the shut-down of the forest-rainfall feedback (that we explicitly include in our work), which keeps the humid-forest system resilient in its current basin of attraction. Climate forcing, combined with deforestation, simplification of the biome (e.g. loss of biodiversity) weaken this vital feedback and hence may tip the system over into a new regime, in which the Amazon irreversibly drifts into a degraded state. This may not be an abrupt collapse (although a transition that may last 100 years seems remarkably abrupt over the millions of years of Amazonian history). Yet, what is important is that it represents an irreversible trajectory, as the system becomes dominated by a new set of self-drying feedbacks - weakened convective rainfall, and low capacity to maintain humidity due to radically higher VPD (vapor pressure deficit), very likely associated with hysteresis (see for instance Flores et al. 2024 Extended Data 1 <https://www.nature.com/articles/s41586-023-06970-0/figures/5>). So, in summary, the timescales of transitions are inherently uncertain and likely span a wide range, from decades to centuries (Armstrong McKay et al., 2022, *Science*), which is also why we ran equilibrium simulations in this work and overshoots are beyond the scope of this work. We elaborate on this in our manuscript and present the mentioned timescales with greater nuance (see II 242-246).

References:

1. Flores, B.M. and Holmgren, M., 2021. White-sand savannas expand at the core of the Amazon after forest wildfires. *Ecosystems*, 24(7), pp.1624-1637.
2. Veldman, J.W. and Putz, F.E., 2011. Grass-dominated vegetation, not species-diverse natural savanna, replaces degraded tropical forests on the southern edge of the Amazon Basin. *Biological Conservation*, 144(5), pp.1419-1429.
3. Armstrong McKay, D.I., Staal, A., Abrams, J.F., Winkelmann, R., Sakschewski, B., Loriani, S., Fetzer, I., Cornell, S.E., Rockström, J. and Lenton, T.M., 2022. Exceeding 1.5 C global warming could trigger multiple climate tipping points. *Science*, 377(6611), p.eabn7950.

Point 2. Deforestation impacts on rainfall

The manuscript emphasizes studies suggesting strong impacts of deforestation on rainfall. However, many other studies report considerably smaller effects. For example, Spracklen's review concludes that even complete deforestation of the Amazon would reduce rainfall by approximately 16–18%. Given that the authors rely on a single modeling framework, it is essential to acknowledge this broader body of work and to present a more balanced discussion of the uncertainties in rainfall sensitivity to deforestation.

Answer #7: We place our findings within the broader body of work and mention relevant uncertainties (see II 268-274). In doing so, we also contextualize the following studies: A recent analysis from across the tropics concludes that tropical deforestation causes large reductions in observed precipitation (Smith et al., 2023, *Nature*). The effect of deforestation (forest clearing) on evapotranspiration over large scales has been studied for at least four decades (see Salati and Vose, 1984, *Science*). And it is also important to mention that current ESMs usually can not resolve forest-rainfall feedbacks, while our approach allows us to quantify this forest-rainfall feedback and its impacts on the stability of the forest. We can even do so using the original NorESM output, which itself is likely in the same range as other ESMs.

References:

1. Smith, C., Baker, J.C.A. and Spracklen, D.V., 2023. Tropical deforestation causes large reductions in observed precipitation. *Nature*, 615(7951), pp.270-275.
2. Salati, E. and Vose, P.B., 1984. Amazon basin: a system in equilibrium. *Science*, 225(4658), pp.129-138.

Point 3. Addressed.

Point 4. Addressed.

Point 5. Forest regeneration capacity

This point remains insufficiently developed. The central issue is that Amazonian forests have high natural regeneration capacity, but this is often reduced by repeated clearing before they reach the legal threshold for recognition as forest under Brazilian law. This reflects land-use dynamics more than ecological limitations. Moreover, several studies—including one cited by the authors—show that evapotranspiration tends to recover rapidly after disturbance. This nuance should be incorporated into the manuscript.

Answer #8: We agree with the reviewer and mention that nuance in the new manuscript (see II 745-747): The evapotranspiration of the Amazon forest can have high regenerative capacities on timescales of a few years if vegetation is undisturbed after clearance (see e.g. Brando et al., 2019, *Global Change Biology*). However, most of the area that is cleared once, is cleared repeatedly (Nunes et al., 2020, *ERL*).

Reference:

1. Brando, P.M., Silvério, D., Maracahipes-Santos, L., Oliveira-Santos, C., Levick, S.R., Coe, M.T., Migliavacca, M., Balch, J.K., Macedo, M.N., Nepstad, D.C. and Maracahipes, L., 2019. Prolonged tropical forest degradation due to compounding disturbances: Implications for CO₂ and H₂O fluxes. *Global Change Biology*, 25(9), pp.2855-2868.
2. Nunes, S., Oliveira, L., Siqueira, J., Morton, D.C. and Souza, C.M., 2020. Unmasking secondary vegetation dynamics in the Brazilian Amazon. *Environmental Research Letters*, 15(3), p.034057.

Point 6. Addressed.

Point 7. Role of CO₂

The response here does not fully resolve the concern. The issue is not what is known or unknown about CO₂ effects per se, but rather that the framing of “tipping points” could be strongly influenced by CO₂-driven reductions in rainfall in the datasets used for modeling, especially given that the model is run offline. This potential bias should be explicitly acknowledged.

Answer #9: We agree with the reviewer and thank the reviewer for clarifying this issue. Because our model is run offline and forced by CMIP6-derived climate variables, CO₂ physiological effects on precipitation may implicitly be included in the forcing. Consequently, CO₂-driven reductions in evapotranspiration and moisture recycling may contribute to changes in MAP and MCWD that push the system across identified thresholds. We mention this caveat in our manuscript (see II 279-283).

Referee #2 (Remarks to the Author):

Thank you once more for inviting me to assess the paper: “Pinpointing Amazon forest tipping in global warming and deforestation pathways” by Wunderling et al.

What is immediately obvious from scanning the entire set of requests and replies is how thorough the suggestions are and how, overall, the authors have responded with great care.

We appreciate the reviewer’s thoughtful evaluation of our revisions and are grateful for the additional feedback provided.

After reviewing the responses to my requests, I have a few further points below that the authors might consider.

Author Answer #3. Thank you for considering the possibility of using ERA5 data to force the UTrack transport model. I find it particularly interesting that the authors regard NorESM to be more accurate, given that ERA5 is likely “too wet”. Ultimately, we need the best possible data and a broader range of climate models to (1) ensure there are no initial biases when simulating the current climate, and (2) enable us to estimate a wide range of uncertainties for future projections. I am content that this particular paper does not scan across all CMIP6 models, but this should be more clearly acknowledged as a “next step” in this line of research. Although not perfect, differences across CMIP6 models naturally and usefully define uncertainty bounds on expected future changes. Maybe stronger recommendations can be made in the Discussion and Conclusions, urging reanalysis products to become as accurate as possible, especially in remote regions? And, additionally, it should be requested that the CMIP6 ensemble ideally stores the relevant variables needed to force tracking models at finer scales (and finer temporal resolutions?).

Answer #10: We agree with the comment of the reviewer and have acknowledged that the ideal uncertainty range would be the full ensemble of the CMIP6 models complemented by improved reanalysis products. However, for the latter a way to continue their timeseries (the one of the reanalysis products) into the future would be required. In addition, it would be great if the CMIP6 and CMIP7 ensemble models would include the relevant variables for moisture tracking, namely *pr* (Precipitation), *evspsbl* (surface evaporation), *hus* (specific humidity), *prw* (precipitable water), *wap* (vertical pressure velocity), *ua* (zonal wind) and *va* (meridional wind) at sub-daily resolution.

We have adapted our manuscript accordingly in II 290-291.

Author Answer #4. An attractive feature of the manuscript is the mapping of findings onto a nonlinear dynamical system structure, with the full equation (2). The cubic equation format is a generic form for illustrating tipping points with two states. The advantage of simpler descriptions is that they allow scanning of parameters. It is appreciated that the authors have done exactly this, by modifying parameter sigma (which indirectly, is a form of bifurcation parameter). The

authors refer to this parameter as describing adaptation capacity. It is good that the authors included the related diagrams as a new Figure S20. Please make the citation in the main paper to this figure as clear as possible, as it provides a robust form of uncertainty analysis.

Answer #11: We thank the reviewer for the appreciation of our work and, indeed, the modification of the sigma parameter that indirectly is a bifurcation parameter. Therefore, our Extended Data Fig. 9 (formerly Fig. S20) serves as a sensitivity experiment that we are now more clearly referencing from the main manuscript (see II 275-277).

Author Answer #4. Would the authors be willing to expand a bit more (for example, in the Discussion) on the factors they expect to influence the sigma parameter, such as water access features, stomatal response, and acclimation effects? An additional benefit of a simpler underlying model with clearly defined parameters is that it simplifies mapping future research and findings (for example, on land surface resilience) onto this common framework. Emphasising this point in the Discussion is important, as it will also encourage citation of the manuscript for its contribution in providing a key dynamical system with parameters that are useful for others to set.

Answer #12: The parameter sigma represents local adaptive capacity by scaling the distance between historical climate variability and critical thresholds. It can be interpreted as an aggregate measure of processes such as rooting depth and water access, stomatal regulation and hydraulic safety, species composition, and acclimation to local climate. While these mechanisms are not explicitly resolved, sigma provides a compact way to incorporate their net effect on the resilience of the forest. Importantly, this abstraction facilitates mapping future experimental, observational, or model-based insights onto a common dynamical framework, enabling systematic refinement of tipping risk estimates. We have expanded our explanation in the manuscript (see II 248-256).

Author Answer #5. Thank you for this amendment, and I can see these additional lines (i.e. 509-514). Please help the reader a little more though, so it is possible to see how a different ECS value translates through Eqn (1). Presumably, this is in the standard equation for evaporation. The reason this is helpful is because it allows translation of other analyses, such as the ongoing efforts to constrain ECS more tightly, onto the dynamical system present in this paper for Amazon multi-states.

Answer #13: We thank the reviewer for this additional clarification need to which we agree. We use the IPCC AR6 WG1 (see their Fig. 4.40) mean values for global temperature change across the 21st century in response to the emission pathways SSP2-4.5, SSP3-7.0, and SSP5-8.5. We mention this explicitly in the new manuscript (see II 566-568).

Author Answer #8. This partly answers the point raised above about linking parameter sigma to underlying physical or ecological processes.

Answer #14: We thank the reviewer for this comment and have integrated this answer also above to respond to the reviewer's comment on the meaning and possibilities to constrain sigma (see answer #12).

In summary, I believe the paper is now very close to publication. The minor points raised above concern ensuring that the underlying equations, assumptions, and open questions are communicated as clearly and unambiguously as possible. Therefore, I suggest the authors review the manuscript one more time, considering how other researchers conducting Amazonian studies might most easily grasp the ideas and equations presented here, so they can incorporate them into future work. The significant benefit of presenting a set of underlying equations is that it provides a framework for future insights and datasets to align with. This clarity should be emphasised as much as possible in the concluding section of the manuscript.

Again, I have enjoyed reviewing this manuscript.

We thank the reviewer for his assessment.

Referee #3 (Remarks to the Author):

Revised Reviewer Comments

As I mentioned in my initial review, this manuscript focuses on the use of the trajectory model. The authors have made a commendable effort to recheck the results as requested and to clarify the methodology and its limitations. The inclusion of an additional SSP has increased the manuscript's value, and the comparison of precipitation and wind across different datasets is a useful contribution. The manuscript has improved considerably.

However, a number of important issues still remain and need careful revision before the manuscript can be considered for publication.

We are very grateful for this positive assessment of the reviewer and have responded to their additional comments below.

Major points:

- The UTrack model is not properly introduced. Its first mention (line 66) appears disconnected from the context. A clear introduction is needed, including a brief description and a reference to the Methods section.

Answer #15: We changed the text to the following (see II 69-75): “It is a new version of the Lagrangian moisture tracking model UTrack. UTrack constructs the spatial connections between evapotranspiration and precipitation by following the trajectories of moisture through the atmosphere. Instead of using reanalysis data, this new version builds on the output of the second version of the Norwegian Earth System Model (NorESM2) and allows for identifying scenario-dependent changes in atmospheric moisture transports throughout this century (see Methods).”

- Figure captions throughout the manuscript contain results, interpretations, or robustness statements that should be moved to the main text. Captions should be limited to figure description.

Answer #16: We agree with the reviewer, went through the captions of the figures 1 and 2, and moved results, interpretations and robustness analysis to the main text where appropriate (see e.g. II 154-162 and II 209-211).

- Several figures (e.g., S1–S3, S8–S12, S20) either lack clarity, omit explanation of elements (e.g., dashed circles, reference lines), or exclude SSP scenarios inconsistently. These require revision and, in some cases, expansion (e.g., SSP5-8.5 maps).

Answer #17: We have increased clarity and explanation of our supplementary figures.

Due to computational constraints, we cannot run all experiments for all scenarios but had to limit our analysis to the five extensive robustness analyses covered in the methods section and the supplementary figures (see methods section *Robustness checks*).

- The rationale for using SSP2-4.5 because it aligns with recent climate trajectories is important and must appear in the main text, not only in responses to reviewers.

Answer #18: We fully agree with the reviewer and have adapted our manuscript accordingly (see I 128-131).

- The methodology requires more context. The UTrack model should be discussed in relation to other Lagrangian approaches, acknowledging differences in results. Additionally, the choice of a 30-day trajectory period should be justified, as it exceeds typical atmospheric residence times.

Answer #19: We explain the 30-day trajectories below in Answer #43. For other Lagrangian approaches, see answers #40 and #58.

- The supplementary validation with ERA5 data requires clarification: (i) the meaning of “quasi-validation,” (ii) the reason for selecting only 5 years, and (iii) why only SSP2-4.5 was compared. Figure R10 should be added with an explanation.

Answer #20: Previously we used “quasi-validation”, because ERA5 itself is also a model product and therefore does not provide a true validation. However, to prevent confusion we changed our wording to “validation” in the Supplement. We also added an explanation for why we chose these specific years in II 153-154 in the Supplement, and the justification for SSP2-4.5 in I 162 (in SI). We have added Fig. R10 as Figure S17 (see also supplementary information II 228-229).

Minor points:

- References are missing in several places (e.g., SSPs, 10-year averages, optimistic deforestation scenarios).

Answer #21: We have added references to the respective sections. On SSPs, we use Riahi et al., 2022 (GEC). On the 10-year averages, we use Staal et al., 2025 (ESD), and the land-use change/deforestation scenarios of the SSPs are optimistic because they remain close to the historical level. Therefore, with the deforestation scenario used in the main manuscript, we can assess different levels of deforestation.

References:

- Riahi, K., Van Vuuren, D.P., Kriegler, E., Edmonds, J., O’neill, B.C., Fujimori, S., Bauer, N., Calvin, K., Dellink, R., Fricko, O. and Lutz, W., 2017. The Shared

Socioeconomic Pathways and their energy, land use, and greenhouse gas emissions implications: An overview. *Global environmental change*, 42, pp.153-168.

- Staal, A., Meijer, P., Nyasulu, M.K., Tuinenburg, O.A. and Dekker, S.C., 2025. Global terrestrial moisture recycling in Shared Socioeconomic Pathways. *Earth System Dynamics*, 16(1), pp.215-238.
- Acronyms (e.g., MAP, MCWD) should not be repeated once defined.

Answer #22: We agree with the referee and have used MAP and MCWD only.

- Redundant or unclear text appears at multiple points (e.g., lines 262–268, figure references).

Answer #23: We thank the reviewer for catching this redundancy and have accordingly reworked the text and figure references.

- Typographical and language issues persist (e.g., line 67 on NorESM2 data type).

Answer #24: We went through the manuscript and removed typographical and language errors where we found them, including on the lines that the reviewer outlined.

General comment:

The study calculates tipping points based on recycled precipitation. It should be stated more clearly that Amazon rainfall is not exclusively sourced from within the basin and that external moisture contributions could affect results. Please add references on Lagrangian studies of Amazon moisture sources.

Overall, the manuscript is promising but requires substantial clarification and restructuring, particularly regarding figure captions, model introduction, and methodological justification.

Point by point comments:

Answer #25: We thank the reviewer for their overall positive assessment and add references to Lagrangian studies of Amazon moisture sources (see answer #58). Please note that in our approach, precipitation keeps being sourced from outside the Amazon but how much exactly depends on the SSP, year and moisture recycling strength.

Overall, we thank the reviewer for their second very detailed read and are very happy for the additional feedback, which helped to further improve our manuscript. Also and in particular, for the detailed comments below.

1. Line 62: A reference on SSPs is required.

Answer #26: We have now added a reference to SSPs (see I 65).

2. The first mention of the UTrack model appears in line 66, with the phrase “This version...”. However, the model has not been introduced in the text before this point. The sentence reads disconnected from the context. The model should be properly introduced here. This paragraph must be carefully rewritten, providing at least a minimal introduction to UTrack. Readers should also be directed to the Methods section for a complete description.

Answer #27: We thank the reviewer and have responded to this concern in our answer #15.

3. Line 67: NorESM2 is a model that generates data, but what type of data? The sentence is poorly written and lacks precision.

Answer #28: We thank the reviewer and have clarified the sentence (see II 69-75).

4. Line 96: Fig S1 show which is referred in the sentence?

Answer #29: Extended Data Fig. 1 (formerly Fig. S1) shows the main moisture recycling network directions. We clarified our sentence (see I 102-103).

5. Figures S1–S2: Captions currently contain results and interpretations, which should be removed. For example, Fig. 1 includes comments such as “In this scenario, the tipping risk is very low.” These should be deleted. Check all figure captions throughout the manuscript and remove such comments.

Answer #30: We went through all the figures of the main manuscript and omitted unnecessary statements. For the supplementary figures, we keep a little bit more information as the captions should be self-contained and understandable from themselves.

6. Line 108: Figure S3 does not show SSPs.

Answer #31: Extended Data Figure 4 (formerly Fig. S3) shows MAP and MCWD (plus standard deviations) from the adaptation period, which is the historical period from 1950-2014. We indicated that in the new caption of Extended Data Fig. 4.

7. Lines 123–124: A reference is needed regarding the use of 10-year averages.

Answer #32: This is our time-window to average out the effect of single years. A suitable reference for this is Staal et al., 2025 (ESD), which we cite now in the manuscript.

8. General comment on captions: Statements concerning robustness must be included in the main text, not in captions. Comments, results, or references to important issues should be removed from captions and incorporated into the manuscript body.

Answer #33: We agree and have moved those statements into the main text, streamlining our figure captions substantially.

9. Figure 1 caption: The caption is unclear and must be rewritten. Panels d–f are not equivalent to panels a–c.

Answer #34: We have rewritten the caption of figure 1, clarifying that panels a-c are climate change only scenarios and panels d-f are climate change plus deforestation scenarios.

10. The map representations need more explanation in the text. Why is a certain percentage used in the tipped area? Figures 1d, e, and f do not show the same rationale.

Answer #35: A percentage is used to represent the likelihood that a certain cell is tipped or not. This likelihood is the frequency in percent that a certain cell in the Amazon forest transitions, which is calculated from the Monte Carlo ensemble, see methods). The panels d-f in Fig. 1 show the same as panels a-c but including not only climate change induced tipping/transitioning events but also deforestation. The deforested cells are shown as hatched regions.

We added more text to the main manuscript (see II 128-131).

11. Line 173: The term RCP appears for the first time here, within SSP-RCP-based. It should be introduced earlier, or changed.

Answer #36: We agree and have removed RCP.

12. Line 186. Confirm whether Fig. 1d actually illustrates the stated point.

Answer #37: This is a typo and should be Fig. 2b. We thank the reviewer for catching this.

13. Fig2: The dashed circle in panel c is not described in the caption.

Answer #38: We have added the description to the caption. It should have meant that the size of the dashed region is the region that is deforested on average.

14. Lines 262-268: Text is repeated. Please correct such errors carefully.

Answer #39: We thank the reviewer and have accordingly streamlined our text.

15. Methods: The UTrack model should be placed in the context of other Lagrangian methodologies. As mentioned in the first review, many techniques exist that do not always produce the same results. While UTrack is an appropriate choice, this discussion should be explicitly included.

Answer #40: We added the following to the methods: “Being a three-dimensional Lagrangian tracking model that reconstructs moisture trajectories using evaporation and precipitation directly, UTrack is conceptually similar to some other Lagrangian methods (e.g. Dey & Döös, 2020, GRL; Holgate et al., 202, J. Climate), but differs from other widely used tracking methods that are Eulerian (e.g. Kalverla et al., 2025, GMD) or follow changes in specific humidity instead (e.g. Sodemann, 2025, GMD).” See manuscript II 506-510.

References:

- Sodemann, H., 2025. The Lagrangian moisture source and transport diagnostic WaterSip V3. 2. *Geoscientific Model Development*, 18(22), pp.8887-8926.
- Kalverla, P., Benedict, I., Weijenborg, C. and Van Der Ent, R.J., 2025. Atmospheric moisture tracking with WAM2layers v3. *Geoscientific Model Development*, 18(14), pp.4335-4352.
- Dey, D. and Döös, K., 2020. Atmospheric freshwater transport from the Atlantic to the Pacific Ocean: A Lagrangian analysis. *Geophysical Research Letters*, 47(6), p.e2019GL086176.
- Holgate, C.M., Evans, J.P., Van Dijk, A.I.J.M., Pitman, A.J. and Di Virgilio, G., 2020. Australian precipitation recycling and evaporative source regions. *Journal of Climate*, 33(20), pp.8721-8735.

16. Line 468: A density plot of particle altitude distributions would be useful. If most particles are not near the surface, results could be misinterpreted.

Answer #41: In UTrack, moisture parcels get redistributed regularly along the atmospheric column. This is done such that every time step, every parcel has a chance of random relocation, such that it occurs on average once every 24 hours for any given parcel. The probability of the new vertical position is weighted by the vertical distribution of specific humidity across the 8 pressure layers. We added this information to the Methods in the previous iteration of the manuscript (II 471-476). Therefore, parcel altitude distributions follow specific humidity distributions. We believe it is not necessary to include specific humidity profiles from NorESM2 in our manuscript and we trust the reviewer agrees with that, given this explanation.

17. Lines: 472-476: The response in the “comment to reviewers” should be incorporated here. The distance traveled by particles is important to show their movement. Figure R4, currently in the supplementary material, is valuable and should be included.

Answer #42: We have added Fig. R4 to the Supplementary Information (now SI Fig. S18) and included a short explanation (see II 245-255 in the supplementary Note).

18. Line 480: Why was a 30-day period chosen? This is too long. The mean residence time of water vapor in the atmosphere is typically 10–15 days, which would be sufficient. This is particularly relevant for recycling processes, where residence times are even shorter.

Answer #43: In UTrack, moisture parcels lose moisture gradually along their trajectories. This means that some moisture may be allocated to precipitation very rapidly after parcel release and some moisture later along the trajectory - all the while the parcels still exist. Most moisture has been allocated to precipitation after 15 days and, indeed, atmospheric residence times of moisture are ~10 days according to UTrack. Both in reality and in the model environment, some of the moisture rains out later than after 15 days. Note that any parcel that has allocated 99% of its tracked evapotranspiration to precipitation will no longer be tracked. After 30 days, only very few parcels are still in the system. In other words, the maximum-30-day tracking does not assume residence times that long, but allows for tracking of moisture well beyond the average residence time, which results in

realistic estimates of residence time. Also note that every study that applied UTrack so far used these settings.

19. Line 488: The statement about good correspondence needs to be integrated into the main text, not just captions. Correlation coefficients (e.g., R^2 values) can be indicated in figures, but their statistical significance must be clearly discussed in the manuscript.

Answer #44: We thank the reviewer for this suggestion. We have expanded the discussion of the validation of our moisture recycling networks, including the correlation metrics, in the Supplementary Material. We keep this discussion there to present the validation results coherently in one place and avoid fragmenting the explanation across the manuscript.

20. Line 489: The absence of bias requires explanation. Please include this, as in Supplementary Material (Answer #11, Review 3).

Answer #45: As the Supplementary Figures S13 and S14 show, there are no meaningful systematic biases between ERA5 and NorESM2 when looking at Amazon basin recycling ratios and wind speeds. Of course, differences exist ($R^2 = 0.59$ and $R^2 = 0.84$, respectively), but our point was that our findings of cascading transitions are not an artefact of bias in NorESM2. This increases our confidence that our results truly reflect cascading transitions, as we explain in our manuscript. Forcing UTrack with ERA5 and NorESM2 yields a qualitatively consistent representation of the atmospheric hydrological cycle over the Amazon. To us, this means that both forcing datasets represent a physically and hydrologically coherent depiction of the Amazon and its functioning in the Earth system. We prefer not to go into depth in absences of bias, because we do not consider them relevant enough in the trade-offs with the readability of our manuscript.

21. Supplementary material. Supplementary Note: Validation of moisture recycling based on ERA5 reanalysis data.

- a. What is meant by “quasi-validation”?

Answer #46: We changed to “validation” (see our Answer #20).

- b. Why only for 5 years? Were modes of climate variability (e.g., ENSO) absent during this period?

Answer #47: We added the following in lines 153-154 of the Supplementary Note: “... 2015-2019, the period prior to that of our main results with temporal overlap between ScenarioMIP and ERA5”. There was no complete absence of climate variability during this period, but the fact that ERA5 and NorESM2 agree well for these years gives us confidence that interannual climate variability undermined our validation.

- c. Why was SSP2-4.5 chosen for comparison with ERA5 in the historical period? What about the other SSPs?

Answer #48: Because SSP2-4.5 is closest to the trajectory that the global climate has been on (Fricko et al., 2017; Lee et al., 2023). We added this information to I 162 of the Supplementary Note.

References:

- Fricko, O., Havlik, P., Rogelj, J., Klimont, Z., Gusti, M., Johnson, N., Kolp, P., Strubegger, M., Valin, H., Amann, M. and Ermolieva, T., 2017. The marker quantification of the Shared Socioeconomic Pathway 2: A middle-of-the-road scenario for the 21st century. *Global Environmental Change*, 42, pp.251-267.
- Lee, H., Calvin, K., Dasgupta, D., Krinner, G., Mukherji, A., Thorne, P., Trisos, C., Romero, J., Aldunce, P., Barret, K. and Blanco, G., 2023. IPCC, 2023: Climate change 2023: Synthesis report, summary for policymakers. Contribution of working groups I, II and III to the sixth assessment report of the intergovernmental panel on climate change. IPCC, Geneva, Switzerland.

- d. The sentence “UTrack by default is forced by hourly ERA5 data for 25 pressure layers at 0.25° spatial resolution instead of daily data for 8 pressure layers at 1.25° × 0.9375° in this study. Therefore, the time step is also set lower, to 0.25 hours instead of 4 hours is unclear. Please rewrite for clarity.

Answer #49: Thank you for pointing this out. We revised this text to: “In its original version, UTrack is forced by hourly ERA5 reanalysis data and, therefore, has time steps of 0.25 hours rather than 4 hours in this NorESM2-based study. For consistency, we set the time step of the NorESM2-forced validation runs to 0.25 hours as well.” (II 169-172 of the Supplementary Note)

- e. Figure R10 is illustrative. please include it in the supplementary material with explanatory text.

Answer #50: We have included Fig. R10 as SI Fig. S17 and added a short explanatory text (see II 228-229 in the Supplementary Note).

22. Line 496; why are different periods for SSPs?

Answer #51: The reason is that computational constraints have not allowed us to compute the full period from 2021-2099 for SSP1-2.6 but currently only the years 2090-2099 are available for SSP1-2.6.

23. In the response to reviewers, the authors state that “The rationale for using SSP2-4.5 was because it aligns most with the recent trajectory of the global climate.” This important rationale must be clearly stated in the manuscript.

Answer #52: We agree and have added it to the main manuscript (see I 130).

24. Fig S12: Where is the map for SSP5-8.5? Also, the statement “Overall, the SSP-based deforestation scenarios are very optimistic.” requires a reference.

Answer #53: The SSP scenario-based deforestation essentially halts deforestation at current levels (for SSP2-4.5 and SSP5-8.5, see dot-dashed line in Fig. S5c) never exceeding 20%, and only slightly increasing from 20 to 25% for SSP3-7.0 (see Fig. S5d). This is much more optimistic than the deforestation scenario we use in the main manuscript by Soares-Filho et al., 2013.

In order to avoid confusion, we have nevertheless removed the according sentence.

25. Lines 593: Acronyms MAP and MCWD MF have already been introduced; no need to repeat them here.

Answer #54: We agree and have removed the sentence.

26. Fig S8 and S9: why not for the other SSPs, as in S7a,b?

Answer #55: Due to computational constraints, we constrain this robustness analysis to SSP2-4.5 and SSP3-7.0 (see also our answer #17). Note that former Fig. S7 is now Extended Data Fig. 6

27. Fig S10: Results must be moved from the caption to the main text. In addition, the meaning of the horizontal line at 20% in Fig. S10a is not explained. Please delete “Fig. S15 shows the tipped area dependent on MAP and MCWD, given its respective moisture transport or global warming level.”

Answer #56: Several of our supplementary figures are robustness checks and should be understandable in a self-contained manner. Therefore, we prefer to present those in one place and avoid fragmenting across the manuscript and supplementary material.

28. Line 692: Results are reported in the caption of Fig. S20 but should be in the main text. All substantive results must be incorporated into the manuscript body, not left in figure captions.

Answer #57: The results of Extended Data Fig. 9 (formerly Fig. S20) are now also reported in the main text (see discussion in II 275-277, main results in II 154-162 and methods in II 751-760). However, our supplementary figures should also be understandable in a self-contained manner. Therefore, we prefer to keep the captions in the SI material.

29. General comment: The tipping point was calculated using recycled precipitation, but the text should make it clearer that the moisture for precipitation in the Amazon does not come solely from the region itself. The Amazon has other sources of moisture that could modify the precipitation calculated here. Add references to Lagrangian methods for moisture sources in the Amazon.

Answer #58: We made it more clear in the text that moisture in the Amazon is only partly internally supplied and recycled (see II 46-50). In the methods, we now also refer to other Lagrangian analyses of moisture trajectories (see II 506-510).

Additional comments from referee 2, Chris Huntingford:

Unfortunately, "tipping points" are often misused in climate science to mean anything that changes in a nonlinear way. As is well known, the precise (and correct) mathematical definition is:

1. A complete jump in state, which here would be a substantial or full transition from trees to climate-driven deforestation.
2. Hysteresis effects, where even if climate change is reversed, the system would not return to its original state (i.e., the rainforest would not be restored).

Therefore, it might be tempting to ask the authors to remove all mention of tipping points and instead describe the phenomena as "more rapid changes to savannah," or use the contexts suggested by reviewer #1.

However, if I recall correctly, the authors' underlying dynamical model did include the possibility of two distinct states under certain conditions. So it might not be appropriate to remove all references to tipping points.

To do justice to both the authors and the reviewer, it would be helpful to see the manuscript again. If I use the original link, I can no longer access the paper they submitted.

Answer #59: This assessment of reviewer #2 is correct and we agree with both reviewers (reviewer #1 and #2) that we should very explicitly define what we mean by tipping point. After careful consideration of all reviewer and editorial suggestions, we decided to remove the tipping point framing from the manuscript and instead speak of *transitions*. We changed the manuscript accordingly and also mention the limitation that we do not test for irreversibility and hysteresis in the discussion (II 242-246, see also our answer #3).

We have now included a definition of tipping points explicitly in the manuscript, which we use to simulate local-scale transitions. We include this in II 642-650 and Extended Data Fig. 2 (Fig. R1 below) and also outline it below. Based on the broad literature on tipping elements (e.g. the latest Global Tipping Points Reports 2025, 2023 and earlier key literature, e.g., Armstrong McKay et al., 2022, Science; van Nes et al., 2016, TREE), we define a tipping point as:

A change in state away from a forest state towards an alternative state. Beyond the tipping point, a destabilizing feedback drives the forest towards an alternative state in a self-reinforcing manner.

In our study, these state changes can occur on a local scale (Brando et al., 2025, ARER) and spread via the moisture recycling network.

Then, crossing such a tipping point oftentimes does (but not necessarily always, see e.g. van Nes et al., 2016, TREE; Armstrong McKay et al., 2022, Science) result in hysteresis. In many cases, the emergence of self-destabilizing feedbacks makes it very difficult to return to the original state because the feedback mechanisms that previously maintained stability are lost.

This irreversibility and hysteresis characteristics cannot be fully tested for by us because the forcing is not reversed in our scenarios (SSP2-4.5, SSP3-7.0, SSP5-8.5). This means that the backward direction (from alternative to forest state) cannot be tested for. However, the forward direction (from forest to alternative state) has been assessed in the literature (e.g. in Flores et al., 2024, Nature).

Based on the previous, we have selected our mathematical model of local-scale tipping points (hereafter called critical transitions) that can be crossed either due to deforestation or global warming induced impacts, aligning with reviewer #2 (see below from answers #59 on). The resulting stability landscape is represented in Fig. R1. In our study, each 1x1° grid cell is represented by such a stability landscape (local scale transition), which can be modelled with the following dynamical systems equation Eq. R1 (see first three terms of Eq. 2 in the manuscript):

$$\frac{dx_i}{dt} = -x_i^3 + x_i + C_{\text{crit},i}(\text{MAP}_i, \text{MCWD}_i) \quad (\text{Eq. R1})$$

Fig. R1: Map of the Amazon basin with a grid representing our study area. In this grid, each 1x1° grid cell is represented by a dynamical systems equation with two stable equilibrium states (forest and alternative state, see inset) that result from increasing drought (MCWD) and decreasing

rainfall (MAP), resembling the dynamical systems equation Eq. R1. We include this figure into our new version of the manuscript as *Extended Data Figure 2*.

This bistability on a local scale can then spread via the lack of moisture transport (domino effects) to regions far away (see full equation 2 of the main manuscript):

$$\frac{dx_i}{dt} = -x_i^3 + x_i + C_{\text{crit},i}(\text{MAP}_i, \text{MCWD}_i) + \sum_{k=1, k \neq i}^{N_{\text{grid cells}}} R_{ki}(\Delta\text{MAP}_{ki}, \Delta\text{MCWD}_{ki}) \frac{x_k}{2} \quad (\text{Eq. R2})$$

Fig. R1 is in line with the recent review by Brando et al. (2025, ARER) that has suggested local to regional tipping points instead of a global tipping point for the Amazon forest (summary point 2) that can spread via a domino effect (their summary point 3). We use this and show that under certain circumstances (i.e. global warming of 1.5-2.0°C plus deforestation 22-28%), local transitions (tipping events) can spread to large-scale (system-relevant) tipping events across substantial areas of the Amazon forest. In particular, the latest assessments by Flores et al. (2024, Nature) and Brando et al. (2025, ARER) provide observational and Earth System modelling efforts that support the notion of a true tipping point on local scales - in particular regarding the largest and most vulnerable trees in the Amazon (see our answer #3).

Again, in summary, as we cannot fully test for hysteresis and irreversibility, we decided to remove the tipping points framing and use *transitions* instead.

References:

- Lenton, T.M., Milkoreit, M., Willcock, S., Abrams, J.F., Mc Kay, D.L.A., Buxton, J.E., Donges, J., Loriani, S., Wunderling, N., Alkemade, F. and Barrett, M., 2025. Global tipping points report 2025.
- Armstrong McKay, D.I., Staal, A., Abrams, J.F., Winkelmann, R., Sakschewski, B., Loriani, S., Fetzer, I., Cornell, S.E., Rockström, J. and Lenton, T.M., 2022. Exceeding 1.5 C global warming could trigger multiple climate tipping points. *Science*, 377(6611), p.eabn7950.
- Brando, P.M., Barlow, J., Macedo, M.N., Silvério, D.V., Ferreira, J.N., Maracahipes, L., Anderson, L., Morton, D.C., Alencar, A., Paolucci, L.N. and Jacobs, S., 2025. Tipping points of Amazonian forests: beyond myths and toward solutions. *Annual Review of Environment and Resources*, 50.
- Flores, B.M., Montoya, E., Sakschewski, B., Nascimento, N., Staal, A., Betts, R.A., Levis, C., Lapola, D.M., Esquivel-Muelbert, A., Jakovac, C. and Nobre, C.A., 2024. Critical transitions in the Amazon forest system. *Nature*, 626(7999), pp.555-564.

- Van Nes, E.H., Arani, B.M., Staal, A., van der Bolt, B., Flores, B.M., Bathiany, S. and Scheffer, M., 2016. What do you mean, 'tipping point'? *Trends in Ecology & Evolution*, 31(12), pp.902-904.

Here are some points that might help:

1. A key reason why I liked the paper is because Equation (2) is of the form $dx/dt = -x^3 + x + \text{forcing}$.
2. This is a typical mathematical description of a tipping point, and it is commendable that the authors have used such a dynamical systems model to try to map onto.
3. Specifically, this formulation allows [A] a jump to a different state, but also [B] hysteresis, so lowering drought forcing by reversing climate change might not bring back the rainforest. A true tipping point has to have both [A] and [B].
4. If the authors have confirmed that, locally, the rainforest maps onto this equation, then in my view, they are close to confirming the rainforest can "tip" locally. So in some places, using the phrase "tipping point" might be valid.

Answer #60: This is indeed the equation that we have used and also collected multiple lines of evidence that local scale tipping points have been monitored in experimental and observational studies, summarized in two recent reviews (Flores et al., 2024; Brando et al., 2025, ARER), see also our answer #59 above.

References:

- Flores, B.M., Montoya, E., Sakschewski, B., Nascimento, N., Staal, A., Betts, R.A., Levis, C., Lapola, D.M., Esquivel-Muelbert, A., Jakovac, C. and Nobre, C.A., 2024. Critical transitions in the Amazon forest system. *Nature*, 626(7999), pp.555-564.
- Brando, P.M., Barlow, J., Macedo, M.N., Silvério, D.V., Ferreira, J.N., Maracahipes, L., Anderson, L., Morton, D.C., Alencar, A., Paolucci, L.N. and Jacobs, S., 2025. Tipping points of Amazonian forests: beyond myths and toward solutions. *Annual Review of Environment and Resources*, 50.

This brings two issues:

Firstly, the local confirmation might be incomplete if we want to include the possibility of hysteresis. While the authors may have demonstrated locally that vegetation can jump from trees to grasses as drying increases, I don't think they have confirmed the hysteresis aspect. The reason for this is that the climate change scenarios they use all assume increasing change – there is no "overshoot" scenario used for reversing climate change.

Answer #61: This is a correct assessment of the reviewer that we cannot fully test for hysteresis in this manuscript as none of our scenarios is an overshoot scenario. However, while all our scenarios (SSP2-4.5, SSP3-7.0, and SSP5-8.5) are no overshoot scenarios, we can test for alternating rainfall and MCWD conditions (i.e. decreasing and later increasing rainfall) within a single scenario, e.g. SSP2-4.5. In this scenario, we find throughout the 2050s (see Fig. 1a, timeseries below the map) that moisture conditions (rainfall and MCWD) could push parts of the forest across the tipping point and transition towards an alternative state. Then, later in the century (2060s and 2090s), the rainfall and MCWD-values recover to values below large-scale transitions for the Amazon forest. This means that under global warming alone (without deforestation), Amazon transition risks are still comparably low following an SSP2-4.5 even though the system may be partially beyond local tipping points for some time. By contrast, a qualitative shift occurs under the higher emission scenarios (SSP3-7.0 and SSP5-8.5). Once global warming exceeds approximately 3.5°C, the Amazon does not return to a state with a low tipped area if global warming continues to rise to or beyond 4.0°C. In other words, beyond 3.5°C, there is no rainfall and MCWD-condition that allows for recovery or reversibility (see Fig. 1b and c, timeseries).

In summary: We agree with the reviewer that we do not test for timescales in this manuscript (i.e. do not fully consider overshoots) because (i) neither of our scenarios is a true overshoot scenario and (ii) we would require transition timescales of Amazon forest tipping/transitioning. As the reviewers note above, the latter are quite uncertain and likely on the order of decades to centuries. Instead we run equilibrium experiments.

However, if we wanted to investigate pathways, the following two dimensions would be needed:

1. Global Warming Overshoots assessing the impact of global warming decreases
2. Deforestation Overshoots assessing the impact of restoration efforts

This goes beyond the current manuscript but is a worthwhile future effort.

We add a short discussion on the points above to the manuscript (see II 242-246).

The second issue is whether evidence of local tipping points indicates that the Amazon as a whole is reaching a tipping point. I would argue that this is valid, providing enough locations do “tip over”. On some of the red curves, Figure 1 (d-f), the changes do reach very high percentages.

Answer #62: We agree with this notion of the reviewer since each of our 1x1° grid cells is a local-scale tipping element. So, local-scale tipping events (due to global warming or deforestation) can spread via the lack of moisture transport to distant regions (see Fig. 1b-f). However, following all the reviewer comments and editorial suggestions, we remove the tipping point framing but only keep it where it is very solidly defined (at the local scale). Otherwise we speak of *transitions*.

So, I don't believe any of this should prevent publication. However, I agree with the reviewer that the phrase "tipping point" should be used carefully. At the same time, there are parts of the paper that come very close to a rigorous description of a tipping point, making it valid to use this terminology.

Answer #63: We agree that we need to define what we mean by tipping point very clearly, which is the definition of the reviewer as noted above (see our answer #59), and as we cannot test for hysteresis, we follow the combined advice of reviewer #1 and #2 (and the editorial suggestion), and remove the tipping point framing from the manuscript and instead speak of *transitions*.

OK, in summary, I think everything is broadly fine, but could the authors:

1. Use the expression "tipping point" sparingly.
2. Please promote this notation where the proposed underlying model of Equation (2) is presented. This brings rigour to the discussion of why we can talk about tipping points.
3. Acknowledge that a full test of tipping is needed to confirm hysteresis effects. When climate "overshoot" simulations become available, this further work can be undertaken. So, please make a note like this in caveats.
4. Also acknowledge that the existence of "local tipping points" (e.g., projected for single grid boxes) does not automatically imply a full Amazon tipping point. However, if enough points do "tip" (e.g., Fig 1d,e,f), then I believe it is valid to say, overall, that "There is a risk the Amazon will pass through a tipping point."

As long as there are various caveats around the words "tipping points", I think it is fine to keep that title as it is. Especially considering point (4) just above.

Answer #64: We now establish a clear definition of the tipping point framing and provide clear reference to Equation 2 in our manuscript. We also note that we do not investigate overshoots in this manuscript (while worthwhile for the future). Lastly, we have also reworked our justification regarding the evidence of local-scale tipping points (the detailed responses can be found above in the answer #59).

Many climate research studies refer to any rapid transition between states as a "tipping point". This is despite the formal mathematical description of (1) a jump in state, and (2) hysteresis effects.

In my view, it would be a shame to pick on this paper to correct for the notation that many others are misusing. Having said that, the paper could get things right by:

1. Using "Tipping point" a little less towards the beginning.
2. Recognising that their Eqn (2) does have the true tipping features.

3. Noting, as mentioned earlier, the hysteresis effects are not tested as the Earth System Models are not routinely operated for "overshoot" scenarios.

Answer #65: We thank the reviewer for his kind feedback and providing a clear pathway to improve our manuscript. We have followed this advice in the response letter (see answers above) and the manuscript.

With this paper, the authors have the opportunity to actually say something like "Here, we use a tipping point terminology loosely, illustrating a relatively rapid transition from one state to another". Then a sentence around Eqn (2) saying "this equation can admit solutions in the strictest sense of a tipping point, with a jump in state and hysteresis".

Then finally in the Conclusions, something like "Eqn (2) can allow for hysteresis, and future analysis can be to extend the research presented here to climate overshoot scenarios, to test whether reversing climate change will not necessarily cause a return in rainforest cover".

Answer #66: We thank the reviewer for his suggestions and have followed the recommendations of the reviewer (see also our responses above).

I have thought about this a bit more.

My feeling remains that the manuscript should be accepted, as long as the authors add a few additional sentences of:

1. Where the text is using simply a rapid but transition from trees to grasses, then state "Using tipping point notation loosely"
2. Around Eqn (2), say this equation has the capability to allow its solution for tipping points in the strict sense: "this equation can admit solutions that are strictly tipping points, i.e. a jump to another state, and hysteresis effects".
3. Add a caveats sentence that "future ESM climate calculations simultaneous with deforestation scenario would ideally include the possibility of climate overshoot, thereby testing hysteresis effects".

Answer #67: We have included these additional suggestions of the reviewer and included them in our manuscript.

Let me know what you decide! Climate-induced Amazon dieback remains so important to understand, and especially in tandem with deforestation. The authors do well at including both effects, making it one of the most complete studies so far.

Answer #68: We thank the reviewer for his positive assessment of our work.

Dear Reviewers, dear Editor,

We are very grateful for the detailed comments by the reviewers that have substantially improved our work. We have considered all final feedback points raised by the reviewers. Please find below a point-by-point response to the comments. All improvements are colored in blue in the revised manuscript. We are grateful for this opportunity to improve our manuscript.

Sincerely yours,

Nico Wunderling, Boris Sakschewski, Johan Rockström, Bernardo Flores, Marina Hirota, and Arie Staal

Referee #1 (Remarks to the Author):

The authors have done substantial work to address my previous comments, and the manuscript is considerably improved. The framing is now more nuanced, moving away from the stronger claims of the tipping point framework. However, I remain skeptical of some of the core elements of the study.

My primary concern is that the binary representation of vegetation states — either forest or savanna/dry forest — in the model is a huge oversimplification. Given the well-documented structural and functional complexity of Amazonian ecosystems, including the continuum of forest types, transitional vegetation, and the spatial heterogeneity of climate and soil conditions, I find it difficult to imagine that such a simplification can capture the dynamics of these systems. This concern is compounded by the absence of robust model validation, lack of uncertainty for projected changes in precipitation, and unrealistic deforestation scenarios, which make it difficult to assess the reliability of the projected transitions.

That said, I recognize this as an important contribution that addresses a major scientific gap. The study demonstrates, under a set of clearly stated and strong assumptions, that combined changes in regional climate driven by deforestation and global climate change could push parts of the Amazon toward non-forest states. This is a major finding that warrants attention, provided readers interpret the results with an awareness of the model's limitations.

We thank the reviewer for their assessment and guidance during the last two rounds of reviews. Based on their comments above, we are explicitly summarizing the simplifications in our revised manuscript (see II 255-256).

Referee #2 (Remarks to the Author):

Thank you for asking me to look again at the responses to reviewer requests for the paper: "Pinpoint Amazon forest transition under global warming and deforestation".

I can see that the authors have again taken the reviewers' suggestions seriously (both mine and those of Reviewer #1 and Reviewer #2), and there are no further comments on the responses. They look fine.

Climate change continues to suffer from a mismatch between (1) Earth System Models, which differ in their projections but many indicate tipping points ahead (including, potentially, for the Amazon) and (2) the theory of nonlinear dynamical systems, which characterises well the mathematics of tipping caused by a change in a bifurcation parameter.

Unfortunately, to date, very few researchers have linked (1) and (2) above, which is a real shame. By making this link, the dynamical system becomes available for other researchers to test different forcing profiles (e.g. climate change "overshoot"), better understand what the tipping point might look like, and potentially use a calibrated dynamic system to develop early warning systems.

Hence, I have always liked this manuscript because Wunderling et al. make that link through their Equation (2). That equation is very clearly a dynamical system, and the "cubic" term gives the potential for a tipping point, along with hysteresis in an overshoot scenario.

We thank the reviewer for their appreciation of our manuscript.

So, I have one very final small request. To ensure this manuscript is picked up by the mathematical community and recognised as linking the two disciplines (climate science and nonlinear dynamics), could the authors add a sentence or two that makes this explicit?

Somewhere near Equation (2), something like: "This nonlinear equation is a typical dynamical system that can exhibit tipping point behaviour, and where the C_{crit} , and the Summation terms can be interpreted as a time-evolving bifurcation parameter". That then gets the keywords of "nonlinear, tipping point, dynamical system and bifurcation" together.

We agree with the reviewer, have added their suggestion (see II 651-653), and hope the paper will be taken up by the mathematical community on nonlinear dynamics.

Otherwise, this is a superb paper, and I have enjoyed assessing it. I very much hope Nature will now formally accept the manuscript.

We thank the reviewer for their enthusiastic assessment of our work.

Referee #3 (Remarks to the Author):

Overall, the manuscript has improved substantially in clarity, structure, and methodological transparency. The revisions have addressed the main concerns raised previously, and the paper is now much stronger.

We thank the reviewer for their positive assessment of our work.

Only a pair of minor issues remain:

Answer #15 -> Lines 70-71: The description of UTrack as “following the trajectories of moisture through the atmosphere” is somewhat misleading. In a strict Lagrangian sense, the model does not track moisture itself as an independent physical entity. Rather, it follows the trajectories of air parcels, to which specific humidity values are assigned based on reanalysis or model output fields. The moisture content is therefore not dynamically resolved along an independent moisture trajectory, but diagnostically attributed to moving air parcels using externally provided meteorological data.

I recommend clarifying this distinction to avoid conceptual ambiguity, particularly for readers from an atmospheric dynamics background, for whom the difference between tracking moisture as a substance and advecting air parcels with assigned humidity is non-trivial.

This is correct. We have clarified this distinction in the revised manuscript (see II 67-69).

Line 46: "Part of the Amazon's precipitation is externally sourced ... " , this sentence needs a citations: e.g. Gimeno et al., 2012 and Sorí, et al., 2017

Gimeno, L., Stohl, A., Trigo, R. M., Dominguez, F., Yoshimura, K., Yu, L., Drumond, A., Durán-Quesada, A. M., & Nieto, R. (2012). Oceanic and terrestrial sources of continental precipitation. *Reviews of Geophysics*, 50(4), RG4003. <https://doi.org/10.1029/2012RG000389>

Sorí, R., Nieto, R., Drumond, A., Gimeno, L., & Vicente-Serrano, S. M. (2017). A Lagrangian perspective of the hydrological cycle in major global river basins. *Earth System Dynamics*, 8(4), 1009–1022. <https://doi.org/10.5194/esd-8-1009-2017>

We thank the reviewer for their additional references, which we are happy to include with the second reference being slightly adapted to be accessible (see I 44).